# A novel anti-epileptogenesis strategy of temporal lobe epilepsy based on nitric oxide donor

Xian-Hui Zhu[1,2,13], Ya-Ping Zhou[1,3,13], Qiao Zhang [ID][1,4,13], Ming-Yi Zhu[1,5,13], Xiao-Wei Song [ID][2,6], Jun Li[7], Jiang Chen[8], Yun Shi [ID][8], Kang-Jian Sun[9], Yong-Jie Zhang[10], Jing Zhang[1,11], Tian Xia[1✉], Bao-Sheng Huang [ID][2✉], Fan Meng [ID][1,11✉] & Qi-Gang Zhou [ID][1,11,12✉]

## Abstract

**The molecular mechanism underlying the role of hippocampal hilar interneuron degeneration in temporal lobe epilepsy (TLE) remains unclear. Especially, very few studies have focused on the role of neuronal nitric oxide synthase (nNOS, encoded by *Nos1*) containing hilar interneurons in TLE. In the present study, *Nos1* conditional knockout mice were constructed, and we found that selective deletion of *Nos1* in hilar interneurons rather than dentate granular cells (DGCs) triggered epileptogenesis. The level of nNOS was downregulated in patients and mice with TLE. *Nos1* deletion led to excessive epilepsy-like excitatory input circuit formation and hyperexcition of DGCs. Replenishment of hilar nNOS protein blocked epileptogenic development and memory impairment in pilocarpine-induced TLE mice. Moreover, chronic treatment with DETA/NONOate, a slowly released exogenous nitric oxide (NO) donor, prevented aberrant neural circuits of DGCs and the consequent epileptogenesis without acute antiseizure effects. Therefore, we concluded that NO donor therapy may be a novel anti-epileptogenesis strategy, different from existing antiseizure medications (ASMs), for curing TLE.**

**Keywords** Epileptogenesis; Neuronal Nitric Oxide Synthase; Hilar Interneurons; Dentate Granule Cells; Pilocarpine
**Subject Category** Neuroscience

## Introduction

Epilepsy is the second most prevalent neurological disorder in the world (Cano et al, 2021). Temporal lobe epilepsy (TLE), the most frequent form of epilepsy, is characterized by spontaneous recurrent seizures (SRS) arising from seizure foci such as the hippocampus in the temporal lobe (Hesdorffer et al, 2011). About 70% of TLE patients eventually suffer from drug-resistant epilepsy (DRE), defined as resistance to clinically used antiseizure medications (ASMs). The majority of them, especially those with hippocampal sclerosis, require surgical resection (Loscher et al, 2020; Schmidt and Loscher, 2005). Hippocampal sclerosis is one of the most common symptoms in TLE/DRE patients (Sisodiya, 2003; Sisodiya and Bates, 2006), which is ascribed to different patterns of cell loss, such as the death of interneurons and mossy cells in the hippocampal hilus, and gliosis (Rusina et al, 2021). Besides, multiple histopathological changes, such as synaptic reorganization, altered neurogenesis in the dentate gyrus (DG) of the hippocampus and mossy fiber sprouting, may also contribute to the development of hippocampal sclerosis (Casillas-Espinosa et al, 2012; Polli et al, 2014; Rusina et al, 2021). A statistical correlation is found between the hippocampal sclerosis and TLE, namely, the presence of hippocampal sclerosis suggests higher probability of TLE falling into the category of pharmacoresistant or medically intractable epilepsy (Majores et al, 2007; Rakhade and Jensen, 2009). Synaptic reorganization of dentate granular cells (DGCs) has been shown as a characteristic change in the hippocampus of TLE/DRE patients (Kobayashi et al, 2011; Nearing et al, 2007). Therefore, research has focused strongly on the role of DG in the development of chronic TLE/DRE (Kahn et al, 2019).

DGCs, with apical dendrites growing towards the molecular layer and axons extending into the hilus and CA3 regions, are the principal cells in the granule layer of DG (Jessberger et al, 2007; Zhou et al, 2019). Increasing evidence supports that DGCs could be "detonators", which, together with CA3, form a focus of seizures (Krook-Magnuson, 2017; Scharfman, 2019). Abnormal morphological development and circuit formation of DGCs have been described in different types of mouse models of epilepsy (Du et al, 2017; Pun et al, 2012). Mossy fiber sprouting, ectopic migration, extension of basal dendrites into the

[1]Department of Clinical Pharmacology, School of Pharmacy, Nanjing Medical University, Nanjing 211166, China. [2]Sir Run Run Hospital, Nanjing Medical University, Nanjing 211166, China. [3]Changzhou Hygiene Vocational Technology College, Changzhou 213002, China. [4]Suqian First Hospital, Nanjing Medical University, Suqian 223800, China. [5]Centre of Medicinal Preparations, Institute of Dermatology, Chinese Academy of Medical Sciences & Peking Union Medical College, Nanjing 210042, China. [6]Department of Neurosurgery, The Second Affiliated Hospital of Nantong University, Nantong First People's Hospital, Nantong 226001, China. [7]Department of Pharmacy, Nanjing First Hospital, Nanjing Medical University, Nanjing 210006, China. [8]Department of Neurology, Affiliated Drum Tower Hospital of Nanjing University Medical School, Nanjing University, Nanjing 210032, China. [9]Department of Neurosurgery, Nanjing Jinling University, Nanjing 210002, China. [10]Department of Human Anatomy, Human Brain Tissue Resource Center of Nanjing Medical University, National Health and Disease Human Brain Tissue Resource Center-sub-center of Nanjing Medical University, Nanjing 211166, China. [11]The Key Center of Gene Technology Drugs of Jiangsu Province, Nanjing Medical University, Nanjing 211166, China. [12]Department of Pharmacy of First Affiliated Hospital of Nanjing Medical University, Nanjing 210029, China. [13]These authors contributed equally: Xian-Hui Zhu, Ya-Ping Zhou, Qiao Zhang, Ming-Yi Zhu. ✉E-mail: 137012558@qq.com; bs.huang@njmu.edu.cn; mengfan199005@njmu.edu.cn; qigangzhou@njmu.edu.cn

hilus, and altered spine density of DGCs are observed after some pathological conditions including status epilepticus (SE) (Tejada and Roque, 2014). Elevated excitatory inputs onto mature DGCs and adult-born DGCs, mainly from the entorhinal cortex and forebrain, contribute to epileptogenesis in experimental TLE (Du et al, 2017; Zhou et al, 2019). The reorganization of the DGCs may contribute to pathological conduction pathways for synchronous discharges in the temporal lobe (Sparks et al, 2020). However, the molecular mechanism underlying the aberrant circuits of DGCs and the consequent epileptogenesis remains indistinct, hindering the development of new therapeutic strategies for epilepsy (Pitkanen and Lukasiuk, 2009).

A reduced number of GABAergic inhibitory cells, including peptide-containing dendritic interneurons, in the hilus has been observed in chronic animal models of epilepsy and human epileptic hippocampus (Gorter et al, 2001; Sundstrom et al, 2001), which is believed to contribute to the persistence of SRS. Grafting fetal GABAergic cells or inhibitory interneuron progenitors into epileptic foci produces antiepileptic effects in a variety of animal models (Henderson et al, 2014; Hunt et al, 2013). Therefore, the loss of inhibitory interneurons is presumed to be a key factor underlying the increased excitability of the epileptic hippocampus. Besides PV-, SST-, or NPY-containing interneurons, neuronal nitric oxide synthase (nNOS, encoded by *Nos1*) containing interneurons is another subtype of GABAergic neurons in the hilus (Marx et al, 2013) with little attention. The nNOS-positive (nNOS$^+$) interneurons account for half of hilar GABAergic neurons in rodents (Jinno et al, 1999; Liang et al, 2013). However, the role of nNOS$^+$ interneurons in the pathology of TLE remains unclear.

As the smallest signaling molecule ever known, nitric oxide (NO) is produced by three distinct isoforms of nitric oxide synthase (NOS), including inducible NOS (iNOS), endothelial NOS (eNOS) and nNOS, which utilize L-arginine and molecular oxygen as substrates. iNOS can be expressed in various cells in response to stimuli and contributes to the inflammatory pathological process. eNOS is mostly expressed in endothelial cells. nNOS is constitutively distributed in neurons, which mainly regulates synaptic plasticity, neuronal excitability, learning, memory, and neurogenesis in the central nervous system (CNS) (Wan et al, 2024), as well as blood pressure, smooth muscle relaxation and vasodilatation in the peripheral nervous system (PNS) (Forstermann and Sessa, 2012). Apparently, nNOS is the major synthetic enzyme of NO in the CNS taking part in many pathophysiological processes in the brain.

In this study, a significant low level of nNOS was found in the hippocampus of TLE/DRE patients. In addition, nNOS expression and NO content in the DG dropped in pilocarpine-induced epileptic mice. Selective deletion of *Nos1* from hilar interneurons was sufficient to induce SRS in about 2 months. Lack of nNOS caused epilepsy-like hyperexcitatory afferent circuit integration and higher excitability of DGCs. Replenishment of hilar nNOS by lentivirus (LV) encoding full-length cDNA of nNOS transduction, or transplantation of GABAergic progenitors derived from medial ganglionic eminence (MGE) containing or not containing nNOS, showed an essential role of hilar nNOS deficiency in epileptogenesis of the mouse model of pilocarpine-induced TLE. More importantly, the results of this study confirmed a critical role of NO in the hilus for aberrant circuit formation of DGCs and thereby the development of SRS in epilepsy, based on which a novel treatment strategy for TLE was developed by applying NO donors. Collective evidence

suggests that DG may be the "brain gates" of TLE, making it necessary to control the pathological changes of the DG in TLE (Kahn et al, 2019). This study could provide fresh insights into a potential new approach to antiepileptic therapy by blocking the progression of pathological development in the DG of TLE patients. Unlike ASMs currently used in the clinic, NO donors could be developed as a new type of anti-epileptogenesis drug.

# Results

## nNOS-NO level significantly declines in epileptogenic hippocampus of patients and mice with TLE

Surgical specimens of the hippocampus were collected from patients who were diagnosed with DRE based on magnetic resonance imaging (MRI) fluid attenuated inversion recovery (FLAIR) images, electrocorticography (ECoG), and stereoelectroencephalography (SEEG) and underwent therapeutic resection (Fig. 1A–C and Table EV1). The representative sample of a patient showed left hippocampal sclerosis (Fig. 1A). ECoG recording showed frequent spike-and-wave discharges in the left temporal lobe (Fig. 1B). SEEG monitoring confirmed that the left hippocampus exhibited abnormal discharges and served as a seizure focus (Fig. 1C). Then, RNA-Seq was applied to examine the transcriptome of the hippocampus in DRE patients. The control hippocampus was collected from patients who died from cancers without epilepsy experience (Table EV1). In comparison with that in the controls, the transcriptional level of *NOS1* was significantly downregulated (Fig. 1D). Subsequently, the RNA-seq findings were verified by RT-qPCR and western blot, which also showed obvious reduced mRNA and protein levels of nNOS in hippocampus tissues resected from DRE patients in comparing with control patients (Fig. 1E,F).

To determine whether nNOS deficiency is involved in pilocarpine-induced epilepsy in mouse models, we measured hippocampal nNOS expression by RT-qPCR and western blot in the DG of the hippocampus at different time points after pilocarpine-induced SE (275 mg/kg, s.c., 1 time) (Fig. 1G–J). The data of RT-qPCR analysis showed that the mRNA level of *Nos1* began to decline significantly at 7 d and maintained stable at 14 d and 2 months after pilocarpine administration (Fig. 1G). However, no obvious change was observed in the mRNA level of *Nos1* during acute phases including 2, 12, 24, and 48 h after pilocarpine administration (Fig. 1G). Moreover, mRNA levels of iNOS and eNOS, the other two isozymes of nNOS, remained unchanged at 7 d after pilocarpine administration (Fig. 1H). Western blot analysis of nNOS protein supported the same conclusion: the protein expression of nNOS in the DG deceased at 7, 14, and 60 d, but remained unaltered at 3, 12, 24, and 48 h after pilocarpine administration (275 mg/kg, s.c., 1 time) (Fig. 1I,J). The results of immunofluorescence staining using an anti-nNOS monoclonal antibody showed that this decline was attributed to the loss of nNOS-expressing interneurons in the hilus (Fig. 1K). The number of nNOS$^+$ cells in the CA1 and CA3 regions was not affected (Fig. 1K).

Consequently, NO production in the DG was significantly reduced at 7 and 60 d after pilocarpine-induced SE (Fig. EV1A,B), resulting in decreased levels of nitrotyrosine-modified proteins in

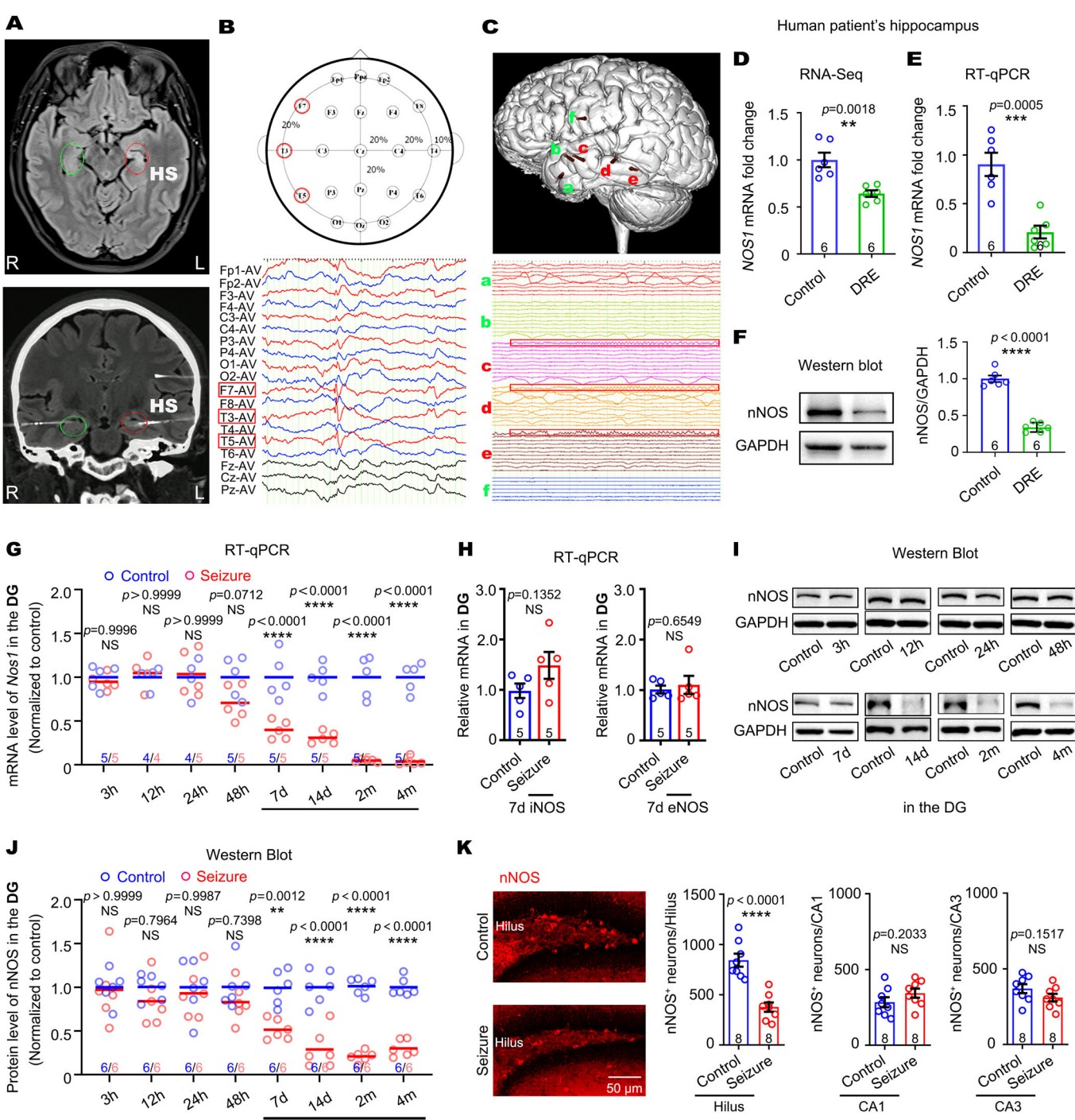

the DG at 60 d (Fig. EV1C,D). In addition, the other two mouse models of kainic acid (KA)- and pentylenetetrazol (PTZ)-induced epilepsy also displayed significant decreases in nNOS and NO levels (Fig. EV1E,F).

## Lack of nNOS in the DG leads to elevated susceptibility to pilocarpine

To probe into whether hippocampal nNOS deficiency plays a crucial role in the pathology of epilepsy, we treated *Nos1* knockout (nNOS KO, *Nos1*$^{-/-}$) mice and their wild type (WT) littermates with different dosages of pilocarpine (Fig. 2). Surprisingly, a regular dose of pilocarpine (275 mg/kg, s.c.) for the establishment of a chronic epilepsy model in WT mice resulted in 100% mortality in *Nos1*$^{-/-}$ mice within 1 h due to severe convulsion attacks, indicating abnormal acute hypersensitive reaction of *Nos1*$^{-/-}$ mice to pilocarpine treatment. Therefore, we reduced the dosage of pilocarpine to 230 mg/kg and 100 mg/kg. Seizures were scored (rank 1–5) every 20 min for 3 h following pilocarpine administration, and it was found that the cumulative seizure score was

**Figure 1.    nNOS is reduced in epileptogenic hippocampus of patients and TLE mouse models.**

(**A**) MRI FLAIR images of a patient (case 1) showing a left hippocampal sclerosis (HS). L: left; R: right. (**B**) Interictal spike-wave discharges was recorded by the ECoG using the International 10–20 EEG system in the left temporal lobe. The F7, T3, and T5 electrodes captured EEG discharges in the left anterior temporal lobe, middle temporal lobe, posterior temporal lobe, and adjacent brain regions, respectively. (**C**) Position of deep brain electrodes and their discharges on the brain surface in SEEG monitoring. 'Electrode c' ran from the anterior part of the left middle temporal gyrus to the head of the hippocampus, 'electrode d' from the middle part of the left middle temporal gyrus to the hippocampus body, and 'electrode e' from the posterior part of the left middle temporal gyrus to the hippocampus tail. The topmost channel of each electrode represented the innermost contact, and the bottommost channel represented the outermost contact. Thus, the topmost channels of c, d, and e reflected the head, body, and tail of the hippocampus, respectively, showing discharges from the left hippocampus. (**D**) RNA-Seq analysis showing reduced nNOS transcription in the hippocampus of patients with DRE. Students' $t$-test, $n = 6$ subjects. (**E**) RT-qPCR analysis confirming low levels of *NOS1* in the hippocampus of patients with DRE. Students' $t$-test, $n = 6$ subjects. (**F**) Western blot analysis showing low levels of nNOS protein in the hippocampus of patients with DRE. Students' $t$-test, $n = 6$ subjects. (**G**) The data graph showed the mRNA level of *Nos1* in DG measured by qPCR 3, 12, 24, 48 h, 7, 14 days, or 2 months after pilocarpine-induced SE. Two-way ANOVA, $n = 4$–6 mice. (**H**) qPCR data showed no change in the levels of iNOS and eNOS mRNA in DG 7 days after pilocarpine-induced SE. Students' $t$-test, $n = 5$ mice. (**I, J**) Western blot analysis showed the protein level of nNOS in the DG 3, 12, 24, 48 h, 7, 14 days, or 2 months after pilocarpine-induced SE. Two-way ANOVA, $n = 6$ mice. (**K**) Representative photos of nNOS immunoreactivity in the DG of control and pilocarpine-induced SE. Data graph showing the number of nNOS$^+$ cells in the HL, CA1, and CA3 regions. Students' $t$-test, $n = 8$ mice. Error bars correspond to ± s.e.m. **$P < 0.01$, ***$P < 0.001$, ****$P < 0.0001$, NS, not significant. Source data are available online for this figure.

significantly higher in *Nos1*$^{-/-}$ mice than in their control littermates at both the high dose (230 mg/kg, Fig. 2A) and low dose (100 mg/kg, Fig. 2B). Next, electroencephalography (EEG) was adopted to determine whether *Nos1* deletion could lead to epileptogenesis by implanting electrodes into the DG of the hippocampus and the cortex (Fig. 2C). Spontaneous epileptic-like spikes, but not SRS, were recorded in the DG, but not in the cortex, of *Nos1*$^{-/-}$ mice (Fig. 2C). In the chronic pilocarpine model with a lower dose of pilocarpine (200 mg/kg, s.c.), *Nos1*$^{-/-}$ mice developed more epileptic spikes (SPKs) and SRS, while control littermates had only a few spikes without SRS at 2.5 months after SE induction (Fig. 2D). Notably, local SRS was found in the DG of the hippocampus rather than in the cortex at 45 d after pilocarpine administration (Fig. 2E), providing an important clue that epileptogenesis in pilocarpine-treated *Nos1*$^{-/-}$ mice may originate from the DG of the hippocampus.

Thus, nNOS in the DG was then knocked down by constructing LV vector expressing small hairpin RNA (shRNA) of nNOS (LV-nNOS-RNAi-GFP) or nonsense sequence (LV-scramble-control-GFP) as negative control and injected into the hilus (Fig. EV2A,B). One month after virus injection, we found a higher cumulative seizure score in DG nNOS knockdown mice during pilocarpine-induced SE (275 mg/kg, s.c.) (Fig. EV2C) and more SPKs and SRS 2 months later (Fig. EV2D).

These data indicate that lack of nNOS in the DG increases susceptibility of mice to pilocarpine and aggravates epilepsy.

## Conditional deletion of nNOS in the dentate hilar interneurons causes epileptogenesis

Considering the presence of nNOS both in the postsynaptic region of glutamatergic DGCs in the molecular layer of the DG and in the cell body of GABAergic interneurons in the hilus of the DG, we then identified how distinct subtypes of neurons in the DG were affected by single-nucleus RNA-seq of the DG from mice with SE 2 months before (Fig. 3A). Excitatory granule cells and GABAergic inhibitory neurons were sorted by three markers, respectively, and analyzed (Fig. 3B). Decreased levels of nNOS transcription were observed in GABAergic inhibitory neurons but not in glutamatergic excitatory neurons (Fig. 3C).

To further distinguish the role of nNOS in glutamatergic DGCs and GABAergic interneurons in the development of epilepsy, we constructed *Nos1* conditional knockout (nNOS cKO, *Nos1*$^{loxp/loxp}$)

mice, in which the exons 4–7 of *Nos1* were flanked by two LoxP sites using the CRISPR/Cas9 system (Figs. 3D and EV3A). Upon Cre expression, the exons 4-7 were deleted in certain types of neurons in *Nos1*$^{loxp/loxp}$ mice (Fig. 3E,F). Cre expressing adeno associated virus (AAV) driven by promoter CaMKIIα (AAV9-pCaMKIIα-Cre) (Fig. 3E) or GAD67 (AAV9-pGAD67-GFP-2A-Cre) (Fig. 3F) was microinjected into both DGs of WT or *Nos1*$^{loxp/loxp}$ mice.

The AAV9-pCaMKIIα-Cre virus mainly transduced DGCs in the granular layer, while AAV9-pGAD67-GFP-2A-Cre virus mainly transduced GABAergic interneurons in the hilus (Fig. EV3B). The AAV9-pCaMKIIα-Cre virus transduced GFP$^+$ cells in the hilus were rare in both WT and *Nos1*$^{loxp/loxp}$ mice with no significance (Fig. EV3C). The AAV9-pCaMKIIα-Cre virus transduction had no influence on the number of nNOS$^+$ cells in the hilus (Fig. EV3D), but significantly reduced nNOS content in the granular layer and molecular layer other than in the hilus (Fig. EV3E). In contrast, the AAV9-pGAD67-GFP-2A-Cre virus transduction completely blocked the nNOS expression in transduced cells (Fig. EV3F), and led to knockdown of nNOS in the hilus (Fig. EV3G).

SE was induced by pilocarpine (275 mg/kg, s.c.) at 1 month after AAV9-pCaMKIIα-Cre virus injection, and it was found that selective knockout of nNOS in glutamatergic neurons did not alter the cumulative seizure score during SE induction (Fig. 3G). Surprisingly, when injected with AAV9-pGAD67-GFP-2A-Cre virus, we found that selective knockout of nNOS in GABAergic interneurons in the hilus significantly increased the cumulative seizure score during SE induction (Fig. 3H). These experiment results suggested a role of nNOS of hilar GABAergic interneurons in epilepsy development.

Next, WT and *Nos1*$^{loxp/loxp}$ mice were examined by EEG at 2.5 months after AAV9-pCaMKIIα-Cre virus transduction in the DGCs without SE induction, and EEG data showed normal type brain waves without SPKs and SRS (Fig. 3I). Strikingly and surprisingly, at 2.5 months after AAV9-pGAD67-GFP-2A-Cre virus transduction in the hilus of *Nos1*$^{loxp/loxp}$ mice without SE induction, frequent SPKs and SRS were measured by EEG (Fig. 3J). The results of this study discovered that selective deletion of hilar nNOS protein was sufficient and essential for epileptogenesis in the temporal lobe. However, general knockout of *Nos1* only led to increased sensitivity to SE induction, possibly attributed to compensatory mechanisms during development or the effects of nonselective loss of nNOS in excitatory neurons.

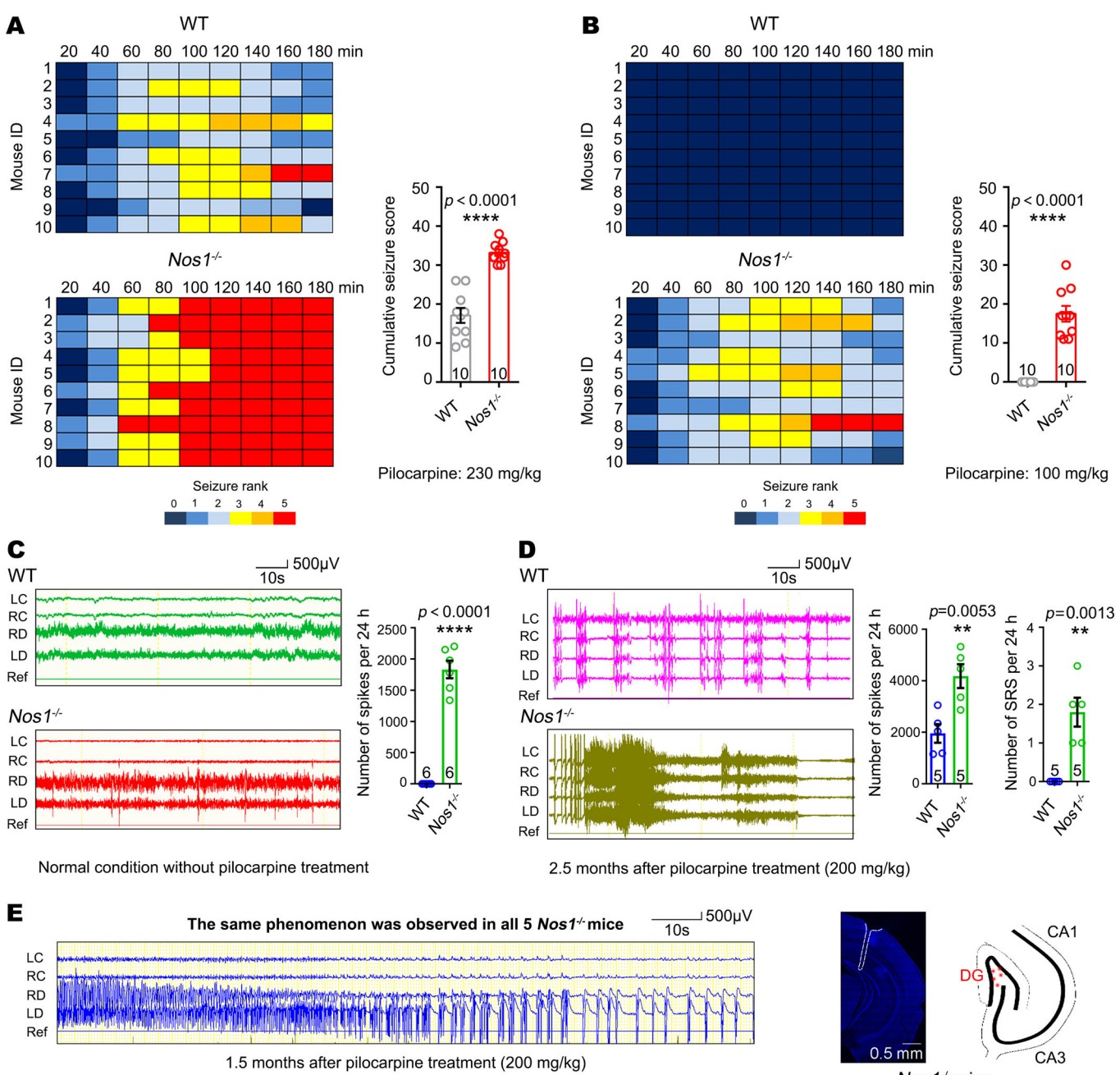

**Figure 2. Increased susceptibility to pilocarpine-induced status epilepticus and epileptogenesis in Nos1⁻/⁻ mice.**

(A) Heat map and data graph showing cumulative seizure score for 3 h after pilocarpine treatment at a dose of 230 mg/kg in *Nos1⁻/⁻* and WT mice. Student's *t*-test, *n* = 10 mice. (B) Heat map and data graph showing cumulative seizure score during 3 h after pilocarpine treatment at a dose of 100 mg/kg in *Nos1⁻/⁻* and WT mice. Student's *t*-test, *n* = 10 mice. (C) Representative pictures of EEG recording and data graph showing normal brain waves in the brain of WT mice and epileptic spikes in the brain of *Nos1⁻/⁻* mice. Student's *t*-test, *n* = 6 mice. (D) Representative images of EEG recordings and data graph showing epileptic spikes in the brain of WT mice without SRS, and epileptic spikes as well as SRS in the brain of *Nos1⁻/⁻* mice 2.5 months after pilocarpine administration at a dose of 200 mg/kg. Student's *t*-test, *n* = 5 mice. (E) Representative images of EEG recordings showing local SRS in DG of the hippocampus of *Nos1⁻/⁻* mice 1.5 months after pilocarpine treatment at a dose of 200 mg/kg. Notably, normal EEG waves were detected in the cortex at this time. The same phenomenon was observed in 5 mice. The photo of the electrode trace and the illustration showed that all the electrodes targeted DG from all 5 mice. LC left cortex, RC right cortex, RD right DG, LD left DG, Ref Reference. Error bars correspond to ± s.e.m. **P < 0.01, ***P < 0.001, ****P < 0.0001. Source data are available online for this figure.

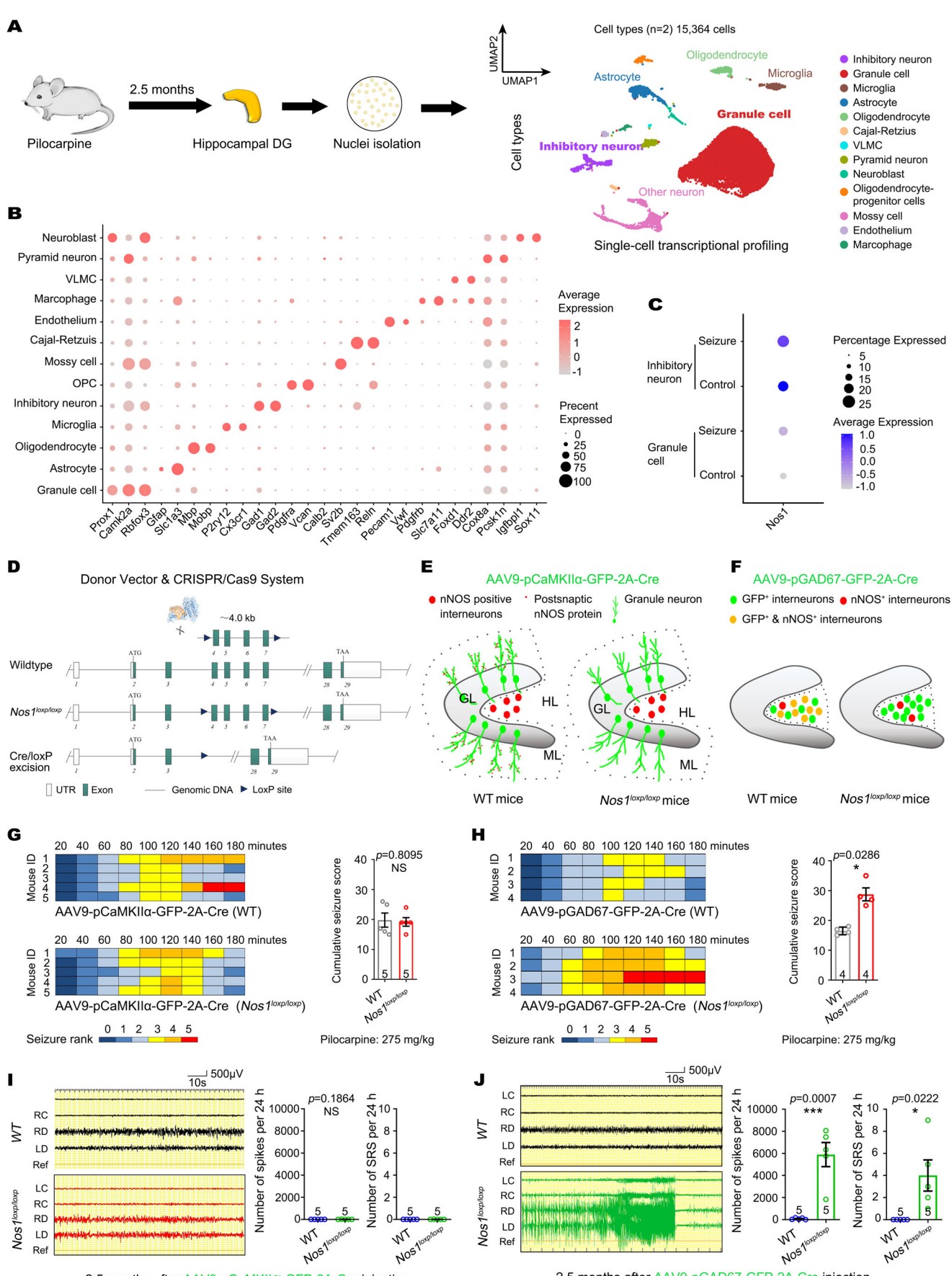

**Figure 3. Selective deletion of nNOS from hilar interneurons is sufficient to cause epileptogenesis.**

(A) The procedures of single-nucleus RNA sequencing and the sorted subclusters of principle cells in DG. (B) The dot plot demonstrated the expression of the discriminative marker genes in the 13 cell populations. (C) nNOS expression in hilar inhibitory interneurons and DGCs of mice with pilocarpine-induced SE in comparison with control. (D) The construction strategy of *Nos1^loxp/loxp* mice. The exons 4–7 were flanked by loxP site using CRISPR-Cas9 technique and were deleted upon the Cre expression. (E) Illustration of the impact of AAV9-pCaMKIIα-GFP-2A-Cre transduction on nNOS expression 2.5 months after injection into the DG of WT or *Nos1^loxp/loxp* mice. (F) Illustration of the effect of AAV9-pGAD67-GFP-2A-Cre transduction on nNOS expression 2.5 months after injection into the DG of WT or *Nos1^loxp/loxp* mice. (G) Heat map and data graph showing cumulative seizure score for 3 h after pilocarpine treatment at a dose of 275 mg/kg in WT and *Nos1^loxp/loxp* mice. The pilocarpine administration was performed 1 month after injection 1 μL of AAV9-pCaMKIIα-GFP-2A-Cre into the hilus. No significant difference was observed. Student's *t*-test, n = 5 mice. (H) Heat map and data graph showing cumulative seizure score for 3 h after pilocarpine treatment at a dose of 275 mg/kg in WT and *Nos1^loxp/loxp* mice. The pilocarpine administration was performed 1 month after injection 1 μL of AAV9-pGAD67-GFP-2A-Cre into the hilus. Student's *t*-test, n = 4 mice. (I) None epileptic spikes were observed 2.5 months after injection of AAV9-pCaMKIIα-GFP-2A-Cre. Students' *t*-test, n = 5 mice. (J) Epileptic SPKs and SRS were detected 2.5 months after injection of AAV9-pGAD67-GFP-2A-Cre. Students' *t*-test, n = 4–5 mice. GL granule layer, ML molecular layer, HL hilus, LC left cortex, RC right cortex, RD right DG, LD left DG, Ref Reference. Error bars correspond to ± s.e.m. *P < 0.05, **P < 0.01, NS, not significant. Source data are available online for this figure.

The recording and analysis of the onset of SRS showed that epileptogenesis originated in the hippocampal DG (Appendix Fig. S1A,B) other than in the cortex (Appendix Fig. S1C,D) of *Nos1^loxp/loxp* mice, suggesting that nNOS deficiency in GABAergic interneurons in the hilus was sufficient to cause epileptogenesis.

Altogether, the data here suggest a crucial role of nNOS deficiency in GABAergic interneurons in the hippocampal hilus region in the development of TLE.

## nNOS deficiency induces epilepsy-like hyperexcitatory afferent circuit integration of DGCs

To determine whether the observed epileptogenesis was related to excessive afferent inputs onto hippocampal DGCs in the case of nNOS deficiency, we employed a proven retrograde tracing system mediated by rabies virus (RbV) (Wickersham et al, 2007). Based on the intrinsic property of RbV to transport exclusively and retrogradely in synaptically connected neurons, a pseudotyped RbV vector (RbV-Enva-ΔRgp-MCh) was constructed. In the RbV-Enva-ΔRgp-MCh vector, the gene encoding RbV glycoprotein (Rgp), essential for transsynaptic propagation, was replaced with a mCherry (MCh) reporter gene to disrupt the jump. Meanwhile, the gene encoding avian virus envelope protein (EnvA) was inserted to recognize and transduce only cells with TVA receptor, an avian cell-specific receptor (Etessami et al, 2000). Besides, a LV expressing a histone-tagged and nucleus-localized green fluorescent protein (hGFP), TVA receptor, and Rgp under the control of neuronal-specific synapsin promoter (LV-Syn-GTRgp) was also constructed. The LV-Syn-GTRgp mainly transduced granular cells, and the RbV-Enva-ΔRgp-MCh tracked input cells to these DGCs (Fig. 4A,B). The whole brain-wide mapping analysis revealed that hippocampal DGCs received afferent inputs from five major brain structures, including the entorhinal cortex, forebrain, brainstem, and hippocampus (Fig. EV4A,B and Fig. 4C). nNOS deficiency caused excessive connectivity with neurons mostly located in the entorhinal cortex including lateral and medial entorhinal area (LEA and MEA), forebrain including medial septum (MS) and diagonal band (DB), brainstem including ventral tegmental area (VTA) and dorsal raphe (DR), and CA3. Meanwhile, inputs from the hilus remained unchanged (Fig. 4C).

To explore the effect of specific knockout of nNOS in hilar inhibitory neurons on the circuit rewiring of DGCs, we microinjected AAV9-pGAD67-2A-Cre into the hilus of WT and *Nos1^loxp/loxp* mice, followed by microinjection of LV-Syn-GTRgp 2 weeks later and RbV-Enva-ΔRgp-MCh microinjection 2.5 months later (Fig. 4D). The whole brain-wide mapping analysis confirmed that specific deficiency of nNOS in hilar inhibitory neurons resulted in significantly more afferent inputs from the entorhinal cortex, forebrain, brainstem and CA3, but not the hilus, to DGCs (Fig. 4E–G).

Consistently, by applying the same tracing method and strategy, we found that forced deprivation of NO by chronic administration of c-PTIO (1 mg/kg/d, i.p., 2.5 months after LV-Syn-GTRgp injection to the right DG) without SE induction (Appendix Fig. S2A) also caused a similar pattern of afferent circuit formation (Appendix Fig. S2B,C). Consequently, the production of NO in the DG was significantly reduced after c-PTIO treatment (Appendix Fig. S2D).

Taken together, these data suggest that nNOS-derived deficiency in hilar nNOS+ inhibitory neurons accounts for hyperexcitatory afferent circuit development of DGCs.

## nNOS deficiency leads to hyperactivity of DGCs

The susceptibility to pilocarpine-induced SE in *Nos1^−/−* mice was likely caused by over-activated excitatory neurons through epilepsy-like hyperexcitatory afferent circuits in the hippocampal DG. To evaluate the activity of DGCs in *Nos1^−/−* mice, we measured the number of neurons expressing cFOS, usually used as an indicator of neuronal activity, in the hippocampus of both WT and *Nos1^−/−* mice. Interestingly, cFOS immunoreactivity detection showed that nNOS deletion significantly increased the number of cFOS-positive (cFOS+) cells in the DG layer but not in the hilus (Fig. 5A). In addition, the number of cFOS+ cells in the pyramidal cell layer of the CA3 and CA1 regions remained unaltered (Fig. 5B). To further explore the excitatory presynaptic transmission into DGCs, we adopted whole-cell patch-clamp recording to measure miniature excitatory postsynaptic currents (mEPSCs) and paired-pulse ratio (PPR) in adult hippocampal slices prepared from WT and *Nos1^−/−* mice. The results showed a significant increase in mEPSCs frequency, but not in mEPSCs amplitude (Fig. 5C), as well as a decrease in EPSC PPR measured at 50-ms intervals (Fig. 5D), reflecting increased excitatory synapses and enhanced excitatory presynaptic release to DGCs in *Nos1^−/−* mice. However, the frequency and amplitude of miniature inhibitory postsynaptic currents (mIPSCs) were not affected by knockout of *Nos1* (Fig. 5E). Consistently, IPSC PPR was not changed (Fig. 5F), together

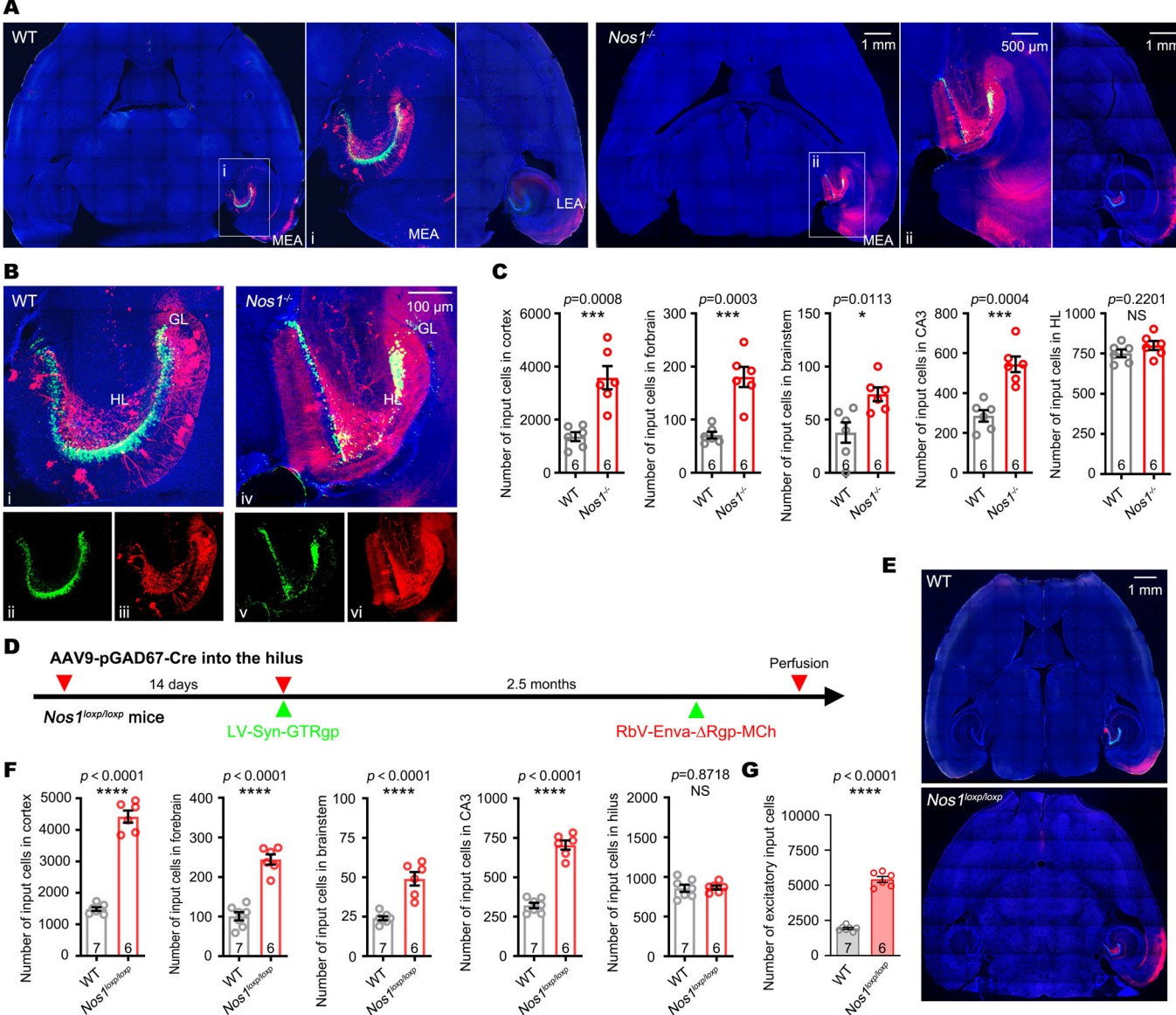

**Figure 4. Deficiency of nNOS causes excessive epilepsy-like excitatory inputs onto DGCs.**

(A) Representative photos showing the distribution of traced cells projecting to DGCs in the brain of *Nos1*⁻/⁻ and WT mice. (i) and (ii) were zoomed photos from white squares, respectively. (B) Representative photos of 'starter cells' (yellow color) transduced both with LV-Syn-GTRgp and RbV-Enva-ΔRgp-MCh in the DGCs enlarged from (A). (ii) and (iii) were split channels from (i). (v) and (vi) were split channels from (iv). (C) Data graphs showed connectivity ratios of input neurons located in the cortex, forebrain, brainstem, CA3, and hilus. Students' *t*-test, *n* = 6 mice. (D) Experimental paradigm for tracing the inputs onto DGCs after selective deletion of *Nos1* in the interneurons in the hilus. (E) Representative photos of 'traced cells' 2.5 months after injection of AAV9-pGAD67-2A-Cre into the DG of WT or *Nos1*ˡᵒˣᵖ/ˡᵒˣᵖ mice. (F) Data graphs showing the number of input neurons located in the cortex, forebrain, brainstem, CA3, and hilus of mice 2.5 months after selective knockout nNOS in hilar inhibitory neurons. Students' *t*-test, *n* = 6–7 mice. (G) Data graphs showing the brain-wide excitatory inputs onto DGCs after selective knockout *Nos1* in the hilar inhibitory neurons. Student's *t*-test, *n* = 6–7 mice. Error bars correspond to ± s.e.m. *P < 0.05, ***P < 0.001, ****P < 0.0001, NS, not significant. Source data are available online for this figure.

suggesting that inhibitory synapse formation to DGCs was not involved in hyperexcitation of DGCs in *Nos1*⁻/⁻ mice. More importantly, we found that nNOS deficiency had no significant effects on the intrinsic membrane properties of DGCs, including input resistance, membrane potential, minimal current and spike numbers (Fig. 5G). It verified indirectly that nNOS deficiency induced hyperexcitation of DGCs through hyperexcitatory afferent inputs to DGCs other than intrinsic changes in DGCs. Similarly, a remarkable elevation was observed in the number of cFOS⁺ DGCs

at 2.5 months after microinjection of AAV9-pGAD67-Cre into the hilus of *Nos1*ˡᵒˣᵖ/ˡᵒˣᵖ mice, compared with WT mice (Fig. 5H).

Taken together, these data indicate that nNOS deficiency accounts for hyperexcitation of DGCs. Therefore, long-term nNOS/NO deficiency-induced abnormal excitatory afferent circuit development may increase the hyperactivity of DGCs, giving rise to the hyperexcitation of the DG and eventually becoming a primary factor of epileptogenesis in the pathological development of TLE.

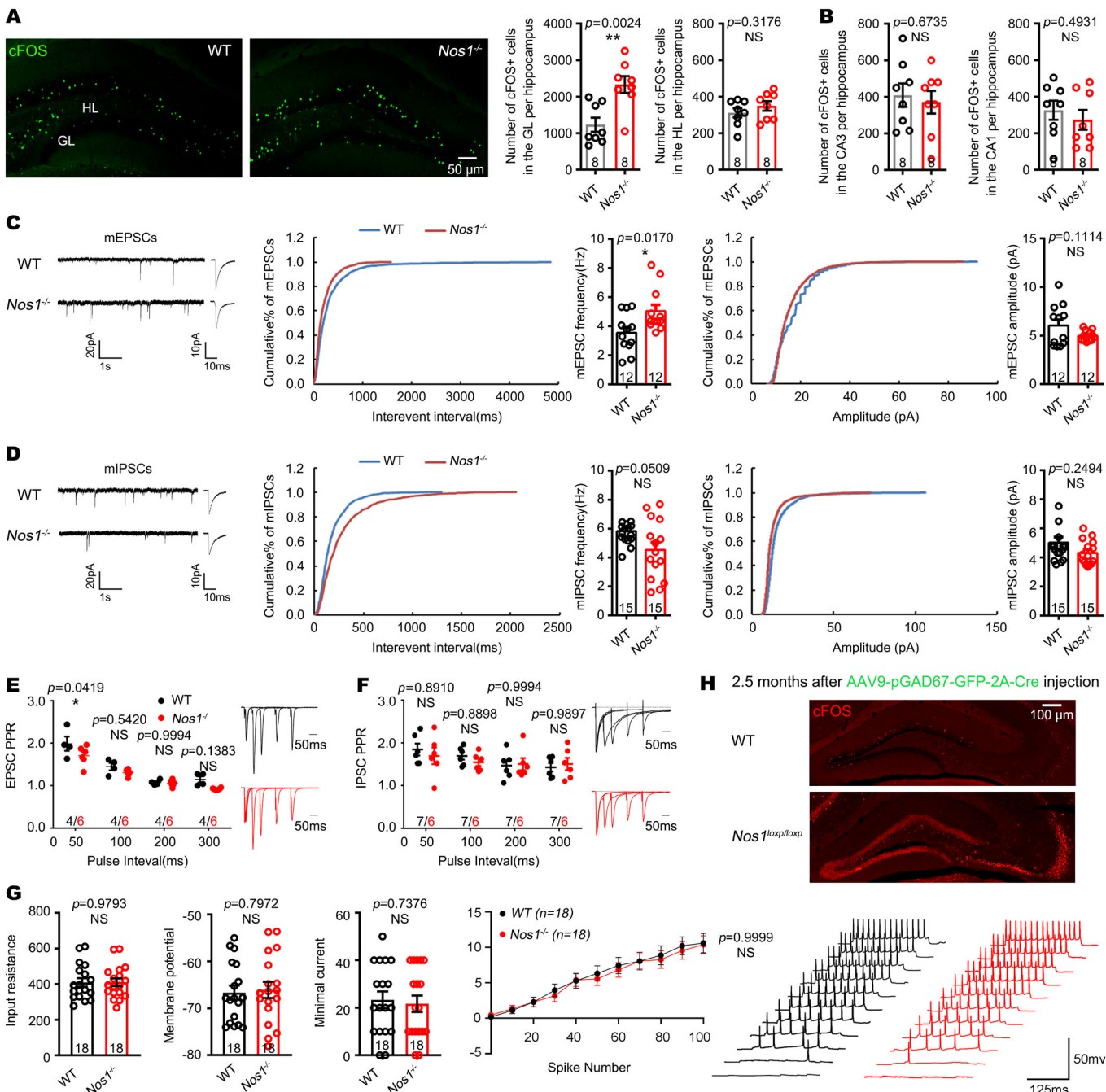

**Figure 5. Deficiency of nNOS causes hyperexcitability of DGCs.**

(A, B) Representative photos showing cFOS+ cells in the DG of WT and Nos1−/− mice. Students' t-test. Data graph showing cFOS+ cells in GL, HL, CA3, and CA1 of WT and Nos1−/− mice. Students' t-test, n = 8 mice. (C) The mEPSCs of DGCs were measured in hippocampal slices of WT and Nos1−/− mice. Students' t-test, n = 12 cells from 3 mice. (D) EPSC PPR was measured in hippocampal slices of WT and Nos1−/− mice. Students' t-test, n = 15 cells from 3 mice. (E) The mIPSCs of DGCs were measured in hippocampal slices of WT and Nos1−/− mice. Two-way ANOVA, n = 4–6 cells from 3 mice. (F) IPSC PPR was measured in hippocampal slices of WT and Nos1−/− mice. Two-way ANOVA, n = 6–7 cells from 3 mice. (G) Input resistance, membrane potential, and minimal current were measured in hippocampal slices of WT and Nos1−/− mice. Students' t-test, n = 18 cells from 3 mice. Spike numbers were analyzed by Two-way ANOVA, n = 18 cells from 3 mice. (H) Representative photos and data graph of cFOS+ cells in the DG 2.5 months after injection of AAV9-pGAD67-2A-Cre into the DG of WT or Nos1loxp/loxp mice. GL granule layer, HL hilus. Error bars correspond to ± s.e.m. *P < 0.05, **P < 0.01, NS, not significant. Source data are available online for this figure.

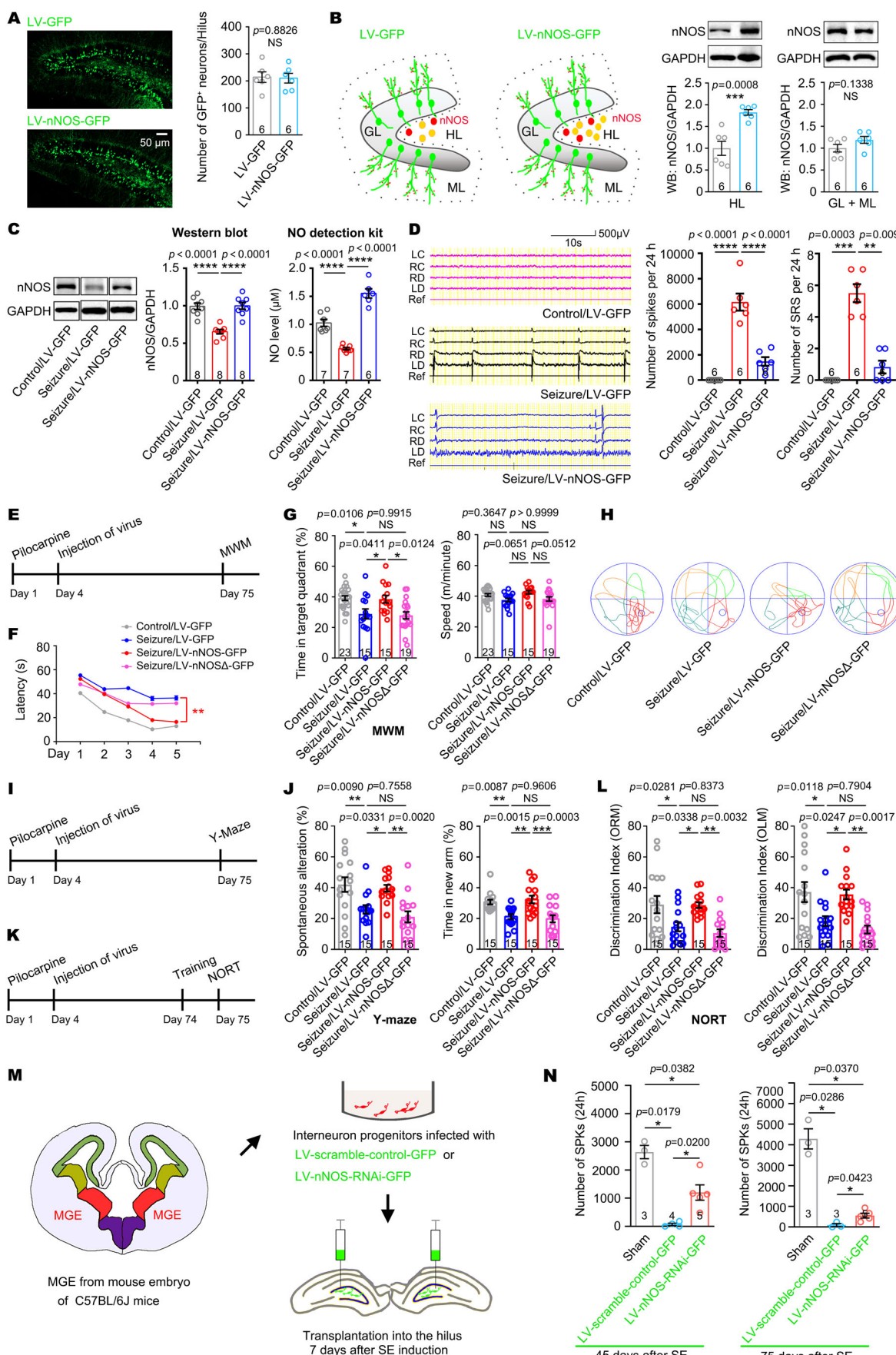

**Figure 6. Sufficiency and necessary of replenishing hilar nNOS for blocking the development of epileptogenesis in epilepsy model mice.**

(A) Representative immunofluoresence images and data graph showing LV-nNOS-GFP-transduced interneurons in the hilus. Student's *t*-test, $n = 6$ mice. (B) Illustration describing the cell types of LV-nNOS-GFP-transduced cells and data graphs showing the expression of nNOS in the hilus and in the granular layer plus molecular layer regions. Student's *t*-test, $n = 6$ mice. (C) A volume of 1 µL of LV-GFP or LV-nNOS-GFP was injected into the DGs of mice 3 days after pilocarpine-induced seizure. Representative photos and data graphs showing nNOS expression levels and the concentration of NO in DG 2.5 months after virus transduction. One-way ANOVA, $n = 6$–8 mice. (D) Representative images and data graphs of EEG recordings showing brain waves in mice 2.5 months after pilocarpine-induced SE and virus transduction. One-way ANOVA, $n = 6$ mice. (E) Timeline showing the experimental design. Pilocarpine administration was applied to induce SE at day 1 and lentivirus was injected into the DGs 3 days later. The Morris water maze test was performed at day 75. (F) Escape latency during 5-day training in MWM test. Seizure experience caused prolonged latency which was reversed by nNOS but not nNOSΔ re-expression in the DG. One-way ANOVA, $n = 13$–22 mice. (G) Time spent (%) in target quadrant during probe test at day 6 of MWM test. Pilocarpine-induced SE reduced searching time in the target quadrant which was reversed by nNOS but not nNOSΔ re-expression in DG. The moving speed of mice was unchanged. One-way ANOVA, $n = 15$–23 mice. (H) Example locomotion tracking plots showing the total path length during the probe test at day 6 of the MWM test. (I) Timeline showing the experimental design of Y-maze. (J) The tests analyzing spontaneous alternation and exploring time in a new arm. One-way ANOVA, $n = 15$ mice. (K) Timeline showing the experimental design of novel object recognition test. (L) Quantitation of discrimination index in novel object recognition memory (ORM) and object location memory (OLM) test. One-way ANOVA, $n = 15$ mice. (M) Design of transplantation of MGE-derived interneuron progenitor cells transduced with LV- scramble-control-GFP or LV-nNOS-RNAi-GFP into the hilus of mice with pilocarpine administration 1 week before. (N) Data graph of EEG recordings. One-way ANOVA, $n = 3$–5 mice. GL granule layer, ML molecular layer, HL hilus. Error bars correspond to ± s.e.m. *$P < 0.05$, **$P < 0.01$, ***$P < 0.001$, ****$P < 0.0001$, *NS*, not significant. Source data are available online for this figure.

## Replenishment of hilar nNOS blocks the development of epileptogenesis in pilocarpine-induced TLE models

To replenish nNOS in the DG, we constructed a LV encoding the full-length cDNA of nNOS (LV-nNOS-GFP) and injected into both DGs of each mouse (1 µL each) at 3 days after SE induction by pilocarpine (Fig. 6A,B). After 2.5 months, LV-nNOS-GFP successfully reversed the pilocarpine-induced reductions in nNOS protein content and NO concentration in the DG (Fig. 6C). Importantly, EEG recordings showed that replenishment of nNOS in the DG by LV-nNOS-GFP blocked the development of chronic epilepsy in pilocarpine-treated mice (Fig. 6D). In addition, LV-nNOS-GFP transduction significantly restored cognitive and memory functions following epileptic brain insult by pilocarpine, as shown in Morris Water Maze (Fig. 6E–H), Y-maze (Fig. 6I,J) and novel object recognition tests (Fig. 6K,L). To know whether the function of nNOS depends on its catalytic activity, we constructed a type of LV encoding full-length cDNA of nNOS with a nucleotide mutation which is critical for the catalytic activity of nNOS (LV-nNOSΔ-GFP), and it was found that replenishment of nNOS without catalytic activity failed to block epileptogenesis development and restore memory despair after pilocarpine administration (Fig. 6E–L), suggesting an important function of NO produced by nNOS in the DG in the development of epileptogenesis and memory deficit in epileptic mice.

Previous studies have reported that transplantation of MGE-derived GABAergic precursor cells (MPCs) in the hilus blocked the development of epileptogenesis after pilocarpine-induced injury (Hunt et al, 2013). To understand the role of nNOS in this process, we cultured MGE cells, in which nNOS was knocked down by transducing LV-nNOS-RNAi-GFP, or LV-scramble-control-GFP as negative control. Next, $3 \times 10^4$ cells transduced with LV-scramble-control-GFP or LV-nNOS-RNAi-GFP were injected into both DGs of each mouse at 7 d post SE induction (Fig. 6M; Appendix Fig. S3A,B), followed by observation of GFP⁺ MPCs in the hilus at 68 d after transplantation (Appendix Fig. S3C). As measured by EEG, transplantation of MPCs transduced with LV-scramble-control-GFP successfully blocked the progression of epilepsy at both 38 and 68 d after transplantation, compared with sham-operated mice (Fig. 6N). However, the rescue effect of MPCs transplantation was significantly attenuated after knocking down nNOS in grafted cells (Fig. 6N;

Appendix Fig. S3D), suggesting a major contributing function of hilar nNOS in the antiepileptic effect of GABA progenitor cells transplantation.

Taken together, these data suggest an important role of nNOS deficiency in the DG region in the development of epilepsy.

## NO donors could serve as a novel antiepileptogenic strategy by blocking pathological development in TLE models

The next question was that if NO is supplemented by an exogenous NO donor, whether the abnormal excitatory afferent circuits can be rescued. To answer it, we treated mice with DETA/NONOate (1 mg/kg/d, i.p.), a type of NO donor with long half-life period, for 2 months after LV-Syn-GTRgp injection to the right DG (Fig. 7A). Based on the two-virus tracing strategy, it was found that persistent replenishment of NO by exogenous DETA/NONOate during epileptogenesis of TLE blocked the formation of excessive excitatory afferent circuits of DGCs majorly from the entorhinal area (Fig. 7B,C).

To examine whether replenishment of NO can block the development of epileptogenesis in TLE by blocking the hyperexcitatory afferent circuit development of DGCs, SE were induced in mice by systemic application of pilocarpine (275 mg/kg, s.c.), and DETA/NONOate (1 mg/kg/d, i.p.) was then administered for 2 months, at 8–68 d post SE induction (Fig. 7D). Excitingly, long-term treatment with DETA/NONOate effectively blocked the development of epileptogenesis in the model of pilocarpine-induced TLE (Fig. 7E,F) due to the level of NO in the DG (Fig. 7G), however, DETA/NONOate itself did not induce abnormalities in brain waves in control mice. Furthermore, 2 months of continuous exposure to DETA/NONOate had no obvious systemic toxicity, especially in the liver [indicated by the levels of alanine aminotransferase (ALT) and total bilirubin (TBIL)], or kidneys [indicated by the levels of creatinine (Cr) and lactate dehydrogenase (LDH)], or heart [indicated by the level of cardiac troponin (CTn)] based on examinations using ELISA kits (Fig. EV5A). Besides, mice were treated with DETA/NONOate at different periods post SE, and the results uncovered that replenishment of NO during the early phase of epileptogenesis (the first month after pilocarpine administration) attenuated the development of SPKs (Fig. EV5B–D), and replenishment of

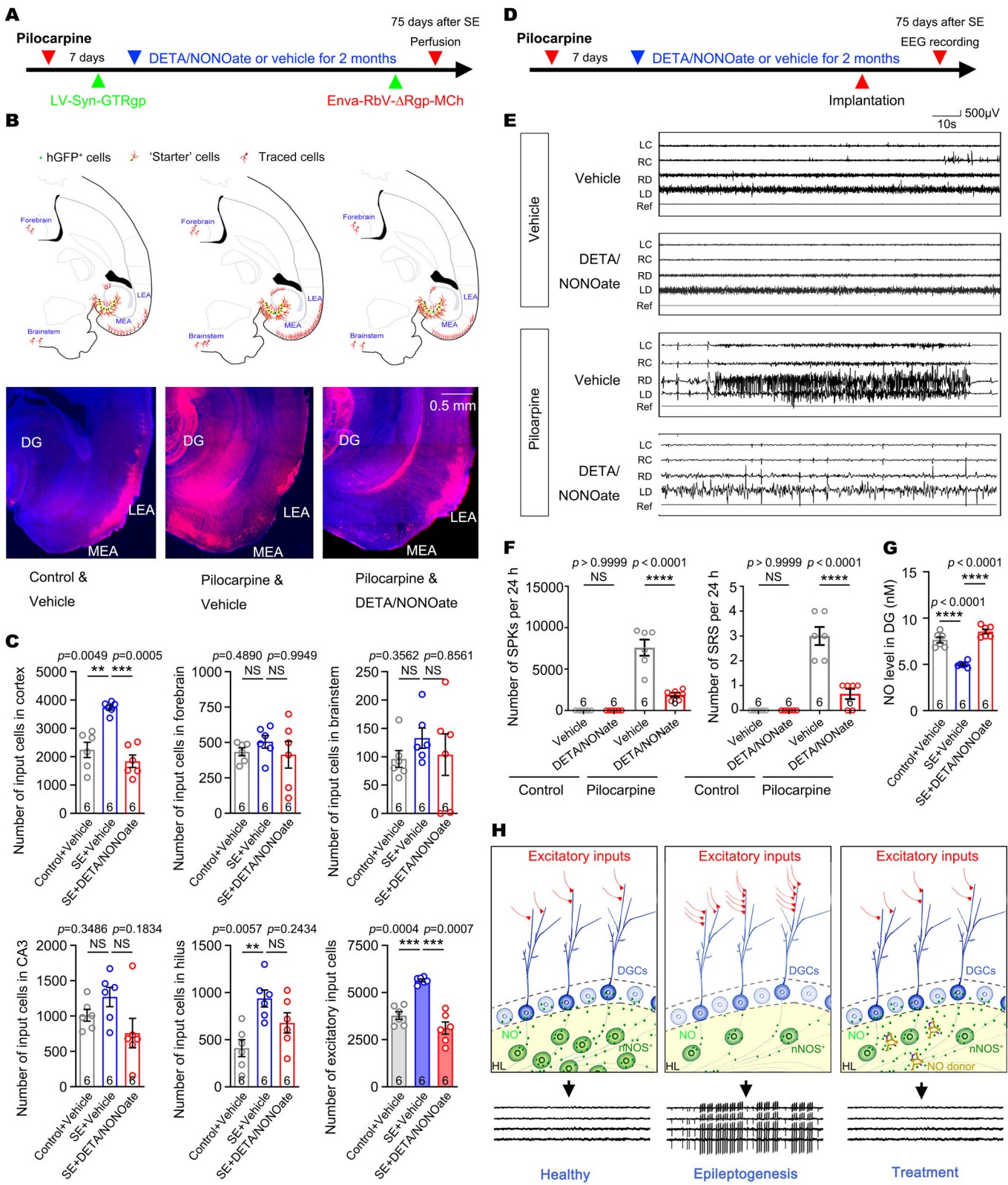

NO during the late phase of epileptogenesis (the second month after pilocarpine administration) was sufficient to block epileptogenesis, and the effect of which lasted at least for 1 month post treatment (Fig. EV5E–G).

Thereafter, whether DETA/NONOate exerts acute antiseizure effects in mice that have already developed spontaneous chronic seizures was investigated. However, EEG data showed that the single-dose administration of DETA/NONOate (1 mg/kg, i.p.)

◄ **Figure 7. Chronic treatment with NO Donor prevents epileptogenesis.**

(A–C) Experimental paradigm (A), representative images of input cells (B), and data graph (C), showing chronic treatment with DETA/NONOate (1 mg/kg, i.p., 1 time per day) for 2 months post pilocarpine-induced SE prevented the formation of excessive excitatory afferent circuit. One-way ANOVA, $n = 6$ mice. (D–F) Experimental paradigm (D), representative EEG recordings (E), and data graph (F), showing chronic treatment with DETA/NONOate (1 mg/kg, i.p., 1 time per day) for 2 months post pilocarpine-induced SE prevented the development of SPKs and SRS. Two-way ANOVA, $n = 6$ mice. (G) Data graphs showing the levels of NO in DG in SE mice with or without long-term treatment with DETA/NONOate. One-way ANOVA, $n = 6$ mice. (H) A model for mechanisms of TLE: NO deficiency due to hilar nNOS loss causes aberrant excitatory circuit of DGCs, contributing to epileptogenic development, and that NO donor treatment could be a novel strategy and a promising approach to cure TLE. LC left cortex, RC right cortex, RD right DG, LD left DG, Ref Reference. Error bars correspond to ± s.e.m. **$P < 0.01$, ***$P < 0.001$, ****$P < 0.0001$, NS, not significant. Source data are available online for this figure.

exerted no acute antiseizure effects at 5 min, 1 h, and 24 h after administration (Appendix Fig. S4A–C), whereas diazepam halted seizures almost immediately (Appendix Fig. S4D).

These data signify that DETA/NONOate prevents the epileptogenic development of TLE and NO donors can be developed as a novel ASM, different from currently available ASMs in the clinic.

Taken together, this study highlights a critical role of hilar nNOS deficiency in the epilepsy pathologies, contributing to aberrant excitatory circuits of DGCs during epileptogenic development of TLE, and NO donor therapy can serve as a novel strategy and promising approach to cure TLE (Fig. 7H).

## Discussion

In the hippocampal hilus, nNOS-containing interneurons are a unique subtype of GABAergic inhibitory neurons producing NO, a gas transmitter (Calabrese et al, 2007). The results of this study revealed that selective deletion of hilar nNOS protein was sufficient and essential for epileptogenesis in TLE. Persisted deficiency of nNOS was observed in the hippocampus of human patients suffering from TLE and in the hilus of mouse models of epilepsy, the role of which in epileptogenesis was demonstrated as leading to excessive epilepsy-like excitatory input circuit formation and thereby hyperexcitation of DGCs. Besides, we further confirmed a critical role of chronically lacking NO in this process and found that chronic treatment with DETA/NONOate, a NO donor with a long half-life, exerted a substantial therapeutic effect on the pathological development of epileptogenesis in TLE.

The sparse activity of DGCs is thought to be critical for cognitive and memory functions mediated by the hippocampus, whereas excessive DGCs activity induced hyperexcitation of the DG contributes to epileptogenic processes (Botterill et al, 2019). Unlike physiological conditions with low intrinsic excitatory DGCs that are strongly inhibited by hilar interneurons (Acsady et al, 2000), TLE is characterized by aberrantly integrated hippocampal DGCs associated with their hyperactivity in both animal models and clinical human patients (Hester and Danzer, 2014). Hyperexcitability of granule cells with prolonged mEPSPs and more action potentials (APs) was detected in hippocampal slices from epileptic rats, and mIPSP conductance in epileptic slices was less evoked than control (Kobayashi and Buckmaster, 2003). Although the previous study conducted by our research team showed that aberrant circuit integration of DGCs plays a causal role in epileptogenesis in the temporal lobe (Zhou et al, 2019), and circuit-based interventions targeting DGCs can rescue TLE-associated dysfunction, the molecular mechanism remains unrevealed (Goldberg and Coulter, 2013; Kahn et al, 2019). The results

of the present study revealed that there was an epilepsy-like pattern of neural circuits of DGCs in nNOS-deficient mice. It was clarified that the physiological level of nNOS-derived NO in the hilus maintained the natural circuit development of DGCs. NO is a free radical and penetrates the cell membrane easily. The observations in this study revealed that persistent deficiency in the nNOS-NO pathway contributed to aberrant circuit integration and hyperexcitation of DGCs, as well as eventual epileptogenesis, uncovering an important etiology of epileptogenesis in TLE. However, the molecular signaling pathway by which NO derived from hilar nNOS-expressing cells modulates the inputs to DGCs remains to be clarified in future work.

Transplantation of GABAergic precursor cells in the hippocampus and subsequent integration of inhibitory neurons into neural circuits have been demonstrated as an effective therapeutic strategy for the treatment of epilepsy (Cossart et al, 2005; Hunt et al, 2013). However, it remains unclear whether specific subpopulations of inhibitory neurons differentiated from GABAergic precursor cells contribute to halting seizures and rescuing behavioral deficits in epileptic mice. After grafting from donor mice, ~25% of GABAergic precursor cell-derived interneurons are nNOS$^+$ in the hilus. It was discovered in this study that purely re-expression of nNOS in the hilus of epileptic mice was sufficient to prevent spontaneous seizures and rescue memory formation deficits. More importantly, data in the present study showed that transplantation of GABAergic interneurons without nNOS subtype cells into the hilus did not significantly attenuate antiepileptic effects, especially in the early phase, rendering new insights for re-evaluating the importance of GABA in the recovery from epilepsy. Data in the present study suggested a critical role of nNOS protein per se in the progression of epilepsy and an important function of nNOS$^+$ interneurons in the recovery from epilepsy after hilar GABAergic progenitor transplantation.

However, how to pharmacologically reverse epileptic pathology, rather than cell transplantation, which still has serious ethical and clinical issues, deserves attention. In the present study, it was uncovered that NO, produced via nNOS catalysis, plays a critical role in this pathology, and pharmacological NO donor treatment could be an inspiring approach, providing a strategy to resolve this long-standing problem. A series of studies have reported increased expression of nNOS and concentration of NO in the hippocampus in various mouse models of epilepsy (Kovacs et al, 2009; Yasuda et al, 2001; Zhu et al, 2017). However, all these experiments examined nNOS and NO in the entire hippocampus including DG, CA1, CA2, CA3, SUB and other subregions. The DG region consists of the molecular layer, granule layer and hilus. The nNOS$^+$ interneurons intensively locate in the hilus rather than CA1, CA2, CA3, SUB and other subregions. A reduced number of hilar

interneurons is a hallmark of the pathology of TLE. Therefore, the measurement of nNOS in the DG and the hippocampus should be different and predictable. Although a recent study found increased number of nNOS$^+$ cells in the hilus at 14 d after SE (Yao et al, 2022), which, however, returned to basal levels at 28 d, chronic changes were not examined. Data of this study showed that nNOS-NO and nNOS$^+$ interneurons in the hilus decreased to extremely low levels at 14 d and 2 months after SE induction. NO is a free radical in the CNS, produced by the conversion of L-arginine (Calabrese et al, 2007; Griffith and Stuehr, 1995). Accordingly, L-arginine, the catalytic substrate of nNOS, predictably has no therapeutic value, and direct supplement of NO may be effective. As denoted by this study, there was a specific response of hilar nNOS interneurons to SE insult that eventually led to epileptic pathology, and a molecular target was uncovered for a pharmacological neurorestorative approach beyond ion channel drugs.

There is a growing body of evidence that a physiological amount of NO is neuroprotective by modulating neurogenesis, synaptic transmission, neurosecretion, fluid balance, signal transduction, and cell death (Guix et al, 2005; Packer et al, 2003). NO donor compounds are approved for treatment of myocardial infarction and erectile dysfunction in clinic for a long term. As a NO donor with a half-life of about 20 h, DETA/NONOate is commonly applied to mimicking continuous production of NO in the CNS (Bauer et al, 1995; Hanson et al, 1995; Matsumoto et al, 2003). Although the side effects of systemic supplement of NO and the toxicity risk of overdose of DETA/NONOate are non-negligible (Pervin et al, 2001), NO donors with stable and targeted release may be a potentially valuable novel approach to intervention in epilepsy.

Up to date, most ASMs in clinical use belong to blockers of neuronal ion channels, particularly sodium channels (Brodie, 2017). However, these drugs have only therapeutic advantages in suppressing SE but not the development of epilepsy. In addition, more than 150 mutations in genes encoding sodium channels have been identified, causing drug-resistant problems (Meisler and Kearney, 2005). Therefore, progression of TLE to DRE is found in a substantial percentage of patients after long-term use of these ASMs (Zuliani et al, 2012). Our study discovered that abnormal circuit integration and hyperexcitation of DGCs were attributed to the NO deficiency in the hilus. Although the antiseizure effect was not observed, replenishment of NO prevented epileptogenic development, the action of which is completely different from that of classical sodium channel blockers. Treatment with sodium channel blockers together with NO donors could be a potentially better clinical pathway for curing TLE patients in the future.

## Methods

### Reagents and tools table

| Reagent/Resource | Reference or Source | Identifier or Catalog Number |
| --- | --- | --- |
| **Experimental Models** | | |
| C57BL/6J (*M. musculus*) | Jackson Laboratories | C57BL/6J |
| *Nos1$^{-/-}$* (*M. musculus*) | Jackson Laboratories | B6;129S4-*Nos1$^{tm1Plh}$*/J |
| *Nos1$^{loxp/loxp}$* (*M. musculus*) | GemPharmatech Biotechnology | C57BL/6JNju-*Nos1$^{em1Cflox}$*/Gpt |
| MGE cells | This Study | N/A |
| 293T (*H. sapiens*) with STR profiling and mycoplasma contamination tests | Shanghai Institute of Cell Biology, Chinese Academy of Sciences | GNHu17 |
| DRE subjects | Sir Run Run Hospital, Nanjing Medical University | N/A |
| Control subjects | Human Brain Tissue Resource Center of Nanjing Medical University | N/A |
| **Plasmids/Vectors** | | |
| LV-Syn-GTRgp | Brainvta. Ltd | N/A |
| RbV-EnvA-ΔRgp-MCh | Brainvta. Ltd | N/A |
| LV-nNOS-RNAi-GFP | Genechem Co., Ltd | N/A |
| LV-scramble-control-GFP | Genechem Co., Ltd | N/A |
| LV-GFP | This Study | N/A |
| LV-nNOS-GFP | This Study | N/A |
| LV-nNOSΔ-GFP | This Study | N/A |
| **Antibodies** | | |
| rabbit anti-nNOS | Cell Signaling Technology | CST-4231S |
| mouse anti-nitrotyrosine | Santa Cruz Biotechnology | sc-32757 |
| mouse anti-GAPDH | Kangchen Ltd | KC-5G4 |
| Goat anti-Mouse IgG (H + L) Secondary Antibody, HRP | Invitrogen | 31430 |
| Goat anti-Rabbit IgG (H + L) Secondary Antibody, HRP | Invitrogen | 31460 |
| rabbit anti-cFOS | SYSY | 226003 |
| Goat anti-Rabbit IgG (H + L) Cross-Adsorbed Secondary Antibody, Cyanine3 | Thermo Fisher Scientific | A10520 |
| Goat anti-Rabbit IgG (H + L) Cross-Adsorbed Secondary Antibody, Alexa Fluor™ 488 | Thermo Fisher Scientific | A11008 |
| **Oligonucleotides and other sequence-based reagents** | | |
| nNOS forward primer (*H. sapiens*) | 5′-ATT CGG CTG TGC TTT GAT GG-3′ | N/A |
| nNOS reverse primer (*H. sapiens*) | 5′-TGT CCA CAG CGT GTC CGA A-3′ | N/A |
| GAPDH forward primer (*H. sapiens*) | 5′-CTC CTC TGA CTT CAA CAG CGA C-3′ | N/A |
| GAPDH reverse primer (*H. sapiens*) | 5′-CTC TCT TCC TCT TGT GCT CTT GC-3′ | N/A |
| nNOS forward primer (*M. musculus*) | 5′-ACA CGC ATG TCT GGA AAG GCA C-3′ | N/A |
| nNOS reverse primer (*M. musculus*) | 5′-CTC TGT GGC ATA GAG GAT GGT-3′ | N/A |
| iNOS forward primer (*M. musculus*) | 5′-GCT CTA CAC CTC CAA TGT GAC C-3′ | N/A |

| Reagent/Resource | Reference or Source | Identifier or Catalog Number |
|---|---|---|
| iNOS reverse primer (*M. musculus*) | 5′-CTG CCG AGA TTT GAG CCT CAT G-3′ | N/A |
| eNOS forward primer (*M. musculus*) | 5′-GAA GGC GAC AAT CCT GTA TGG C-3′ | N/A |
| eNOS reverse primer (*M. musculus*) | 5′-TGT TCG AGG GAC ACC ACG TCA T-3′ | N/A |
| GAPDH forward primer (*M. musculus*) | 5′-GTC TCC TCT GAC TTC AAC AGC G-3′ | N/A |
| GAPDH reverse primer (*M. musculus*) | 5′-ACC ACC CTG TTG CTG TAG CCA A-3′ | N/A |
| **Chemicals, Enzymes and other reagents** | | |
| Kainic acid | Sigma-Aldrich | K0250 |
| Pentylenetetrazol | Sigma-Aldrich | P6500 |
| Atropine | Sigma-Aldrich | A-046 |
| Scopolamine | Sigma-Aldrich | S-099 |
| Terbutaline | Sigma-Aldrich | T2528 |
| Pilocarpine | Sigma-Aldrich | P6503 |
| c-PTIO | Sigma-Aldrich | C221 |
| DETA-NONOate | Sigma-Aldrich | D184 |
| isoflurane | Redwood Technology Ltd | R510-22-10 |
| DAPI | Sigma-Aldrich | 10236276001 |
| TRIzol | Thermo Fisher Scientific | 15596018 |
| SuperScript™ II Reverse Transcriptase | Thermo Fisher Scientific | 18064071 |
| SYBR Premix Ex Taq™ | Takara | RR420 |
| SNAP | MedChem | HY-121526 |
| NLB buffer | 250 mM Sucrose, 10 mM Tris-HCl, 3 mM MgAc$_2$, 0.1% Triton X-100, 0.1 mM EDTA, 0.2 U/μL RNase Inhibitor | N/A |
| aCSF | 110 mM NaCl, 0.5 mM CaCl$_2$, 2.5 mM KCl, 7 mM MgCl$_2$, 1.3 mM NaH$_2$PO$_4$, 1.3 mM Na-ascorbate, 0.6 mM Na-pyruvate, 25 mM NaHCO$_3$, and 20 mM glucose | N/A |
| Intracellular solution | 70 mM potassium gluconate, 70 mM KCl, 2 mM NaCl, 2 mM MgCl$_2$, 10 mM HEPES, 1 mM EGTA, 2 mM MgATP, and 0.3 mM NaGTP with pH 7.2–7.3 and osmolarity between 275–285 mOsmol | N/A |
| **Software** | | |
| NeuroWork bench software | Nihon Kohden America Inc | N/A |
| GraphPad Prism 9 | https://www.graphpad.com/ | N/A |
| **Other** | | |
| Ultra micro pump | World Precision Instruments | 160494 F10E |
| Stereotactic apparatus | David Kopf Instruments | N/A |
| Bipolar twisted stainless-steel electrodes | Plastic One | N/A |
| Stainless-steel screws | Plastic One | MX-0090-2 |
| Plastic plug | Solderless Terminal | SMB-06V-BC |
| Dental cement | Henry Schein Inc | N/A |
| qTOWER$^3$G | Analytik Jena | N/A |
| NOx kit | Jiancheng Bioengineering | N/A |
| Free Radical Analyzer | World Precision Instruments | TBR 4100 |
| ISO-NOP NO sensor | World Precision Instruments | N/A |
| Illumina Novaseq 6000 | Illumina | N/A |
| Chromium Single Cell 3′ V3.1 Reagent Kits | 10X Genomics | N/A |
| 10X Genomics Chromium Controller | 10X Genomics | N/A |
| Bioanalyzer 2200 | Agilent | N/A |
| Computer-controlled amplifier | Molecular Devices | MultiClamp 700B |
| Upright microscope | Zeiss | Examiner Z1 |
| Bipolar tungsten microelectrode | Hertfordshire | N/A |
| PowerPac | Bio-Rad | N/A |
| Mini-PROTEAN Tetra | Bio-Rad | N/A |

## Animals

Adult male C57BL/6J mice (8–10 weeks old) were housed 5 per cage under standard laboratory conditions (22 ± 1 °C, 60% humidity, 12 h light/dark cycle with lights on from 07:00 AM to 07:00 PM, food and water ad libitum) for one week before starting experiments. Male homozygous nNOS-deficient mice (B6.129S4-*Nos1*$^{tm1Plh}$/J, *Nos1*$^{−/−}$; Jackson Laboratories, Bar Harbor, ME, USA) and their wild-type littermates of similar genetic background (B6129SF2, *Nos1*$^{+/+}$; Jackson Laboratories, Bar Harbor, ME, USA) were used in this study (Zhou et al, 2007). The *Nos1*$^{loxp/loxp}$ mice were generated by our lab collaborated with Nanjing Biomedical Research Institute of Nanjing University through CRISPR/Cas 9-mediated gene editing. In brief, guide RNAs (5′-CAT TTG CAG GTA ATG TAG GG-3′ and 5′-TGC AGG CTA CGT TCT TTG TA-3′) were obtained by in vitro transcription and purification. The gRNAs were incubated with purified Cas9 protein and injected into fertilized eggs (at the one-cell stage) together with the targeting vector with two loxp sites flanking exons 4, 5, 6, and 7 of *Nos1*. The injected fertilized eggs were cultured to the two-cell stage followed by transplantation into pseudopregnant mice. The targeted genomes of F0 mice were amplified by PCR and sequenced and the chimeras were crossed with wild-type C57BL/6J mice to obtain F1 mice. Southern blot analysis was performed with the tail DNA of F1 mice to confirm the correct cre combination and exclude random insertions of the targeting vector. All procedures concerning animal care and treatment were carried out in accordance with the protocols approved by the ethical committee of Nanjing Medical University (protocol number: IACUC-1704010-1).

## Genotyping

Genotyping Primers of Nos1$^{-/-}$ mice: (1) 5′-ACG ACC TGT GGT CAG ATT CC-3′, (2) 5′-TGG GTG CAT GTG TAT GTG TG-3′, (3) 5′-GAT ACG TGT AGA GGG CAA ATG-3′. The primer stock should be diluted to make 100 picomoles/μL. Then mix 10 μL of each primer 1, 2, and 3 with 70 μL 3-distilled water to make final volume of 100 μL. This 100 μL is the working stock. 1 μL of the working stock can be used for 1 PCR reaction. PCR cycles: 32 cycles. Denaturation: 94 °C, 30 s. Annealing: 55 °C, 30 s. Extension: 72 °C, 1 min. Final extension: 72 °C, 10 min.

## Human tissue procurement and preparation

We obtained tissues from six DRE subjects from the hospital and six age-matched control subjects from the Department of Human Anatomy, Human Brain Tissue Resource Center of Nanjing Medical University (see Table EV1 for details). The informed consent was obtained from all subjects. The experiments conformed to the principles set out in the WMA Declaration of Helsinki and the Department of Health and Human Services Belmont Report, and approved by the ethical committee of Sir Run Run Hospital, Nanjing Medical University (protocol number: V1.0/ 20220226). The autopsy materials were obtained from organ donors within 24 h of death. The hippocampi were isolated from brain specimens from patients undergoing a temporal lobectomy and were divided into parts for RNA-Seq, RT-qPCR, and Western blot in Fig. 1.

## Drugs

Pilocarpine, terbutaline, scopolamine, atropine, pentylenetetrazol, kainic acid, c-PTIO [2-(4-Carboxyphenyl)-4,4,5,5-tetramethylimidazoline-1-oxyl-3-oxide, potassium salt] and DETA-NONOate [2,2'-(hydroxynitrosohydrazono)bis-(ethanamine)] were purchased from Sigma-Aldrich (St Louis, MO, USA) and were dissolved in saline. All these drugs were freshly prepared before using.

## Virus production

Lentiviral (LV-Syn-GTRgp) vectors were generated to express hGFP, TVA, and Rgp under the control of the synapsin promoter. Lipofectamine 2000 (Invitrogen) was used to produce retrovirus by transient transfection of the LV-Syn-GTRgp vector (7.5 μg), as well as the helper vectors CMV-GP (5 μg) and CMV-VSVG (2.5 μg), into 60% confluent 293 T cells grown in 10 cm plates. The supernatant containing virus was harvested 36 h later and ultracentrifuged. The final titers were estimated to be ~10$^8$ CFU/ ml, as determined by infection of 293 T cells with serially diluted virus. RbV-EnvA-ΔRgp-MCh was purchased from Brainvta. Ltd company in Wuhan, China. Briefly, the endogenous rabies virus glycoprotein was replaced with an mCherry reporter gene (Δgp-mCherry), and this rabies virus vector was pseudotyped with EnvA receptor (RbV-EnvA-ΔRgp-MCh). The titer of rabies virus was estimated to be ~10$^8$ CFU/ml, and the virus was diluted to ~5 × 10$^7$ CFU/ml for use.

LV-nNOS-RNAi-GFP particles, and their control LV-scramble-control-GFP particles were purchased (Genechem Co., Ltd, Shanghai, China). LV-nNOS-RNAi-GFP is a pool of concentrated,

transduction-ready viral particles containing 4 target-specific constructs that encode 19–25 nt (plus hairpin) shRNA designed to knock down gene expression. Each vial contains 200 μL frozen stock containing $1.0 \times 10^6$ infectious units of virus (IFU) in Dulbecco's modified Eagle's medium with 25 mM HEPES pH 7.3.

LV-nNOSΔ-GFP, LV-nNOSΔ-GFP and LV-GFP is constructed by our lab. A critical role of the neuronal nitric oxide synthase heme proximal side residue, Arg418, in catalysis and electron transfer has been demonstrated by studies. Briefly, the coding sequence of mouse nNOS was amplified by RT-PCR. The primer sequences were as follows: forward, 5′-TGT CCT ATA CAG CTT CCA GA-3′; reverse, 5′-CAC GAT GTC ATA TTC CTC CA-3′. The PCR fragments were cut by restriction enzymes and inserted into the plasmid pCMV-GFP (For LV-GFP) by T4 DNA ligase to replace the sequence of GFP in pCMV-GFP. GFP was controlled by another promotor, pUbi. The mouse nNOSΔ gene was amplified by our previously constructed LV-nNOS-GFP plasmid carrying the whole gene encoding nNOS by deletion mutant PCR. The primer sequences were as follows: forward, 5′-GAT CTA TTT CCG GTG AAT TCC GCC ACC ATG GAA GAG CAC ACG TTT GGG GTC CAG-3′; reverse, 5′-CTA GTA TCG ATG GAC TCG AGT TAG GAG CTG AAA ACC TCA TCT GTG TC-3′. PCR fragments and pGV287-GFP plasmid were digested with Age 1 and BamH 1 and ligated with T4 DNA ligase to produce LV-nNOSΔ-GFP. Using 100 μL Lipofectamine 2000, 293 T cells were co-transfected with 20 μg of pCMV-nNOS-GFP, 10 μg of VSVG, 7.5 μg of RSV-REV and 3.5 μg pMDL g/p RRE to generate the recombinant lentivirus, LV- nNOSΔ-GFP. After 48 h, supernatant was harvested from 293T cells, filtered at 0.45 μm, and pelleted by ultracentrifugation at $18,000 \times g$ for 2 h at 4 °C. After resuspension by PBS, serially diluted retrovirus was used to transduce 293T cells; 4 days later, labeled 293T cells were counted to calculate the viral titer (~2 × 10$^9$ transducing units/ml). As a control, we also generated a retroviral vector expressing GFP alone (LV-GFP).

## Stereotaxic surgery

The mice were anesthetized with isoflurane (induction 3%, 0.8 L/ min O$_2$; maintenance 1–1.5%, 0.5 L/min O$_2$), placed in a stereotaxic apparatus and kept at physiological temperature by a home-othermic blanket. Stereotaxic surgery was performed to deliver viruses or drugs into the hippocampal DG (1 μL, coordinates: 2.3 mm posterior to bregma, 1.35 mm lateral to the midline, and 2.3 mm below dura), The mice were recovered on hot pad (37 °C) until waken up and then return to home cages.

## Induction of SE by systemic pilocarpine application

At 8–10 weeks of age and 24–25 g of body weight, male mice were given subcutaneous injections of 1 mg/kg methyl scopolamine nitrate in sterile saline,1 mg/kg terbutaline in sterile saline, and 1 mg/kg atropine in sterile saline. 30 min later, mice were given subcutaneous injections of 275 mg/kg pilocarpine in saline. All pilocarpine treatments were conducted between 9:00 A.M. and 9:00 P.M. Mice were observed after the injections for the development of continuous seizure activity, defined behaviorally by ranking, such as immobility and staring (stage 1), forelimb and/ or tail extension, head bobbing, and whisker twitching (stage 2), to severe stages, corkscrew turning and splaying of limbs (stage 3),

showing continuous rearing and falling, and wild running or jumping (stage 4), and continuous tonic/colonic seizures (stage 5). 3 h after the onset of SE, mice were given one dose of 1 mg/kg diazepam at 15 min intervals. Mice were housed in a 32 °C incubator for 2 h after diazepam treatment and given 37 °C saline subcutaneously to maintain pretreatment weight and improve recovery. Control mice received all drugs and treatments, except for saline instead of pilocarpine. Animals that received pilocarpine and did not develop convulsive seizures equal or higher than stage 4 were excluded for chronic investigation. The duration of latent period is around 7–30 days. Epileptic spikes recorded by EEG usually occur from 30 days after SE. SRS recorded by EEG begin ~45 days after SE, reach a peak around 75 days after SE, and maintain relative stability for the following time. All protocols were the same when mice were administered with lower concentrations of 100, 200, or 230 mg/kg pilocarpine.

Cumulative seizure score was measured over a 3-h period after pilocarpine injection. During this period, the seizure activity of mice was ranked (stage 1–5) every 20 min and recorded the highest seizure activity score the mice showed during this period. The scores of a mouse over the 3-h period were summarized as the cumulative seizure score. The heat map does not show the cumulative seizure scores, but the rank scores every 20 min over a 3-h period. For instance, if two stage 2 seizures were observed in 20 min, the heat map still shows a 2. The observer was blinded to the group information.

## Induction of SE by intrahippocampus injection of kainic acid

The mice were anesthetized with isoflurane (induction 3%, 0.8 L/min O$_2$; maintenance 1–1.5%, 0.5 L/min O$_2$), placed in a stereotaxic apparatus and kept at physiological temperature by a homeothermic blanket. Stereotaxic surgery was performed to deliver KA (0.5 μg/μL, 0.5 μL) into the biolateral hippocampal DG (coordinates: 2.3 mm posterior to bregma, 1.35 mm lateral to the midline, and 2.3 mm below dura) using Ultra Micro Pump (160494 F10E, WPI) over a period of 5 min, The mice were recovered on hot pad (37 °C) until waken up and then return to home cages.

## Induction of SE by bentylenetetrazol

Pentylenetetrazol (PTZ, a γ-aminobutyric acid subtype A (GABAA)-receptor antagonist, [75 mg/kg (one dose), dissolved in sterile saline, volume injected: 0.25 mL, intraperitoneal (IP)] (Sigma, St. Louis, MO, USA) was administered for induction of seizures. Mice were observed continuously for at least 2 h with data recorded regarding time to achievement of the respective Racine stage and duration of seizures. Control group animals were given normal saline (NaCl 0.9% solution) only.

## EEG and analyses

As described in our previous research (Zhou et al, 2019), mice were anesthetized and placed in a stereotactic apparatus (David Kopf Instruments, USA). For hippocampal recordings, bipolar twisted stainless-steel electrodes (0.2 mm in diameter; Plastic One, USA) were placed into both hippocampi (left DG [LD]; right DG [RD]; 2.75 mm posterior to the bregma, 2.45 mm lateral to the midline,

and 2.85 mm below the dura). Stainless-steel screws (MX-0090-2; Plastic One, USA) were placed epidurally and bilaterally over the frontal cortices (left frontal [LF]; right frontal [RF], 0.5 mm posterior to the bregma, 2.45 mm lateral to the midline). An additional screw was placed just to the right of the frontal sinus and served as a reference electrode (REF). The electrodes were then connected to a plastic plug (SMB-06V-BC, Solderless Terminal, USA), which was fixed to the skull with dental cement (hygienic repair resin; Henry Schein Inc., USA). Mice were left unrestrained for at least 72 h to recover from surgery before further manipulation and prolonged EEG monitoring were initiated.

As described in our previous research (Zhou et al, 2019), digitized EEG was reviewed using NeuroWork bench software (Nihon Kohden America Inc., Irvine, USA) at a sampling rate of 1 kHz with a high-frequency filter of 70 Hz in synchronization. EEG signals were reformatted in referential and bipolar montages. EEG recordings were performed regularly after pilocarpine injection or virus transduction of 2.5 months. All EEG were visually checked for the presence of seizures. Mice were recorded continuously for 3 days to create 36 files (2 h per file), 25% of which was sampled for manual quantification and multiplied to reflect a 24 h period. The diurnal or nocturnal files were equally chosen. Analyses were performed by double-blind investigators. The SPKs recorded by EEG were defined as scattered and unrepetitive epileptic spikes. The SRS recorded by EEG were defined as continuous, repetitive and evolving spikes with interruption of the background activity for 10 s or longer. As previously reported, epileptiform/interictal spikes were defined as paroxysmal electrical sharp activity lasting 20–150 ms, with an amplitude that was at least five times the background EEG activity.

## Quantitative RT-qPCR

Total RNA was extracted from DG of the hippocampus using TRIzol reagent following reverse transcription with SuperScript™ II Reverse Transcriptase according to the manufacturer's instructions. Real-time PCR was conducted using the SYBR Premix Ex Taq™ on qTOWER³G (Analytik Jena, Germany). The primers for nNOS, iNOS, eNOS, and GAPDH were as follows: for nNOS: Forward, 5′-ACA CGC ATG TCT GGA AAG GCA C-3′, and Reverse, 5′-CTC TGT GGC ATA GAG GAT GGT C-3′; for iNOS: Forward, 5′-GCT CTA CAC CTC CAA TGT GAC C-3′, and Reverse, 5′-CTG CCG AGA TTT GAG CCT CAT G-3′; for eNOS: Forward, 5′-GAA GGC GAC AAT CCT GTA TGG C-3′, and Reverse, 5′-TGT TCG AGG GAC ACC ACG TCA T-3′; For GAPDH: Forward, 5′-GTC TCC TCT GAC TTC AAC AGC G-3′, and Reverse, 5′-ACC ACC CTG TTG CTG TAG CCA A-3′. The relative expression was measured using the 2$^{-\Delta\Delta Ct}$ method and normalized to control.

## Western blot

Western blot analysis of hippocampal DG from human subjects and mice was performed as described previously (Zhou et al, 2017). The primary antibodies were as follows: rabbit anti-nNOS (1:1000; Cell Signaling Technology, Danfoss, USA), mouse anti-nitrotytosine (1:2000; Santa Cruz Biotechnology, CA, USA), and mouse anti-GAPDH (1:3000; Kangchen Ltd, Shanghai, China). Appropriate horseradish peroxidase-linked secondary antibodies

were used for detection by enhanced chemiluminescence (Thermo Fisher Scientific, Waltham, USA).

## Immunofluorescence staining

Mice were anaesthetized with isoflurane (induction 3%, 0.8 L/min $O_2$; maintenance 1–1.5%, 0.5 L/min $O_2$) and perfused transcardially with saline followed by 4% paraformaldehyde. Brains were removed and post-fixed overnight in the same solution. Brains were coronal (for immunostaining) or horizonal (for virus tracing) sectioned into 40-μm thickness slices, which were all collected as long as containing hippocampus. Primary antibodies were as follows: nNOS (rabbit, 1:200, Cell Signaling, Danfoss, USA), cFOS (rabbit 1:100, SYSY, Göttingen, Germany), and mouse anti-nitrotyrosine (1:200; Santa Cruz Biotechnology, CA, USA). They were prepared in 0.1 M PBS with 3% goat serum and 0.3% Triton X-100 and visualized with a Cy3-conjugated secondary antibody (1:200; Thermo Fisher Scientific, Waltham, MA, USA). Nuclei were visualized with 4′-6-diaminodino-2-phenylindole (DAPI, Sigma-Aldrich). Every twelfth section throughout the hippocampus was processed for nNOS or cFOS immunohistochemistry and manually counted. The sections added up to 8 per hippocampus. Both hippocampi were counted. The overall septotemporal extent was analyzed. The areas of interest, including hilus, CA3, CA1, etc., had very distinctive features after DAPI staining and were identified by checking the brain atlas.

## NO content measurement (detection kit)

NO contents from the hippocampus were determined using a detection kit as previously reported (Guix et al, 2005). The DG of the hippocampus was homogenized in 10 volumes of deionized water and centrifuged at $1000 \times g$ for 15 min at 4 °C. NOx content was measured in supernatants using a commercially available kit (Jiancheng Bioengineering Co., Nanjing, China) and expressed as nanomole/mg protein.

## NO content measurement (electrochemical method)

NO contents from DG were measured using an ISO-NOP NO sensor (World Precision Instruments) connected to a TBR 4100 Free Radical Analyzer (World Precision Instruments) according to the manufacturers' instructions. The NO sensor was calibrated with SNAP (HY-121526, MedChem Express) using $CuCl_2$ as a catalyst. To calibrate the NO sensor, 1 mg SNAP was weighed and added to 50 mL of 0.1 M EDTA solution to prevent decomposition. Subsequently, a 0.1 M $CuCl_2$ solution (20 mL) was prepared, and a stirring plate was placed in a beaker. Three aliquots of SNAP solution (5 μL, 10 μL, 20 μL) were injected into the beaker to construct a standard calibration curve. For the actual measurement, DG was homogenized in 1 mL of deionized water. The NO sensor was then immersed in the solution to determine the concentration of NO.

## Transcriptome-seq and bioinformatic analyses

Transcriptome-seq was performed at LC-Bio, Hangzhou, China (Taleb et al, 2021). Briefly, total RNA was isolated from the hippocampi of mice. Approximately 10 μg of the total RNA was purified to isolate poly(A) RNA. After purification, the poly(A) RNA was fragmented into small pieces which were reversely transcribed to create the cDNA

libraries. DNAs were then selected and purified by UDG enzyme digestion and PCR, and the final paired-end libraries were formed (300 ± 50 bp). Paired-end sequencing was performed using the Illumina Novaseq™ 6000 according to the recommended protocols. Cutadapt-1.9 software was used to remove the reads that contained adapter contamination, low-quality bases, and undetermined bases to obtain Clean Data. After mapping to the genome of the Mus musculus (house mouse) using HISAT2 software, the reads of each sample were assembled using StringTie software and merged to reconstruct a comprehensive transcriptome using GffCompare software. With the final transcriptome was generated, StringTie and ballgown were executed to estimate the expression level for mRNAs by calculating FPKM. The differentially expressed genes were selected by integrating both the $p$ value and fold change of each gene ($p$ value < 0.05 and absolute $\log_2$ (fold change) ≥ 1) through R package, following Gene Ontology (GO) and Kyoto Encyclopedia of Genes and Genomes (KEGG) enrichment analysis.

## Nucleus isolation and single-nucleus RNA-Seq

Tissues samples were surgically removed and snap frozen in liquid nitrogen for intact nucleus isolation. The nucleus was isolated and purified as previously described with some modifications. Briefly, the frozen tissue was homogenized in NLB buffer which contain 250 mM Sucrose, 10 mM Tris-HCl, 3 mM $MgAc_2$, 0.1% Triton X-100 (Sigma-Aldrich, USA), 0.1 mM EDTA, 0.2 U/μL RNase Inhibitor (Takara, Japan). Various concentrations of sucrose were used to purify the nucleus. The concentration of nucleus was adjusted to about 1000 nuclei/μL for snRNA-Seq.

The snRNA-Seq libraries were generated using the 10X Genomics Chromium Controller Instrument and Chromium Single Cell 3′ V3.1 Reagent Kits (10X Genomics, Pleasanton, USA) at Shanghai Genechem Co., Ltd. Briefly, cells nuclei were concentrated to ~1000 nuclei/μL than loaded into each channel to generate single-cell Gel Bead-In-Emulsions (GEMs). After RT step, GEMs were broken and barcoded-cDNA was purified and amplified. The amplified barcoded cDNA was fragmented, A-tailed, ligated with adapters and index PCR amplified. The final libraries were quantified using the Qubit High Sensitivity DNA assay (Thermo Fisher Scientific, USA) and the size distribution of the libraries was determined using a High Sensitivity DNA chip on a Bioanalyzer 2200 (Agilent, USA). All libraries were sequenced by Novaseq6000 (Illumina, USA) on a 150 bp paired-end run.

## Whole-cell patch-clamp recordings

Brain slices were prepared with a Vibratome (VT1200s, Leica) in ice-cold, oxygenated artificial CSF (aCSF) containing 110 mM NaCl, 0.5 mM $CaCl_2$, 2.5 mM KCl, 7 mM $MgCl_2$, 1.3 mM $NaH_2PO_4$, 1.3 mM Na-ascorbate, 0.6 mM Na-pyruvate, 25 mM $NaHCO_3$, and 20 mM glucose and then incubated in warm oxygenated CSF (34 °C) for 1 h. Brain slices (350 μm thick) showing the hippocampal region were transferred into the recording chamber and super fused (2 ml/min) with oxygenated aCSF at room temperature (22–25 °C). Whole-cell patch recordings were performed with a computer-controlled amplifier (MultiClamp 700B, Molecular Devices). The pipettes for recording (3–4 MΩ) filled with intracellular solution containing 70 mM potassium gluconate, 70 mM KCl, 2 mM NaCl, 2 mM $MgCl_2$, 10 mM HEPES,

1 mM EGTA, 2 mM MgATP, and 0.3 mM NaGTP with pH 7.2–7.3 and osmolarity between 275 and 285 mOsmol. GFP-positive (GFP$^+$) neurons were targeted in DG under visual guidance of green fluorescence signals using an upright microscope (Examiner Z1, Zeiss). After a gigaohm seal, apply negative pressure to form whole-cell recordings.

For miniature postsynaptic current recordings (mEPSC and mIPSC), add 0.5 μM tetrodotoxin (TTX) to the bath solution to block spontaneous action potentials. 10 μM CNQX or 10 μM bicuculline methiodide was added to the bath solutions to block the mEPSCs and mIPSCs events respectively, then recorded the corresponding current separately.

Current clamp recordings were analyzed using Clampfit software. The resting membrane potential (RMP) was determined as the membrane voltage in current clamp mode immediately after achieving the whole-cell configuration. Input resistance (Rin) was calculated from the slope of the linear fit of the voltage-current (I–V) relationship, generated by applying a series of hyperpolarizing and depolarizing current injections ranging from −60 to +10 pA in steps of 10 pA. The AP firing output was quantified by counting the number of action potentials elicited in response to 500 ms long depolarizing current injections, which varied from 0 to 100 pA in steps of 10 pA.

To evoke EPSCs, electrical stimulation was applied using a bipolar tungsten microelectrode driven by a constant current stimulus isolation unit (Digitimer, Hertfordshire, England). The stimulation electrode (200-kΩ resistance, 50-μm intertip distance) was placed in the PPN, 50–200 μm away from the recorded neurons. Paired-pulse stimuli were delivered at 50-ms intervals every 10 s. Pulse duration was 0.1 ms, and stimulation voltage was adjusted to evoke a consistent response with no failure rate (1.5–1.7 times threshold). The paired pulse ratio was calculated as the ratio of the second evoked EPSC amplitude to the first evoked EPSC amplitude as an index of transmitter release probability. Paired-pulse stimuli were delivered at 50-ms intervals every 10 s.

## Transplantation of MGE cells

According to a previous report (Hunt et al, 2013), ventricular and subventricular layers of the MGE of E13 embryos from C57BL/6J mice were harvested for in vitro culture. After four days transduction with 1 μL of LV-nNOS-RNAi-GFP or LV-scramble-control-GFP, concentrated cell suspensions (~10$^3$ cells/nl) were front loaded into beveled glass micropipettes (40–50 μm tip diameter, Wiretol 5 μL, Drummond Scientific) and injected (3 × 10$^4$ cells per injection) into the hilus of the hippocampus. The stereotaxic coordinates are 2.3 mm posterior to bregma, 1.35 mm lateral to the midline, and 2.3 mm below dura. The number of survived GFP cells in single DG was confirmed as ≥10,000 cells (range = 10,000 to 15,000) and ≥600 μm from the injection site; these criteria were met in all mice.

## Behavioral tests

Mice were handled 5 min/day and habituated in the experimental apparatus 10 min/day for 1 week prior to training.

For Morris water maze (MWM) test, it was performed as described with a few adjustments (Zhou et al, 2017). The training paradigm for the visible platform version of the MWM consisted of

### The paper explained

**Problem**

Most antiseizure drugs developed for epilepsy treatment efficiently suppress status epilepticus, but do not impact the development of epileptogenesis, and can trigger drug resistance.

**Results**

We observed decreased nNOS expression in the hippocampus of patients with temporal lobe epilepsy (TLE). In transgenic mice, selective deletion of nNOS from hilar GABAergic interneurons induced epileptogenesis. Physiological levels of NO maintained the normal afferent circuit of dentate granular cells (DGCs), whereas chronic NO deficiency caused hyperexcitatory afferent circuit integration and hyperexcitability of DGCs. Replenishment of nNOS blocked the development of epileptogenesis and restored memory deficits in epileptic mice. Chronic NO donor treatment was sufficient to prevent the development of epileptogenesis.

**Impact**

Our findings reveal that NO donor prevents progression to TLE, highlighting a novel therapeutic strategy unrelated to neuronal ion channel blockers.

four trials (60 s maximum; 15 min interval) each day for 2 consecutive days. Following visible platform training, the training paradigm for the hidden platform version of the MWM consisted of four trials (maximum 60 s; 15 min interval) each day for 5 consecutive days. The probe trial was carried out 24 h after completion of training at day 6. The swimming paths were recorded with a video tracking system (EthoVision XT). Data from mice that were floating or jumping off the platform in the MWM test was excluded.

For Y-maze test, an alternation was that the mouse left one arm and entered the other. And spontaneous alternation was recorded when the mouse entered the arms sequentially. The ratio of spontaneous alternation was calculated as following: [(total number of spontaneous alternations)/(total number of alternations − 2)].

For novel object recognition test (NORT), two identical objects were placed in the experimental apparatus and mice placed in the presence of that to explore freely for 5–10 min. One day after the training, the mice were allowed to explore two objects again. For object local memory (OLM), the same objects, with one object moved to a novel location while the other was not. For object recognition memory (ORM) one object was replaced with a novel object, but the location remained. The discrimination index was used to represent relative exploration times [(time exploring novel object − time exploring familiar)/(time exploring novel + familiar) × 100].

## Statistics

All data were analyzed using GraphPad Prism 9 (Graphpad Software, Cary, NC, USA). After a homogeneity test of variance, when equal variances were assumed, unpaired Students' t-test was used to estimate the differences between two groups. Unpaired, nonparametric Kolmogorov–Smirnov test was performed for the nonparametric data of cumulative seizure score. One-way ANOVA was performed for comparison among three or four

groups, and to compare the effect of two factors on a numerical outcome, we used two-way ANOVA Bonferroni's multiple comparison corrections. Regarding self-comparison before and after treatment with DETA/NONOate, repeated measures, two-way ANOVA was applied. All experimental results were shown as mean ± s.e.m. When the $p$ value was less than 0.05, these results were considered significant. *$P < 0.05$, **$P < 0.01$, ***$P < 0.001$, NS, not significant. This study complied with randomization. The investigators were double-blind to the group allocation during the experiment and/or when evaluating the outcome. All detailed information for statistics (sample size, $t/F$, $df$, and $p$ value) of all experiments are provided in the supplementary statistical information tables.

## Data availability

Raw data of snRNA-seq of mouse DG samples have been deposited in the Gene Expression Omnibus under accession GSE274303. The source data of this paper are collected in the following database record: https://www.ebi.ac.uk/biostudies/studies/S-BSST1618.

The source data of this paper are collected in the following database record: biostudies:S-SCDT-10_1038-S44321-024-00168-1.

## Peer review information

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

## Acknowledgements

This research was supported by grants from STI2030-Major Projects (2022ZD0211700), the National Natural Science Foundation of China (82071525 and 82325048 to Q-GZ, 82001376 and 82371458 to FM) and Natural Science Foundation of Jiangsu Province (BK20200671 to FM). This study was also supported by the Collaborative Innovation Center for Cardiovascular Disease Translational Medicine for data collection, analysis, and interpretation.

## Author contributions

**Xian-Hui Zhu**: Data curation; Software; Investigation; Methodology; Writing—original draft. **Ya-Ping Zhou**: Resources; Data curation; Software; Methodology. **Qiao Zhang**: Data curation; Software; Methodology. **Ming-Yi Zhu**: Data curation; Software; Methodology. **Xiao-Wei Song**: Data curation; Software; Methodology. **Jun Li**: Data curation; Software; Methodology. **Jiang Chen**: Data curation; Software; Methodology. **Yun Shi**: Resources; Software. **Kang-Jian Sun**: Resources; Investigation. **Yong-Jie Zhang**: Resources; Investigation. **Jing Zhang**: Resources; Methodology; Writing—review and editing. **Tian Xia**: Data curation; Software; Methodology. **Bao-Sheng Huang**: Data curation; Validation; Writing—original draft; Writing—review and editing. **Fan Meng**: Conceptualization; Data curation; Supervision; Funding acquisition; Writing—original draft; Writing—review and editing. **Qi-Gang Zhou**: Conceptualization; Data curation; Supervision; Funding acquisition; Validation; Visualization; Writing—original draft; Project administration; Writing—review and editing.

Source data underlying figure panels in this paper may have individual authorship assigned. Where available, figure panel/source data authorship is listed in the following database record: biostudies:S-SCDT-10_1038-S44321-024-00168-1.

## Disclosure and competing interests statement

The authors declare no competing interests.

# Expanded View Figures

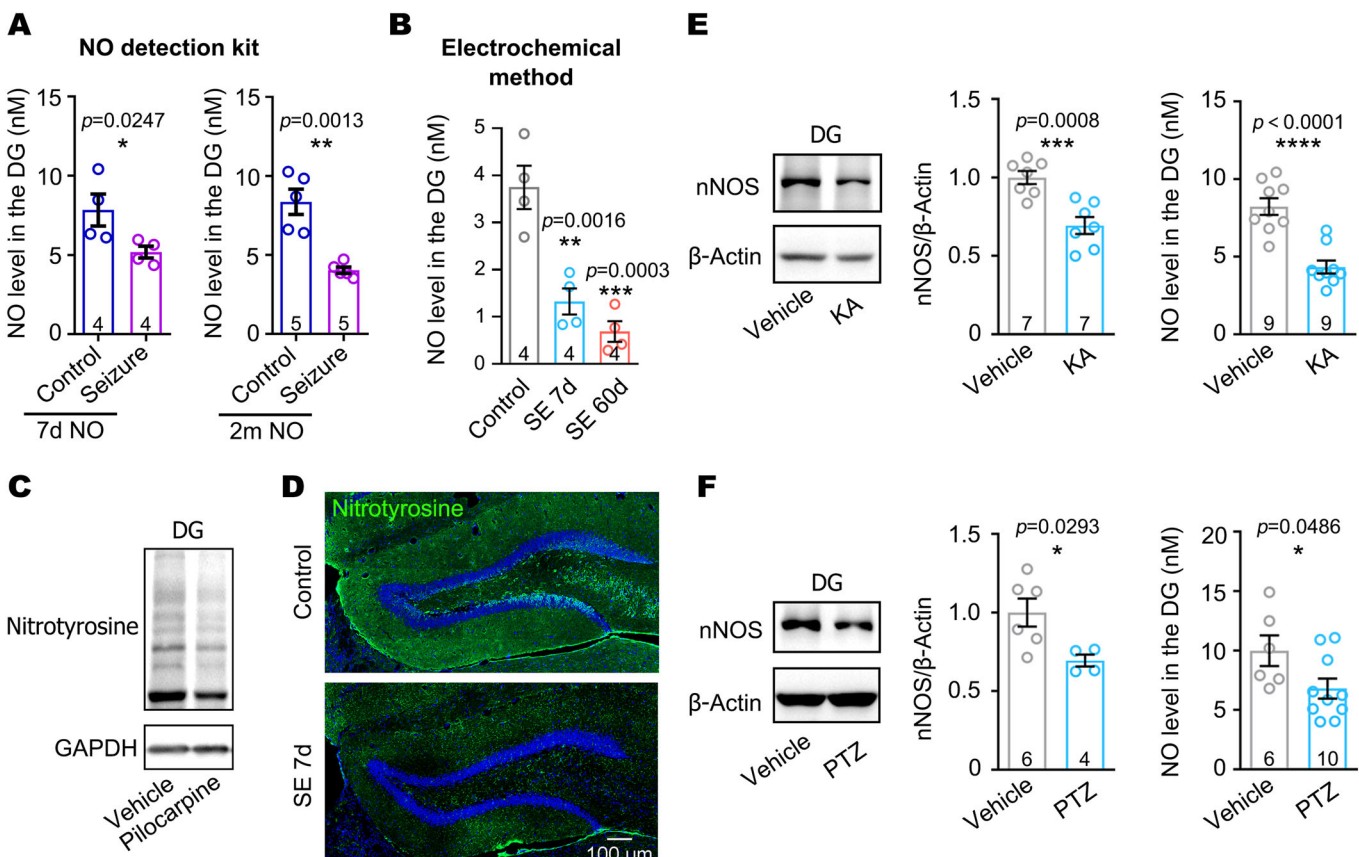

**Figure EV1. NO content decreases in TLE mouse models.**

(A) Data graph showing the concentration of NO in DG 7 days or 2 months after pilocarpine-induced SE using NO detection kit. Students' $t$-test, $n = 4$ mice. (B) NO measurements using electrochemical methods. One-way ANOVA, $n = 5$ mice. (C) Representative images showing decreased nitrotyosine-modified proteins in the DG 2 months after pilocarpine administration. The same result was observed in 5 mice. (D) Representative photos of immunofluorescence of nitrotyosine in the DG 7 days after pilocarpine-induced SE. (E) Representative image and data graph showing nNOS expression in DG 14 days after KA-induced SE. Students' $t$-test, $n = 7$–9 mice. (F) Representative image and data graph showing nNOS expression in DG 28 days after PTZ-induced SE. Students' $t$-test, $n = 4$–10 mice. Error bars correspond to ± s.e.m. $*P < 0.05$, $**P < 0.01$, $***P < 0.001$, $***P < 0.001$, $****P < 0.0001$.

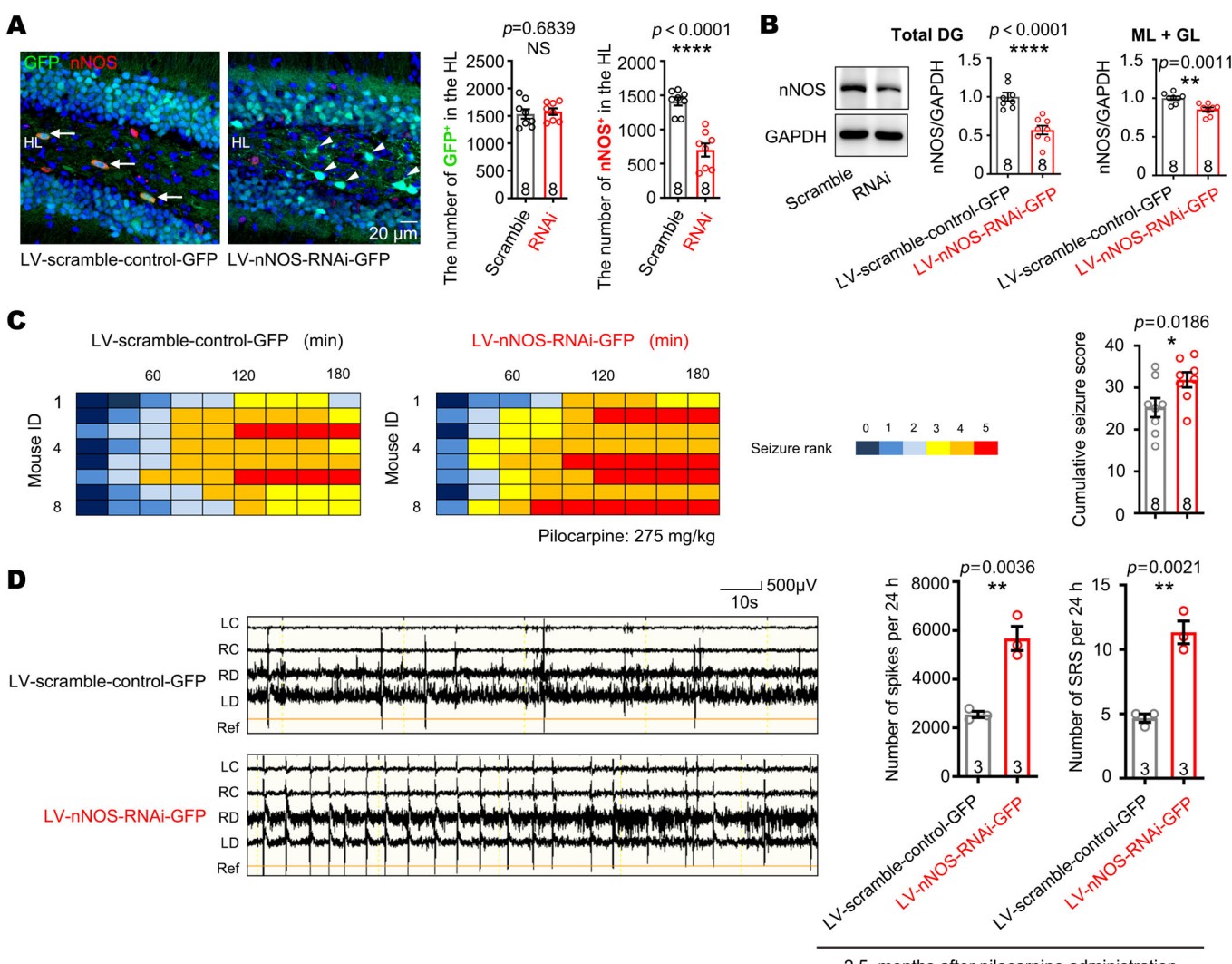

**Figure EV2. Knockdown of nNOS increases susceptibility of inducing status epilepticus and aggravates epileptogenesis in the pilocarpine mouse model of TLE.**

(A) Representative photos and data graphs of nNOS-positive/GFP-positive cells in the hilus 1 month after injection 1 μL of LV-scramble-control-GFP or LV-nNOS-RNAi-GFP into the hilus. Notably, less nNOS+ cells were observed after injection of LV-nNOS-RNAi-GFP. Arrows indicated transduced hilar nNOS-positive interneurons and arrowheads indicated transduced hilar nNOS-negative interneurons. The same observation was repeated in 5 mice. Student's t-test, $n = 8$ mice. (B) Western blot showing deceased content of nNOS protein in the DG and in the ML + GL regions 1 month after injection 1 μL of LV-nNOS-RNAi-GFP into the hilus. Student's t-test, $n = 8$ mice. (C) Heat map and data graph showing cumulative seizure score for 3 h after pilocarpine treatment at a dose of 275 mg/kg in mice. The mice received injection 1 μL of LV-scramble-control-GFP or LV-nNOS-RNAi-GFP into the hilus 1 month before pilocarpine administration. Student's t-test, $n = 12$ mice. (D) Representative images and data graphs of EEG recordings showing brain waves of mice 2.5 months after pilocarpine-induced SE. The mice received injection 1 μL of LV-scramble-control-GFP or LV-nNOS-RNAi-GFP into the hilus 1 month before pilocapine administration. Student's t-test, $n = 3$ mice. GL granule layer, ML molecular layer, HL hilus, LC left cortex, RC right cortex, RD right DG, LD left DG, Ref Reference. Error bars correspond to ± s.e.m. *$P < 0.05$, **$P < 0.01$, ****$P < 0.0001$, ****$P < 0.0001$, NS, not significant.

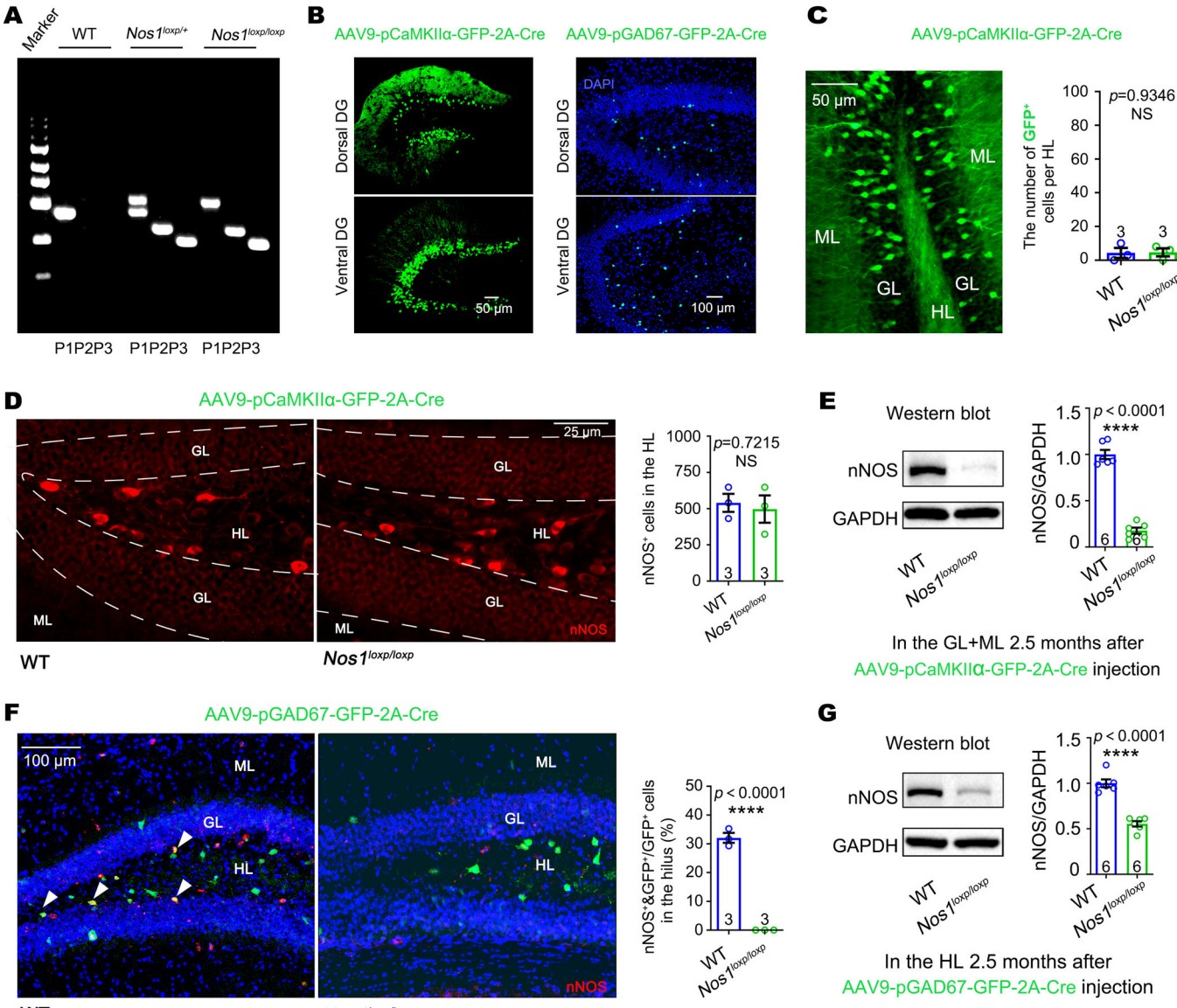

**Figure EV3. Increased susceptibility to SE induction after deletion of nNOS from hilar interneurons but not from DGCs.**

(A) Representative genotyping of Nos1loxp/loxp mice. P1, P2, and P3 showed productions synthesized by three primers, respectively. (B) The distribution of AAV9-pCaMKIIα-GFP-2A-Cre virus or AAV9-pGAD67-GFP-2A-Cre virus transduced neurons along the DG dorsal-ventral axis. The same pattern was observed in all injected mice. (C) Representative photo of AAV9-pCaMKII-GFP-2A-Cre transduction and data graph showing the number of GFP+ cells in the hilus were not altered. Student's t-test, n = 3 mice. (D) Representative photos and data graph showing nNOS-positive cells in the hilus 1 month after injection of AAV9-pCaMKIIα-GFP-2A-Cre into the hilus of WT or Nos1loxp/loxp mice. No significant difference was observed. Student's t-test, n = 3 mice. (E) The expression of nNOS protein in the granule layer and molecular layer were reduced after injection of AAV9-pCaMKIIα-GFP-2A-Cre virus into the hilus of Nos1loxp/loxp mice. Student's t-test, n = 6 mice. (F) Representative photos and data graphs showed that the number of nNOS+ cells in the hilus was reduced and no granule neurons were transduced. Arrowheads indicated transduced hilar nNOS+ neurons. Student's t-test, n = 3 mice. (G) The expression of nNOS protein in the hilus was reduced after injection of AAV9-pGAD67-GFP-2A-Cre virus into the hilus of Nos1loxp/loxp mice. Student's t-test, n = 6 mice. GL granule layer, ML molecular layer, HL hilus. Error bars correspond to ± s.e.m. ****P < 0.0001, NS, not significant.

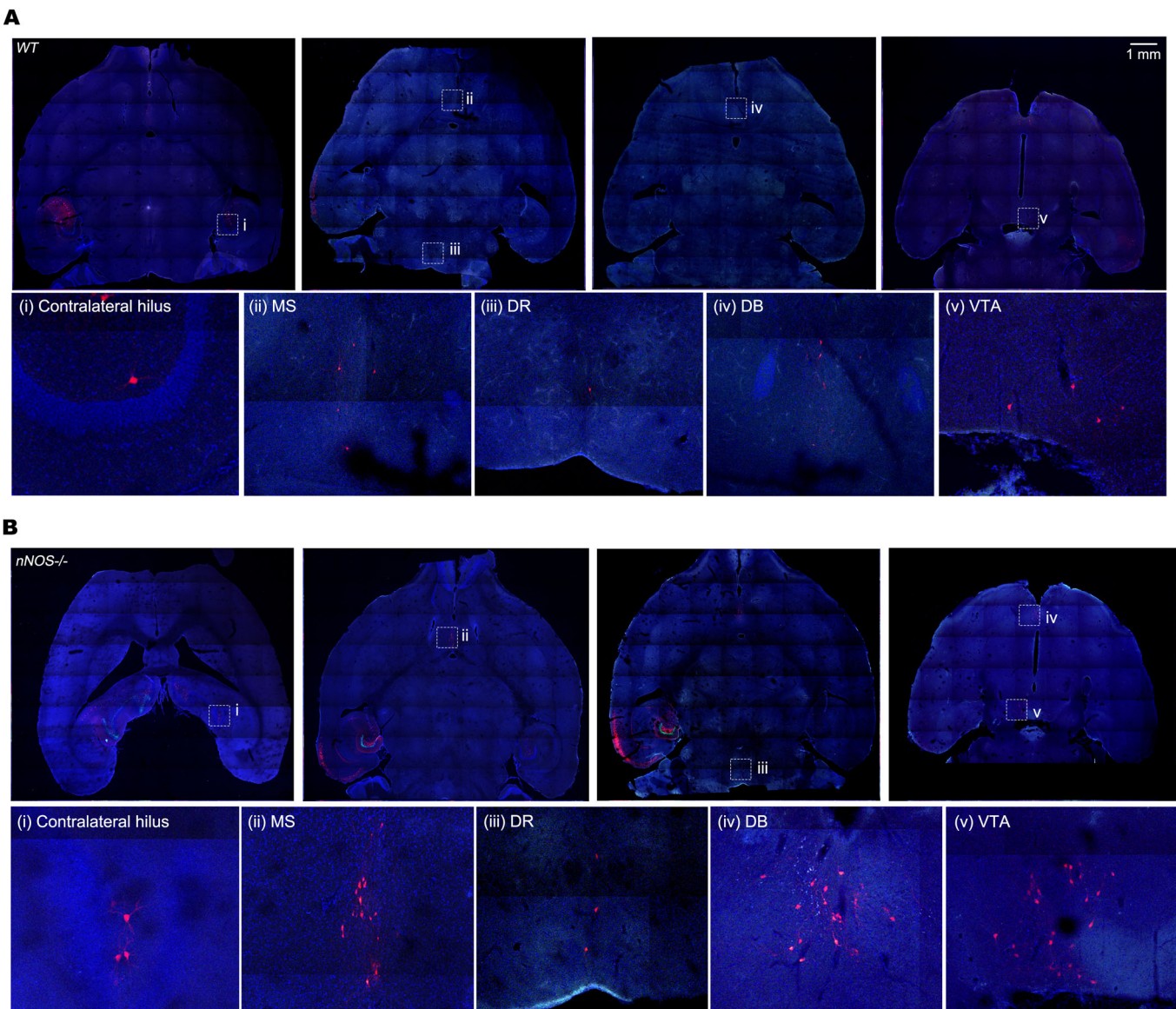

**Figure EV4. Distribution of traced cells projecting to DGCs.**

(A) Representative photos showing the distribution of traced cells projecting to DGCs in the brain of WT mice. (B) Representative photos showing the distribution of traced cells projecting to DGCs in the brain of $Nos1^{-/-}$ mice. (i–v) were zoomed photos from white squares, respectively. MS medial septum, DR dorsal raphe, DB diagonal band, VTA ventral tegmental area.

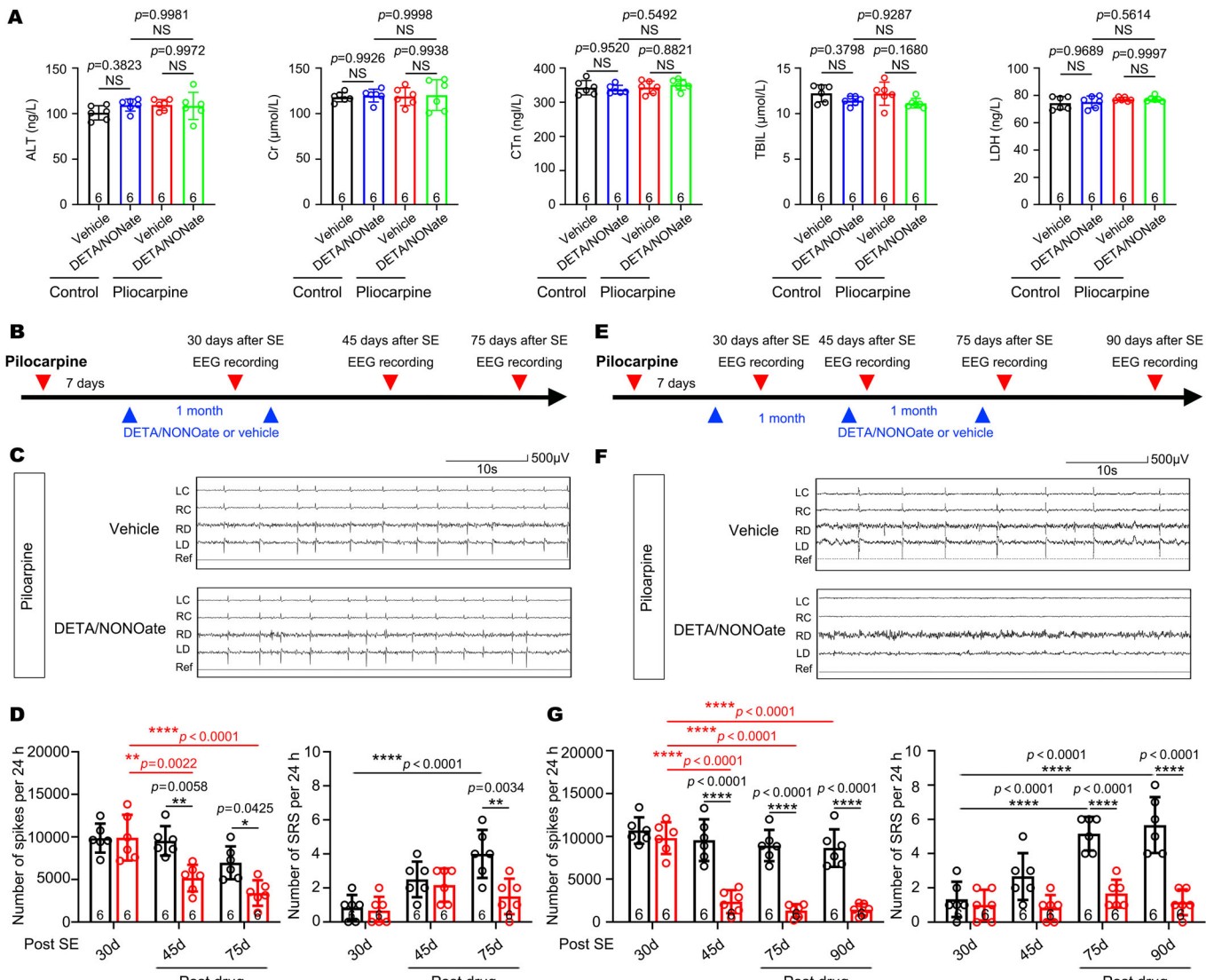

**Figure EV5. Chronic treatment with NO Donor prevents epileptogenesis with long-lasting effect.**

(A) Quantification of plasmatic ALT, TBIL, Cr, LDH, and CTn using ELISA kits. Two-way ANOVA, $n- = 6$ mice. (B–D) Experimental paradigm (B), representative EEG recordings (C), and data graphs (D), showing chronic treatment with DETA/NONOate (1 mg/kg, i.p., 1 time per day) for 1 month after pilocarpine-induced SE for 7 days prevented the development of SPKs and SRS. Two-way ANOVA, $n = 6$ mice. (E–G) Experimental paradigm (E), representative EEG recordings (F), and data graphs (G), showing chronic treatment with DETA/NONOate (1 mg/kg, i.p., 1 time per day) for 1 month after pilocarpine-induced SE for 37 days prevented the development of SPKs and SRS. Two-way ANOVA, $n = 6$ mice. ALT alanine aminotransferase, TBIL total bilirubin, Cr creatinine, LDH lactate dehydrogenase, CTn cardiac troponin. Error bars correspond to ± s.e.m. *$P < 0.05$, **$P < 0.01$, ****$P < 0.0001$, NS, not significant.

