## [Peer Review File · EMBO Molecular Medicine]

A novel anti-epileptogenesis strategy of temporal lobe epilepsy based on nitric oxide donor

Xian-Hui Zhu, Ya-Ping Zhou, Qiao Zhang, Ming-Yi Zhu, Xiao-Wei Song, Jun Li, Jiang Chen, Yun Shi, Kang-Jian Sun, Yong-Jie Zhang, Jing Zhang, Bao-Sheng Huang, Fan Meng, Qi-Gang Zhou

Corresponding authors: Bao-Sheng Huang (bs.huang@njmu.edu.cn), Fan Meng (mengfan199005@njmu.edu.cn), Qi-Gang Zhou (qigangzhou@njmu.edu.cn)

Review Timeline:

Submission Date:	13th Apr 24
Editorial Decision:	6th May 24
Revision Received:	31st Aug 24
Editorial Decision:	1st Oct 24
Revision Received:	10th Oct 24
Editorial Decision:	17th Oct 24
Revision Received:	22nd Oct 24
Accepted:	28th Oct 24

Editor: Lise Roth

Transaction Report:

6th May 2024

Dear Dr. Zhou,

Thank you for submitting your work to EMBO Molecular Medicine. We have now heard back from the three referees who agreed to evaluate your manuscript. As you will see below, the reviewers raise substantial concerns on your work, which unfortunately preclude its publication in EMM in its current form.

The reviewers all mention the novelty and potential interest of the findings, but also raise several concerns related to the lack of thorough experimental description, statistics, quantification, controls, which overall question the validity of the conclusions presented.

If you feel you can satisfactorily address all points listed by the referees, including language editing, you may wish to submit a revised version of your manuscript. Please attach a covering letter giving details of the way in which you have handled each of the points raised by the referees. A revised manuscript will once again be subject to review, and we cannot guarantee at this stage that the eventual outcome will be favorable.

We are expecting your revised manuscript within three-four months, if you anticipate any delay, please contact us.

We require:

- 1) A .docx formatted version of the manuscript text (including legends for main figures, EV figures and tables). Please make sure that the changes are highlighted to be clearly visible.
- 2) Individual production quality figure files as .eps, .tif, .jpg (one file per figure). For guidance, download the 'Figure Guide PDF' (<https://www.embopress.org/page/journal/17574684/authorguide#figureformat>).
- 3) At EMBO Press we ask authors to provide source data for the main figures. Our source data coordinator will contact you to discuss which figure panels we would need source data for and will also provide you with helpful tips on how to upload and organize the files.
- 4) A .docx formatted letter INCLUDING the reviewers' reports and your detailed point-by-point responses to their comments. As part of the EMBO Press transparent editorial process, the point-by-point response is part of the Review Process File (RPF), which will be published alongside your paper.
- 5) A complete author checklist, which you can download from our author guidelines (<https://www.embopress.org/page/journal/17574684/authorguide#submissionofrevisions>). Please insert information in the checklist that is also reflected in the manuscript. The completed author checklist will also be part of the RPF.
- 6) Please note that all corresponding authors are required to supply an ORCID ID for their name upon submission of a revised manuscript.
- 7) It is mandatory to include a 'Data Availability' section after the Materials and Methods. Before submitting your revision, primary datasets produced in this study need to be deposited in an appropriate public database, and the accession numbers and database listed under 'Data Availability'. Please remember to provide a reviewer password if the datasets are not yet public (see <https://www.embopress.org/page/journal/17574684/authorguide#dataavailability>). In case you have no data that requires deposition in a public database, please state so in this section. Note that the Data Availability Section is restricted to new primary data that are part of this study.
- 8) For data quantification: please specify the name of the statistical test used to generate error bars and P values, the number (n) of independent experiments (specify technical or biological replicates) underlying each data point and the test used to calculate p-values in each figure legend. The figure legends should contain a basic description of n, P and the test applied. Graphs must include a description of the bars and the error bars (s.d., s.e.m.). Please provide exact p values.
- 9) Our journal encourages inclusion of *data citations in the reference list* to directly cite datasets that were re-used and

obtained from public databases. Data citations in the article text are distinct from normal bibliographical citations and should directly link to the database records from which the data can be accessed. In the main text, data citations are formatted as follows: "Data ref: Smith et al, 2001" or "Data ref: NCBI Sequence Read Archive PRJNA342805, 2017". In the Reference list, data citations must be labeled with "[DATASET]". A data reference must provide the database name, accession number/identifiers and a resolvable link to the landing page from which the data can be accessed at the end of the reference. Further instructions are available at .

13) Author contributions: CRediT has replaced the traditional author contributions section because it offers a systematic machine readable author contributions format that allows for more effective research assessment. Please remove the Authors Contributions from the manuscript and use the free text boxes beneath each contributing author's name in our system to add specific details on the author's contribution. More information is available in our guide to authors.

16) As part of the EMBO Publications transparent editorial process initiative (see our Editorial at <http://embomolmed.embopress.org/content/2/9/329>), EMBO Molecular Medicine will publish online a Review Process File (RPF) to accompany accepted manuscripts.

In the event of acceptance, this file will be published in conjunction with your paper and will include the anonymous referee reports, your point-by-point response and all pertinent correspondence relating to the manuscript. Let us know whether you agree with the publication of the RPF and as here, if you want to remove or not any figures from it prior to publication.

I look forward to receiving your revised manuscript.

Yours sincerely,

Lise Roth

**** Reviewer's comments ****

Referee #1 (Comments on Novelty/Model System for Author):

Although the topic is of interest and the model system is appropriate, some findings were not supported by the data. Also, this manuscript is poorly written that requires substantial proofreading.

Referee #1 (Remarks for Author):

The authors of this study characterize how selective deletion of nNOS in hilar interneurons of the hippocampal DG can cause detrimental effects, leading to increased aberrant excitatory inputs and epilepsy. To demonstrate this, they created a conditional knockout nNOS mouse model and performed several experiments. Furthermore, they showed that such reductions in nNOS occur in patients diagnosed with drug-resistant epilepsy. In addition, the authors suggested that NO donors such as DETA/NONONOate could potentially be used as a new class of drugs with anti-epileptogenic properties. Overall, this study aimed to address an important question and utilized carefully planned experiments that addressed some of the conflicting roles attributed to NO in TLE. In addition, they made an effort to identify a specific cell type and a hippocampal sub-region that play a role in nNOS-associated hyperexcitability. However, the study also suffers from some major weaknesses that somewhat reduce the impact of this study. Specific comments are as below.

1. No sample size values were provided for any of the experiments in the manuscript or in the figure legends. From the dots in the graphs, some data contain more than 10 repeats but some have only 3. How these numbers were determined is unclear.
2. Related to the point above, some low numbers of samples reduce the confidence to some key findings. For example in Figs 2D and 2F, there were only 3 patients in the group. However, it is hard to imagine that 3 patients with different genders, age and background could show nearly identical nNOS expression level. Another example is Fig-2E. Is it the case that the authors only recorded 3-4 mice? For Fig-6C, were there only 3-4 mice per condition? The authors need to significantly increase the size of the samples.
3. Many experiments were performed at a fixed time point without justification. For example, what is the reason of treating mice with DETA/NONONOate for 68 days? What is the reason of waiting for 7 days before injecting the drug?
4. Related to the point above, elevating NO for 68 days throughout the body is expected to bring strong adverse effects to the animal. It would have been ideal if the treatment period were 2-3 weeks and an additional post-treatment observation to verify if the drug indeed had an long lasting effect on epileptogenesis.
5. It is a common practice to show western blotting results with representative blots. For example, the authors should include blots for Figs 2J, 5A, 5C, etc.
6. The analysis of EEG data for only one day is not sufficient; in these situations, it is best to monitor the seizure frequency for multiple days before, during and after treatment. In addition, a comparison of the resting-state power spectrum of EEG is necessary to identify normal brain waves. In statistics section, authors claim to have performed before and after treatment analysis using repeated measures 2-way ANOVA, but data for the same cannot be found in the manuscript.
7. The authors performed only Morris Water Maze and claimed that their approach restored pilocarpine-induced cognitive

defects. Multiple behavioral experiments are needed to achieve this conclusion.

8. The authors should improve their discussion section. First, the contradictory observation of the present study compared to the past has not been sufficiently addressed. For example, Yao et., 2022 have shown in Pilocarpine model nNOS level in hilus to increase at day 14 and back to basal levels by day 28 (<https://www.sciencedirect.com/science/article/pii/S0891584922004488?via%3Dihub>). Also the function of nNOS in neuronal excitability has been extensively studied, which should be discussed.

9. There are several issues with acronym. First, acronym for DGCs was not provided when first used. What are SPKs? In Figure 2 all the graphs use the acronym TRE. However, in the legend only DRE is mentioned. ASD is extensively used for autism spectrum disorders. Most people use AED for anti-epileptic drugs. Please consider change that.

10. Lastly, this manuscript, including figure legends and supplemental information, requires extensive proofreading. Numerous typos and odd sentences prevent a clear understanding of this study. Several examples are as below:

"Epilepsy is the second incidence rate neurological disorder in the world."

"There is a statistical correlation that once the hippocampus develops into sclerosis" (how can hippocampus "become" sclerosis)

"meanwhile, the content of hilar nNOS was decreased in pilocarpine-induced epileptic mice." (assuming the authors meant the "expression level" of nNOS.)

"Different from low intrinsic excitatory DGCs that powerful inhibited by hilar interneurons in physiological condition, ..." (unable to comprehend this sentence)

Also, the authors should use "transduce" instead of "infect" when using virus to deliver genes.

Referee #2 (Comments on Novelty/Model System for Author):

Explanation for medium technical quality (although a large series of complex state-of-the-art techniques have been used): All statistical analyses lack information. Please give p-values for the ANOVA (not only the post-tests), please also add p-values for the post-tests (e.g. in supplementary tables) such that the reader gets a better estimation of the results of statistical testing. Adding stars to the bar graphs is not sufficient for the evaluation of the data. Just as one example: Fig. 5 L, right part, it is hard to believe that there is no significant difference between Sham and LV-nNos-RNAi-GFP, the p-values would thus be relevant. In addition, please add the N of the experiments in the text or in the legends or at least in a supplementary table. Although the results for individual mice are shown, it is tedious to always count the dots to the N value.

Referee #2 (Remarks for Author):

The authors have done an excessive amount of different experiments, which is very impressive. Overall, these experiments fit well together and are logical steps building upon each other to investigate the entire story of the paper. The results from the rescue experiments are very impressive and promising for novel therapeutic options. However, the experiments sometimes need more description, as it sometimes does not become clear from the main text or the Figure legend, why and experiment has been performed the way it has been performed (e.g. if different time points are analyzed). Some details on how the experiments have been done are also missing in the methods. In addition, sometimes it feels as if in the large number of experiments and parallel controls, the precision in describing the individual experiment is lost. In addition, the measurements performed are different for the various controls, which also makes it difficult to follow and get the entire picture. It is very tedious for the reader to get the role of the individual experiment and the main story line from time to time. A clearer separation of the main experiments, presented in the main Figures and the control experiments, presented in the supplementary Figures might help to guide the reader through the paper here.

1. The methods are lacking some information: What kind of recording system was used for the EEGs (incl. amplification, sampling rate etc.)? How often were the mice recorded during which periods? Were the analyses done for single days (the ones given in the Figure legends) or for more days? Furthermore, in many experiments, the authors counted cells that were stained with a certain antibody. Except for the information that every 12th section was counted, no further information is given in the methods or suppl. Methods on how counting was performed. Please explain how slices were selected, which overall septo-temporal extent was analyzed, how the areas of interest were determined and so on. This is needed for the reader to evaluate what has been really done.

2. All statistical analyses lack information. Please give p-values for the ANOVA (not only the post-tests), please also add p-values for the post-tests (e.g. in supplementary tables) such that the reader gets a better estimation of the results of statistical testing. Adding stars to the bar graphs is not sufficient for the evaluation of the data. Just as one example: Fig. 5 L, right part, it is hard to believe that there is no significant difference between Sham and LV-nNos-RNAi-GFP, the p-values would thus be

relevant. In addition, please add the N of the experiments in the text or in the legends or at least in a supplementary table. Although the results for individual mice are shown, it is tedious to always count the dots to the N value.

3. What is the reason of presenting Fig. 2, which shows the general change in nNOS in humans and the pilocarpine model (without any KO), after Fig. 1 which already determines more detailed mechanisms underlying this observation? Wouldn't the opposite sequence be more logical? It is also hard for the reader to jump from different KO mice in Fig. 1 to WT in Fig. 2, back to KO in Fig. 3. In addition, then, the paper would start with the pilocarpine model and end with rescue in the pilocarpine model, which seems more logical to me.

4. Fig. EV1 and corresponding main text: Why is the dose used for EEG recordings in the chronic stage of epilepsy different (200 mg/kg) from the two doses used to analyze the severity of the status epilepticus (230 mg/kg and 100 mg/kg) (see Fig. EV1 and corresponding results text)?

5. Fig. EV1 and corresponding main text: The counting of epileptic spikes and seizures was performed on day 75, but day 45 is also shown in Fig. EV1G. Has the counting only be performed on day 75? What is the reason to choose that day? What is the reason to show day 45 in addition? Is there something different between those days, or is day 45 the first day when seizures could be observed? This does not become clear in the text. In addition, in the methods the authors describe that spikes have been manually quantified in a 3h window. Was this 3-h count multiplied, such that the number given is supposed to reflect a 24h period? In addition, was the time window chosen for analysis always during the same time of the day for all mice (diurnal or nocturnal changes could affect spike numbers)?

6. Please add the quantification of spontaneous epileptic spikes in Nos1^{-/-} mice in Fig. EV1 E to make it comparable to the values of pilocarpine-injected mice.

7. Was the lentivirus vector expressing small hairpin RNA (Fig. EV2) injected into the hilus of one or both hemispheres? See also comment above: How was the counting of nNOS-positive cells performed (which are of interest, how many sections, were both hippocampi counted or only one etc.)?

8. Some concerns about the Western blots: In some of the Western blots, one band is visible for nNOS (e.g. Fig. EV2B), in others two bands are visible. What is the reason for that? In addition, in Fig. 1E, the background level of the GAPDH bands is very different from the nNos bands. Is this really from the same Western blot? In addition, Fig. 1E shows the Western blot for nNOS for the AAV with CamKII promotor. The corresponding Western plot of the GAD 67 promotor is missing. Please add for completeness. In Fig. EV5C, which of the many band represents the Nitrotyrosine?

9. In the snRNAseq analysis (Figure 1) are very large group or 'other neurons' is found. As only the dentate gyrus was prepared and sequenced, what type of neuron should that be if not excitatory or inhibitory? Are the markers (CamKII and GAD) really good markers to distinguish neurons under epileptic conditions, given that in many epilepsy models granule cells of the DG upregulate GAD under epileptic conditions (see papers from G. Sperk and others)? Besides differential regulation of Nos1 (interestingly a downregulation in interneurons and an upregulation, which is not commented in the text in excitatory neurons) a series of other differentially regulated genes is shown in Fig. 1B. Are the genes shown in B the only ones with differential expression in excitatory or inhibitory neurons or why is this exactly this selection given? In addition, to keep consistency, shouldn't there be Nos1 instead of nNos when representing differentially expressed genes?

10. It is surprising, that a selective KO of Nos1 in inhibitory cells of the hilus leads to epileptic spikes and SRS at 2.5 months after the AAV injection without any pilocarpine injection (Fig. 1I), whereas the constitutive full KO of Nos1 (Nos1^{-/-} mice) only induces epileptic spikes but no SRS. What is the hypothesis of the authors why the more selective approach has a more severe outcome? Compensatory processes in the constitutive KO or other mechanisms? This should be discussed.

11. Fig 2G, 2J and corresponding main text: In the main text, the authors claim a reduction of Nos1 mRNA or nNOS protein at 7d, followed by a maintenance at 14d and 2m. However, in the plot it seems very much as if there was a further decline instead of a maintenance. Please test for this and eventually change the text accordingly. For the two-way ANOVA, please give the variables, which were tested, the results of the ANOVA for the individual values and not only the results of the post-test in form of stars in the image. In addition, please describe what was tested with the post-test, e.g. for Fig. 2G and J, just naming the two-way ANOVA is not sufficient.

12. In the rabies tracing experiments (Fig. 3A, B), I am very surprised that there is no contralateral labeling. Shouldn't contralateral mossy cells project onto ipsilateral granule cells (see different studies of H. Scharfman). Why is there no labeling visible in the septum/DBB? The authors analyze forebrain regions and show a significant difference in the quantification, but I cannot see that in any of that images. How is the number of input cells in distant areas normalized to the number of infected cells in the granule cell layer? Please add separate enlarged view of the dentate gyrus for WT and Nos1^{-/-} mice with green and red channel separated. It seems as if the rate of infection is different in the different genotypes. For quantifying cells in the brainstem - which areas of the brainstem were selected for the counting? Similar for the forebrain - both are large areas, how were the areas of interest for counting selected?

13. How do the authors explain the strong effects after cPTIO. I can imagine substantial differences between WT and Nos1^{-/-} mice, which might develop during development. But the strong differences after a chemical (not-hippocampus-specific, not 100%) NO blockade inducing such massive sprouting from various distant regions is hard to believe. Why did they choose this chemical approach and not their more selective knockdown approaches, which have been used earlier in the paper? Why don't they show the fiber sprouting effects in pilocarpine mice in more detail also here, as these are relevant for the rescue experiments presented later.

14. In Fig. EV6 A and the corresponding main text, the authors apply the NO blocker c-PTIO. The Figure then uses the terms Control and Seizure. However, the existence of seizures after the application of c-PTIO has not been described before. Either the analysis of seizures after this application is necessary or the Figure has to be labelled differently. Similar for EV6B: Does the application of c-PTIO lead to a status epilepticus as the Figure implies? This remains unclear.

15. Fig. 4: The massive upregulation of cFOS in granule cells after injection of the AAV9-pGAD67 virus is surprising compared to the constitutive KO. Why is this the case?

16. The language needs some serious revision by a native speaker or someone sufficiently proficient in English. There are many grammatical errors which make the paper at times hard to read and understand (e.g. lack of articles, lack of prepositions, wrong verb forms, use of plural at the wrong sites and lack of a required plural at other sites and so on). Please correct these throughout the paper, including the Figure legends. In addition, there are many typos, also in the Figures (e.g. Pilocarpine in Fig. 6), some axis labels are misplaced and so on. Overall, the paper needs some serious checking and correcting.

Some minor comments:

1. Different isoforms of Nos (nNOS, iNOS, eNOS) are named in the paper. Please add information on these isoforms to the introduction as not all readers are familiar with the roles and specificities of the different isoforms.

2. How long after death were the post-mortem samples collected, that were used for the analysis of human tissue?

3. Fig. EV4B: How was the cortical delay analyzed? What EEG signature was taken to detect the activity in the cortex? Are the values given the mean delays for all seizures in each mouse? How consistent or variable are the onset delays within individual mice?

4. The methods mention CNO infusion in slices. It is unclear to me for which of the patch-clamp experiments this was relevant.

5. The text mentions a Fig. 4C when describing the tracing - that should probably be Fig. 3C.

6. Introduction: 'Besides, multiple histopathological changes, such as neurons loss, synaptic reorganization, gliosis, and altered neurogenesis in the dentate gyrus (DG) of the hippocampus, also contribute to the development of hippocampal sclerosis.' Hippocampal sclerosis is defined as cell loss and gliosis (see several papers e.g. of I. Blümcke, M.C. Walker and others). Other histological patterns of TLE like neurogenesis, mossy fiber sprouting etc. come in conjunction with hippocampal sclerosis but are not part of it. The sentence is thus misleading.

7. Introduction: However, it remains unclear whether nNOS⁺ interneurons will be damaged or preserved in the epileptic hippocampus and how important role of nNOS plays in the pathology of TLE. Something is wrong in the highlighted part.

8. Methods: Animals that received pilocarpine and did not develop convulsive seizures higher than stage 4 were excluded. Do the authors mean 'equal and higher than' or do they really mean only 'higher than', i.e. only mice that reach stage 5 were used for analysis?

9. Results, 1st section: ...driven by promoter CaMKII α (AAV9-pCaMKII α -Cre) (Fig. 1C) or GABA (AAV9-pGAD67-GFP-2A-Cre) (Fig. 1F) into the bilateral DGs of WT...The promoter seems to be GAD67, not GABA, please change. In addition, what is the bilateral DG? This is either a bilateral injection into the DG or an injection into both DGs but there is no such thing as a bilateral DG.

10. What is shown in Fig. EV3B? What does P1P2P3 stand for? The legend is insufficient.

11. Fig. EV4B: prior instead of priority.

12. Fig. 2B: Please mention the role of the red circles in the scheme and the red squares in the EEG. Why are these chosen?

13. Fig. 2C: The squared areas are hard to see. Please also explain in the legend why these areas are highlighted.

Referee #3 (Comments on Novelty/Model System for Author):

The authors investigated the role of nitric oxide (NO) signaling from nNOS+ inhibitory interneurons of the dentate gyrus in contributing to temporal lobe epilepsy (TLE). In a previous study, they show in a pilocarpine mouse model that induction of TLE leads to aberrant innervation of dentate granule cells by entorhinal cortex. Here, using conditional knockout of Nos1 from interneurons and dentate granule cells, rescue of Nos1, cFos staining, in vivo and in vitro electrophysiology, and monosynaptic retrograde rabies tracing, they show that loss of NO release from nNOS+ interneurons causes this aberrant circuit formation and development of TLE. This finding is novel, the experiments are thorough, and the data are convincing. Furthermore, identifying cell-type-specific mechanisms is crucial for developing targeted therapeutic approaches.

Referee #3 (Remarks for Author):

Addressing the following would improve the manuscript:

- 1) In Figure EV2A, viral-induced GFP and IHC-RFP for nNOS do not overlap. Thus, in this example, it appears the lentivirus has not infected nNOS+ cells.
- 2) The methods section description of whole-cell patch clamp electrophysiology is both incomplete and inaccurate. The authors state they performed "cell attached" recording, but that their data in Figure 4 are whole-cell recordings in voltage clamp. This section needs to be updated with proper method details. Furthermore, what were the ages of the mice used for these experiments?
- 3) If DG cells receive more excitatory drive, which contributes to seizures, how do the authors explain the lack of increased cFos staining in CA3? Wouldn't you expect excessive drive from mossy fiber inputs to cause increased activity in CA3?
- 4) Do the authors have any data on whether loss of NO signaling alters the intrinsic membrane properties of dentate granule cells?
- 5) N's (mice, cells, etc.) are missing throughout.
- 6) On page 8, "Fig. 3C" is mislabeled as "Fig. 4C".
- 7) This manuscript needs editing for grammar.

We sincerely appreciate the reviewer's constructive suggestions and corrections. We have addressed each of these comments below, with highlighted new text that has been added to the manuscript and supplementary documents.

Referee #1 (Comments on Novelty/Model System for Author):

Although the topic is of interest and the model system is appropriate, some findings were not supported by the data. Also, this manuscript is poorly written that requires substantial proofreading.

Referee #1 (Remarks for Author):

The authors of this study characterize how selective deletion of nNOS in hilar interneurons of the hippocampal DG can cause detrimental effects, leading to increased aberrant excitatory inputs and epilepsy. To demonstrate this, they created a conditional knockout nNOS mouse model and performed several experiments. Furthermore, they showed that such reductions in nNOS occur in patients diagnosed with drug-resistant epilepsy. In addition, the authors suggested that NO donors such as DETA/NONOate could potentially be used as a new class of drugs with anti-epileptogenic properties. Overall, this study aimed to address an important question and utilized carefully planned experiments that addressed some of the conflicting roles attributed to NO in TLE. In addition, they made an effort to identify a specific cell type and a hippocampal sub-region that play a role in nNOS-associated hyperexcitability. However, the study also suffers from some major weaknesses that somewhat reduce the impact of this study. Specific comments are as below.

1. No sample size values were provided for any of the experiments in the manuscript or in the figure legends. From the dots in the graphs, some data contain more than 10 repeats but some have only 3. How these numbers were

determined is unclear.

Response:

1) About no sample size values:

We apologize for the missing information of the sample size values. Your reminder is very necessary.

As suggested by you, we have added all the sample size values of the experiments in the legends and supplementary file named 'Supplemental statistics information'.

2) About the determination of repeats numbers:

In general, we used > 12 biologically independent samples for whole cell patch-clamp recordings and behavioral tests, 3 biologically independent samples for experiments related to patients, and ≥ 3 biologically independent samples for other experiments including western blot, RT-qPCR, EEG recording, neural circuit, image statistics, etc.

The specific numerical values of each experiment presented in the paper was determined by the success rate of establishing epilepsy mouse models, as well as virus injection.

As suggested in the next comment, we have expanded sample size for various experiments. Thank you for the comment.

2. Related to the point above, some low numbers of samples reduce the confidence to some key findings. For example in Figs 2D and 2F, there were only 3 patients in the group. However, it is hard to imagine that 3 patients with different genders, age and background could show nearly identical nNOS expression level. Another example is Fig-2E. Is it the case that the authors only recorded 3-4 mice? For Fig-6C, were there only 3-4 mice per condition? The authors need to significantly increase the size of the samples.

Response:

We are sorry for the low numbers of human samples due to that human

brain samples are precious and rare and that we set strict inclusion criteria. As suggested, we have tried our best to expand the samples from 3 to 6, which supported the same conclusion, as shown in new Fig. 1DEF.

As suggested by you, we also have expanded sample sizes for various mice experiments, as shown in new Fig. 1J, Fig. 4C, Fig. 6CD, Fig. 7CF, Fig. EV3E, etc.

Thanks for the suggestion.

3. Many experiments were performed at a fixed time point without justification. For example, what is the reason of treating mice with DETA/NONOate for 68 days? What is the reason of waiting for 7 days before injecting the drug?

Response:

In our pilocarpine model, mice developed stable spontaneous recurrent seizures, which is a core feature of epilepsy, about 75 days after pilocarpine administration. And we found a significant reduction in nNOS expression and NO content 7 days after pilocarpine administration. Meanwhile, mice need 7 days to recover from pilocarpine-induced status epilepticus (SE). Accordingly, we treated mice with DETA/NONOate during the special time window. We have provided detailed information about pilocarpine-induced TLE model in the methods section. Thanks!

4. Related to the point above, elevating NO for 68 days throughout the body is expected to bring strong adverse effects to the animal. It would have been ideal if the treatment period were 2-3 weeks and an additional post-treatment observation to verify if the drug indeed had an long lasting effect on epileptogenesis.

Response:

1) Regarding the adverse effects of elevating NO for 2 months

We appreciate you for raising this concern. To examine the systemic toxicity of DETA/NONOate treatment for 2 months, especially the toxicity to liver,

kidney, and heart, we detected the levels of alanine aminotransferase (ALT), total bilirubin (TBIL), creatinine (Cr), lactate dehydrogenase (LDH) and cardiac troponin (CTn) in mice serum by ELISA kit. Notably, 2 months of continuous exposure to DETA/NONOate did not lead to obvious liver injury indicated by measurement of ALT and TBIL, or kidney injury indicated by measurement of Cr and LDH or heart injury indicated by measurement of CTn (new Fig. EV5A).

2) Regarding the treatment period of 2-3 weeks and an additional post-treatment observation

As suggested, we treated mice with DETA/NONOate at different periods post SE and found that replenishment of NO during the early phase of epileptogenesis (the first month after pilocarpine administration) attenuated the development of SPKs (new Fig. EV5BCD), but replenishment of NO during the late phase of epileptogenesis (the second month after pilocarpine administration) was sufficient to block the development of epileptogenesis, and the effect of which last at least for 1 month post treatment (new Fig. EV5EFG). Thanks for the suggestions!

New Fig. EV5

5. It is a common practice to show western blotting results with representative blots. For example, the authors should include blots for Figs 2J, 5A, 5C, etc.

Response:

We have added representative blots for new Figs 1I, 6B, 6C, etc. Thanks!

6. The analysis of EEG data for only one day is not sufficient; in these situations, it is best to monitor the seizure frequency for multiple days before, during and after treatment. In addition, a comparison of the resting-state power spectrum of EEG is necessary to identify normal brain waves. In statistics section, authors claim to have performed before and after treatment analysis using repeated measures 2-way ANOVA, but data for the same cannot be found in the manuscript.

Response:

Sorry for bringing misunderstanding information. Actually, in our study EEG were recorded continuously for 3 days and the average number per 24 hours were shown in figures. We have corrected the description 'Number of spikes/24h' and 'Number of SRS/24h' as 'Number of spikes per 24h' and 'Number of SRS per 24h'. And we described more clearly in the Methods section.

We have added graph of the power spectrum of EEG recordings in new Appendix Fig. S1D.

In statistics section, '*Regarding self-comparison before and after treatment of DETA/NONOate, repeated measures, Two-Way ANOVA was applied.*' was an explanation for new Appendix Fig. S4C to examine the acute antiseizure effect of DETA/NONOate.

7. The authors performed only Morris Water Maze and claimed that their approach restored pilocarpine-induced cognitive defects. Multiple behavioral experiments are needed to achieve this conclusion.

Response:

We have performed Y-maze and novel object recognition test, and the results suggested the same conclusion (new Fig. 6I-L). Thanks for the suggestion!

New Fig. 6I-L

8. The authors should improve their discussion section. First, the contradictory observation of the present study compared to the past has not been sufficiently addressed. For example, Yao et., 2022 have shown in Pilocarpine model nNOS level in hilus to increase at day 14 and back to basal levels by day 28 (<https://www.sciencedirect.com/science/article/pii/S0891584922004488?via%3Dihub>). Also the function of nNOS in neuronal excitability has been extensively studied, which should be discussed.

Response:

As suggested by you, we have improved the discussion section as following: "A series of studies have reported increased expression of nNOS and concentration of NO in the hippocampus in various models of mice epilepsy (Kovacs, Rabanus et al., 2009, Yasuda, Fujii et al., 2001, Zhu, Dong et al., 2017). However, all these experiments examined nNOS and NO in the whole hippocampus including the DG, CA1, CA2, CA3, SUB and other subregions. The DG region consists molecular layer, granule layer and hilus. The nNOS+ interneurons intensively locate in the hilus rather than CA1, CA2, CA3, SUB and other subregions. A reduced number of hilar interneurons is a hallmark of the pathology of TLE. Therefore, the measurement of nNOS in the DG and

throughout the hippocampus should be different and is predictable. Although a recent study found increased number of nNOS⁺ cells in the hilus 14 days after SE (Yao, Hu et al., 2022), but returned to basal levels by day 28, chronic changes were not examined.”.

Regarding the function of nNOS in neuronal excitability, we discussed in the introduction section as bellow: ‘nNOS is constitutively expressed in neurons, which mainly regulates synaptic plasticity, neuronal excitability, learning, memory, and neurogenesis in the central nervous system (CNS) (Wan, Xia et al., 2024), as well as blood pressure, smooth muscle relaxation and vasodilatation in the peripheral nervous system (PNS) (Forstermann & Sessa, 2012). Apparently, nNOS is the main synthetic enzyme of NO in the CNS involving in many pathophysiological brain processes.’.

Thank you for the suggestions!

9. There are several issues with acronym. First, acronym for DGCs was not provided when first used. What are SPKs? In Figure 2 all the graphs use the acronym TRE. However, in the legend only DRE is mentioned. ASD is extensively used for autism spectrum disorders. Most people use AED for anti-epileptic drugs. Please consider change that.

Response:

1) We feel sorry for missing complete descriptions and have revised the manuscript.

2) SPKs are defined as scattered and unrepetitive epileptic spikes namely high frequency sharp spikes, in comparison with SRS which are defined as continuous, repetitive and evolving spikes with interruption of the background activity for 10 seconds or longer. We have supplemented in the methods section.

3) In new Figure 1, we have corrected TRE to DRE. Thank you for pointing out this typo.

4) As you reminded, ASD was indeed misleading. We have used ASMs

instead. Antiseizure medications (ASMs)¹, previously referred to as anticonvulsant or antiepileptic drugs, have been widely recognized in recent years. Although there are more than thirty types of ASMs currently available in clinic, they only aim to suppress seizures without adverse events, but they do not affect the underlying predisposition to generate seizures. It means that currently used drugs are antiseizure rather than antiepileptic. Therefore, there is an urgent need to explore new targets for antiepileptic therapy. We believe NO donor could be developed as a new type of anti-epileptogenesis drugs. So that we think ASMs is more suitable for this paper.

Thank you for your careful reading.

10. Lastly, this manuscript, including figure legends and supplemental information, requires extensive proofreading. Numerous typos and odd sentences prevent a clear understanding of this study. Several examples are as below:

"Epilepsy is the second incidence rate neurological disorder in the world."

"There is a statistical correlation that once the hippocampus develops into sclerosis" (how can hippocampus "become" sclerosis)

"meanwhile, the content of hilar nNOS was decreased in pilocarpine-induced epileptic mice." (assuming the authors meant the "expression level" of nNOS.)

"Different from low intrinsic excitatory DGCs that powerful inhibited by hilar interneurons in physiological condition, ..." (unable to comprehend this sentence)

Also, the authors should use "transduce" instead of "infect" when using virus to deliver genes.

Response:

We apologize for the errors and typos that bring you confusion and uncomfortable reading experience. As suggested by you, we have checked and corrected the writing throughout the whole manuscript carefully.

Thank you for the suggestion.

Referee #2 (Comments on Novelty/Model System for Author):

Explanation for medium technical quality (although a large series of complex state-of-the-art techniques have been used):

All statistical analyses lack information. Please give p-values for the ANOVA (not only the post-tests), please also add p-values for the post-tests (e.g. in supplementary tables) such that the reader gets a better estimation of the results of statistical testing. Adding stars to the bar graphs is not sufficient for the evaluation of the data. Just as one example: Fig. 5 L, right part, it is hard to believe that there is no significant difference between Sham and LV-nNos-RNAi-GFP, the p-values would thus be relevant. In addition, please add the N of the experiments in the text or in the legends or at least in a supplementary table. Although the results for individual mice are shown, it is tedious to always count the dots to the N value.

Referee #2 (Remarks for Author):

The authors have done an excessive amount of different experiments, which is very impressive. Overall, these experiments fit well together and are logical steps building upon each other to investigate the entire story of the paper. The results from the rescue experiments are very impressive and promising for novel therapeutic options. However, the experiments sometimes need more description, as it sometimes does not become clear from the main text or the Figure legend, why and experiment has been performed the way it has been performed (e.g. if different time points are analyzed). Some details on how the experiments have been done are also missing in the methods. In addition, sometimes it feels as if in the large number of experiments and parallel controls, the precision in describing the individual experiment is lost. In addition, the measurements performed are different for the various controls, which also makes it difficult to follow and get the entire picture. It is very

tedious for the reader to get the role of the individual experiment and the main story line from time to time. A clearer separation of the main experiments, presented in the main Figures and the control experiments, presented in the supplementary Figures might help to guide the reader through the paper here.

1. The methods are lacking some information: What kind of recording system was used for the EEGs (incl. amplification, sampling rate etc.)? How often were the mice recorded during which periods? Were the analyses done for single days (the ones given in the Figure legends) or for more days? Furthermore, in many experiments, the authors counted cells that were stained with a certain antibody. Except for the information that every 12th section was counted, no further information is given in the methods or suppl. Methods on how counting was performed. Please explain how slices were selected, which overall septo-temporal extent was analyzed, how the areas of interest were determined and so on. This is needed for the reader to evaluate what has been really done.

Response:

We apologize for the missing information. As suggested, we have provided detailed information in the methods section as following:

1) Digitized EEG was reviewed using NeuroWork bench software (Nihon Kohden America Inc., Irvine, USA) at a sampling rate of 1 with a high-frequency filter of 70 Hz in synchronization.

2) Mice were recorded continuously for 3 days after pilocarpine injection or virus transduction of 1.5 months and/or 2.5 months.

3) All EEG were visually checked for the presence of seizures. Mice were recorded continuously for 3 days to create 36 files (2 h per file), 25% of which was sampled for manual quantification and multiplied to reflect a 24 h period. The diurnal or nocturnal files were equally chosen.

4) Brains were coronal (for immunostaining) or horizontal (for virus tracing) sectioned into 40- μ m thickness slices, which were all collected as long as

containing hippocampus.

5) Every twelfth section throughout the hippocampus was processed for nNOS or cFOS immunohistochemistry and manually counted. The sections added up to 8 per hippocampus. The overall septo-temporal extent was analyzed.

6) The areas of interest, including hilus, CA3, CA1, etc, had very distinctive features after DAPI staining and were identified by checking the brain atlas.

Thank you for suggestion.

2. All statistical analyses lack information. Please give p-values for the ANOVA (not only the post-tests), please also add p-values for the post-tests (e.g. in supplementary tables) such that the reader gets a better estimation of the results of statistical testing. Adding stars to the bar graphs is not sufficient for the evaluation of the data. Just as one example: Fig. 5 L, right part, it is hard to believe that there is no significant difference between Sham and LV-nNos-RNAi-GFP, the p-values would thus be relevant. In addition, please add the N of the experiments in the text or in the legends or at least in a supplementary table. Although the results for individual mice are shown, it is tedious to always count the dots to the N value.

Response:

We apologize for the incomplete description and missing information. As suggested, we have added all these details of the experiments in the legends or supplementary file named 'Supplemental statistics information', including statistical method, N, p-values, etc.

Thank you for the suggestion.

3. What is the reason of presenting Fig. 2, which shows the general change in nNOS in humans and the pilocarpine model (without any KO), after Fig. 1 which already determines more detailed mechanisms underlying this observation? Wouldn't the opposite sequence be more logical? It is also hard

for the reader to jump from different KO mice in Fig. 1 to WT in Fig. 2, back to KO in Fig. 3. In addition, then, the paper would start with the pilocarpine model and end with rescue in the pilocarpine model, which seems more logical to me.

Response:

We cannot agree more with your suggestion and have changed the sequence of Fig.1 and 2 (new Fig. 1 and Fig. 3). We also make a clearer separation of the main experiments that presented in the main Figures, and the control experiments that presented in the supplementary Figures (new Fig. 2, Fig. 3 and Fig. EV3).

Many thanks for your help.

4.Fig. EV1 and corresponding main text: Why is the dose used for EEG recordings in the chronic stage of epilepsy different (200 mg/kg) from the two doses used to analyze the severity of the status epilepticus (230 mg/kg and 100 mg/kg) (see Fig. EV1 and corresponding results text)?

Response:

When we investigated the sensitivity to SE in *Nos1^{-/-}* mice, the dosages were chosen through a series of tests.

1) The regular dose of pilocarpine (275 mg/kg, s.c.) for setting up chronic epilepsy model in WT mice resulted in a 100% mortality in *Nos1^{-/-}* mice within 1 h due to severe convulsion attacks. To compare the sensitivity to SE induction by pilocarpine, we reduced to a dose of 230 mg/kg which caused severe SE in 100% *Nos1^{-/-}* mice and about 20% in WT mice. Then we reduced to a lower dosage to 100 mg/kg which caused none SE in WT mice but still frequency SE in *Nos1^{-/-}* mice.

2) We found that the mortality of *Nos1^{-/-}* mice significantly decreased in the dose of 230 mg/kg within 24 h but was still very high within 2.5 months. Consequently, we reduced the dose to 200 mg/kg which can successfully induce acute SE and chronic SRS without severe mortality in *Nos1^{-/-}* mice.

Thanks!

5. Fig. EV1 and corresponding main text: The counting of epileptic spikes and seizures was performed on day 75, but day 45 is also shown in Fig. EV1G. Has the counting only be performed on day 75? What is the reason to choose that day? What is the reason to show day 45 in addition? Is there something different between those days, or is day 45 the first day when seizures could be observed? This does not become clear in the text. In addition, in the methods the authors describe that spikes have been manually quantified in a 3h window. Was this 3-h count multiplied, such that the number given is supposed to reflect a 24h period? In addition, was the time window chosen for analysis always during the same time of the day for all mice (diurnal or nocturnal changes could affect spike numbers)?

Response:

1) Regarding the day 45 time point:

To monitor the initiation and progression of epilepsy in *Nos1^{-/-}* mice after pilocarpine administration, we record EEG weekly and found epileptic spikes and SRS in the hippocampus rather than the cortex 1.5 months after pilocarpine administration (day 45). 2.5 months after pilocarpine administration, epileptic spikes and SRS were detected both in the hippocampus and the cortex. These results indicate that the hippocampus is the initiation focus during the development of epilepsy.

2) Regarding the protocol of EEG analyses:

We feel sorry for the missing details and inaccurate expression. Mice were recorded continuously for 3 days to create 36 files (2 h per file), 25% of which was sampled for manual quantification and multiplied to reflect a 24h period. In other words, we chose 3 per 12 files within 24 h for analyses. The diurnal or nocturnal files were equally chosen.

We have supplemented details in the methods section. Thanks for the questions.

6. Please add the quantification of spontaneous epileptic spikes in *Nos1*^{-/-} mice in Fig. EV1 E to make it comparable to the values of pilocarpine-injected mice.

Response:

As you suggested, we have added the quantification of epileptic spikes in *Nos1*^{-/-} mice in new Fig. 2C.

7. Was the lentivirus vector expressing small hairpin RNA (Fig. EV2) injected into the hilus of one or both hemispheres? See also comment above: How was the counting of nNOS-positive cells performed (which are of interest, how many sections, were both hippocampi counted or only one etc.)?

Response:

1) The lentivirus vector expressing small hairpin RNA (Fig. EV2) injected into the hilus of both hemispheres.

2) Every twelfth section throughout the DG was counted manually. The sections added up to 8 per one hemisphere. Both hippocampi were counted. The area of interest was hilus, which had very distinctive features after DAPI staining and were identified by checking the brain atlas.

8. Some concerns about the Western blots: In some of the Western blots, one band is visible for nNOS (e.g. Fig. EV2B), in others two bands are visible. What is the reason for that? In addition, in Fig. 1E, the background level of the

GAPDH bands is very different from the nNos bands. Is this really from the same Western blot? In addition, Fig. 1E shows the Western blot for nNOS for the AAV with CamKII promotor. The corresponding Western plot of the GAD 67 promotor is missing. Please add for completeness. In Fig. EV5C, which of the many band represents the Nitrotyrosine?

Response:

1) When using recycled antibody, a weak unexpected band may visible under the main band of nNOS. We have replaced all the results with newly prepared antibody. The original whole blots have uploaded.

2) The nNOS and GAPDH bands in Fig. 1E was from the same blot. As we cropped the blot according to molecular weight of these proteins and exposed separately by enhanced chemilumi-nescence (Thermo Fisher Scientific).

3) As you suggested, we have added the western blots for nNOS of the AAV with GAD67 promotor in new Fig. EV3G.

4) All the bands represented the nitrotyosine-modified proteins as a result of the interactions with reactive nitrogen species.

9. In the snRNAseq analysis (Figure 1) are very large group or 'other neurons' is found. As only the dentate gyrus was prepared and sequenced, what type of neuron should that be if not excitatory or inhibitory? Are the markers (CamKII and GAD) really good markers to distinguish neurons under epileptic conditions, given that in many epilepsy models granule cells of the DG upregulate GAD under epileptic conditions (see papers from G. Sperk and others)? Besides differential regulation of Nos1 (interestingly a downregulation in interneurons and an upregulation, which is not commented in the text in excitatory neurons) a series of other differentially regulated genes is shown in Fig. 1B. Are the genes shown in B the only ones with differential expression in excitatory or inhibitory neurons or why is this exactly this selection given? In addition, to keep consistency, shouldn't there be Nos1 instead of nNos when representing differentially expressed genes?

Response:

1) Considering the existence of nNOS both in excitatory DGCs and inhibitory GABAergic interneurons of DG, as well as to emphasize the different regulation between them during epilepsy, the DGCs were noted as “excitatory neurons” and GABAergic interneurons were noted as “inhibitory neurons”. The “other neurons” were actually other CamKII-negative and/or GAD-negative neurons. As you questioned, we have realized that this was inappropriate. We have newly distinguished more cell types with more specific markers.

2) We have used new markers (Prox1, CamK2a and Rbfox3) to distinguish granule cells^{2,3}.

3) The genes besides *Nos1* shown in Fig. 1B were top ones with differential expression in DGCs or GABAergic interneurons, which may have little importance and could be acquired from GEO database we uploaded. Therefore, we have not shown in revised new Fig. 3C.

4) We have corrected *Nos1* instead of *nNos* in the revised new Fig. 3C.

Thank you for helping us improve this data analysis.

Figure 3A-C

10. It is surprising, that a selective KO of *Nos1* in inhibitory cells of the hilus

leads to epileptic spikes and SRS at 2.5 months after the AAV injection without any pilocarpine injection (Fig. 1I), whereas the constitutive full KO of *Nos1* (*Nos1*^{-/-} mice) only induces epileptic spikes but no SRS. What is the hypothesis of the authors why the more selective approach has a more severe outcome? Compensatory processes in the constitutive KO or other mechanisms? This should be discussed.

Response:

As advised, we added discussion in the manuscript as bellow: 'This study discovered that selective deletion of hilar nNOS protein was sufficient and essential for the development of temporal lobe epileptogenesis. However, general knockout of *Nos1* only led to increased sensitivity to SE induction, possibly due to compensatory mechanisms during development or the effects of nonselective loss of nNOS in excitatory neurons.'

Thanks for your suggestion!

11.Fig 2G, 2J and corresponding main text: In the main text, the authors claim a reduction of *Nos1* mRNA or nNOS protein at 7d, followed by a maintenance at 14d and 2m. However, in the plot it seems very much as if there was a further decline instead of a maintenance. Please test for this and eventually change the text accordingly. For the two-way ANOVA, please give the variables, which were tested, the results of the ANOVA for the individual values and not only the results of the post-test in form of stars in the image. In addition, please describe what was tested with the post-test, e.g. for Fig. 2G and J, just naming the two-way ANOVA is not sufficient.

Response:

As you suggested, we have added the RT-qPCR and western blot for nNOS 4 months post SE in the new Fig. 1G and Fig.1J.

We have added details information in the supplementary file named 'Supplemental statistics information', including statistical method, N, p-values,

etc. Thank you for the suggestion.

12. In the rabies tracing experiments (Fig. 3A, B), I am very surprised that there is no contralateral labeling. Shouldn't contralateral mossy cells project onto ipsilateral granule cells (see different studies of H. Scharfman). Why is there no labeling visible in the septum/DBB? The authors analyze forebrain regions and show a significant difference in the quantification, but I cannot see that in any of that images. How is the number of input cells in distant areas normalized to the number of infected cells in the granule cell layer? Please add separate enlarged view of the dentate gyrus for WT and *Nos1*^{-/-} mice with green and red channel separated. It seems as if the rate of infection is different in the different genotypes. For quantifying cells in the brainstem - which areas of the brainstem were selected for the counting? Similar for the forebrain - both are large areas, how were the areas of interest for counting selected?

Response:

Due to a small number of traced cells in the contralateral hippocampus, we did not show them in the previous images. We have added new images to show traced cells in the contralateral hippocampus (new Fig. EV4). Medial septum (MS) and diagonal band (DB) are well-known regions projecting to DGCs. Considering the space limitation, we did not show representative images. As suggested we have added representative images in new Fig. EV5. We have also added separate enlarged view of the DG for WT and *Nos1*^{-/-} mice with green and red channel separated in new Fig 4B.

We counted all the traced cells throughout the brain. The fluorescence signal of traced cells was only visible in ventral tegmental area (VTA) and dorsal raphe (DR), which have been described in the manuscript as 'nNOS deficiency caused excessive connectivity with neurons majorly located in the entorhinal cortex including the lateral and medial entorhinal area (LEA and MEA), forebrain including the medial septum (MS) and diagonal band (DB), brainstem including ventral tegmental area (VTA) and dorsal raphe (DR), as well as CA3'. Similarly, we quantified all the traced cells in the forebrain and

only found traced cells in medial septum (MS) and diagonal band (DB).

Regarding the method detail, we have added more description in the method section. Thanks for the advices!

New Figure EV4

13. How do the authors explain the strong effects after cPTIO. I can imagine substantial differences between WT and *Nos1*^{-/-} mice, which might develop during development. But the strong differences after a chemical (not-hippocampus-specific, not 100%) NO blockade inducing such massive sprouting from various distant regions is hard to believe. Why did they choose this chemical approach and not their more selective knockdown approaches, which have been used earlier in the paper? Why don't they show the fiber sprouting effects in pilocarpine mice in more detail also here, as these are

relevant for the rescue experiments presented later.

14. In Fig. EV6 A and the corresponding main text, the authors apply the NO blocker c-PTIO. The Figure then uses the terms Control and Seizure. However, the existence of seizures after the application of c-PTIO has not been described before. Either the analysis of seizures after this application is necessary or the Figure has to be labelled differently. Similar for EV6B: Does the application of c-PTIO lead to a status epilepticus as the Figure implies? This remains unclear.

Response:

The question 13 and 14 are regarding the c-PTIO experiment. We merged our explanations here.

The aim of this experiment was to investigate whether deprivation of NO by c-PTIO in normal mice had similar effects with deficiency of NO in nNOS knockout mice on the development of excitatory inputs onto DGCs. Although both c-PTIO administration and nNOS knockout had general effects on the body, nNOS protein also exerts biological effects beyond NO. Hence, we chose c-PTIO rather than knock down strategy to clarify the role of NO in the development of excitatory inputs onto DGCs.

Regarding the strong effects after cPTIO, the long-term of administration (2.5 months) is a potential reason. The results of the c-PTIO experiment indicated that appropriate amount of NO is important for biological function of the body and for maintaining normal excitatory inputs onto DGCs. Although deficiency of NO induced by cPTIO caused epileptic-like abnormalities of DGCs circuit, the application of c-PTIO did not induce status epilepticus, possibly due to the effects of cPTIO were not as strong as knockout (new Appendix Fig. S2). We also have corrected the labels. Thanks very much!

15. Fig. 4: The massive upregulation of cFOS in granule cells after injection of the AAV9-pGAD67 virus is surprising compared to the constitutive KO. Why is this the case?

Response:

As you concluded in the comment above, the selective KO of *Nos1* in hilar inhibitory interneurons led to epileptic spikes and SRS at 2.5 months after the AAV injection without any pilocarpine injection, whereas the constitutive full KO of *Nos1* (*Nos1*^{-/-} mice) only induced epileptic spikes but no SRS. The representative photos of cFOS staining were collected from *Nos1*^{-/-} mice with epileptic spikes (new Fig. 4A) or *Nos1*^{loxp/loxp} mice injected AAV9-pGAD67 virus with severe SRS (new Fig. 4H). Thanks!

16. The language needs some serious revision by a native speaker or someone sufficiently proficient in English. There are many grammatical errors which make the paper at times hard to read and understand (e.g. lack of articles, lack of prepositions, wrong verb forms, use of plural at the wrong sites and lack of a required plural at other sites and so on). Please correct these throughout the paper, including the Figure legends. In addition, there are many typos, also in the Figures (e.g. Pliocarpine in Fig. 6), some axis labels are misplaced and so on. Overall, the paper needs some serious checking and correcting.

Response:

We apologize for the grammatical errors and typos that bring you confusion and uncomfortable reading experience. As suggested, we have checked and improved the writing throughout the whole manuscript carefully.

Thank you for the suggestion.

Some minor comments:

1. Different isoforms of Nos (nNOS, iNOS, eNOS) are named in the paper. Please add information on these isoforms to the introduction as not all readers are familiar with the roles and specificities of the different isoforms.

Response:

As you suggested, we have added information on three isoforms in the

introduction section.

2.How long after death were the post-mortem samples collected, that were used for the analysis of human tissue?

Response:

The post-mortem samples were collected from organ donors within 24 h of death, and we have added this in the methods section. Thanks!

3.Fig. EV4B: How was the cortical delay analyzed? What EEG signature was taken to detect the activity in the cortex? Are the values given the mean delays for all seizures in each mouse? How consistent or variable are the onset delays within individual mice?

Response:

The cortical delay was analyzed as the time interval between the first spike occurred in DG and the first spike occurred in cortex within every SRS.

The epileptiform/interictal spikes were defined as paroxysmal electrical sharp activity lasting 20-150 ms, with an amplitude that was at least five times the background EEG activity.

The values were given the mean delays for all SRS that DG seizure onset prior to cortex seizure in each mouse (n=5). Thanks!

4.The methods mention CNO infusion in slices. It is unclear to me for which of the patch-clamp experiments this was relevant.

Response:

We apologize for mentioning CNO mistakenly and have deleted the sentence. Thanks.

5.The text mentions a Fig. 4C when describing the tracing - that should probably be Fig. 3C.

Response:

We feel really sorry for this mistake and have corrected in revised manuscript. Thank you for suggestion.

6.Introduction: 'Besides, multiple histopathological changes, such as neurons loss, synaptic reorganization, gliosis, and altered neurogenesis in the dentate gyrus (DG) of the hippocampus, also contribute to the development of hippocampal sclerosis.' Hippocampal sclerosis is defined as cell loss and gliosis (see several papers e.g. of I. Blümcke, M.C. Walker and others). Other histological patterns of TLE like neurogenesis, mossy fiber sprouting etc. come in conjunction with hippocampal sclerosis but are not part of it. The sentence is thus misleading.

Response:

Thanks for the precise advice! We have corrected the sentences as 'Cell loss in the hippocampal hilus, especially the death of interneurons and mossy cells, and gliosis contribute to this pathological process (Rusina, Bernard et al., 2021). Besides, multiple histopathological changes, such as synaptic reorganization, altered neurogenesis in the dentate gyrus (DG) of the hippocampus and mossy fiber sprouting, may also contribute to the development of hippocampal sclerosis (Casillas-Espinosa, Powell et al., 2012, Polli, Malheiros et al., 2014, Rusina et al., 2021).'

7.Introduction: However, it remains unclear whether nNOS+ interneurons will be damaged or preserved in the epileptic hippocampus and how important role of nNOS plays in the pathology of TLE. Something is wrong in the highlighted part.

Response:

We have corrected the sentence as 'However, the role of nNOS⁺ interneurons in the pathology of TLE remains unclear.' Thanks for the suggestion!

8.Methods: Animals that received pilocarpine and did not develop convulsive seizures higher than stage 4 were excluded. Do the authors mean 'equal and higher than' or do they really mean only 'higher than', i.e. only mice that reach stage 5 were used for analysis?

Response:

We mean equal and higher than, i.e. both stage 4 and 5 were used. We have corrected in the methods section.

9.Results, 1st section: ...driven by promoter CaMKII α (AAV9-pCaMKII α -Cre) (Fig. 1C) or GABA (AAV9-pGAD67-GFP-2A-Cre) (Fig. 1F) into the bilateral DGs of WT...The promotor seems to be GAD67, not GABA, please change. In addition, what is the bilateral DG? This is either a bilateral injection into the DG or an injection into both DGs but there is no such thing as a bilateral DG.

Response:

We have changed GABA to GAD67. We injected virus into both DGs. Thank you for suggestion.

10.What is shown in Fig. EV3B? What does P1P2P3 stand for? The legend is insufficient.

Response:

Fig. EV3B showed representative genotyping gel of *Nos1*^{loxp/loxp} mice using three pairs of primers.

11.Fig. EV4B: prior instead of priory.

Response:

We have corrected this typo. Thanks for the suggestion!

12.Fig. 2B: Please mention the role of the red circles in the scheme and the red squares in the EEG. Why are these chosen?

Response:

In new Fig. 1B, interictal spike-wave discharges was recorded by the ECoG using the International 10-20 EEG system in the left temporal lobe. The F7, T3, and T5 electrodes captured EEG discharges in the left anterior temporal lobe, middle temporal lobe, posterior temporal lobe, and adjacent brain regions, respectively. We have added this in the legend. Thanks for the suggestion!

13.Fig. 2C: The squared areas are hard to see. Please also explain in the legend why these areas are highlighted.

Response:

In new Fig. 1C, position of deep brain electrodes and their discharges on the brain surface in SEEG monitoring. 'Electrode c' ran from the anterior part of the left middle temporal gyrus to the head of the hippocampus, 'electrode d' from the middle part of the left middle temporal gyrus to the hippocampus body, and 'electrode e' from the posterior part of the left middle temporal gyrus to the hippocampus tail. The topmost channel of each electrode represented the innermost contact, and the bottommost channel represented the outermost contact. Thus, the topmost channels of c, d, and e reflected the head, body, and tail of the hippocampus, respectively, showing discharges from the left hippocampus. We have added this in the legend. Thanks for the suggestion!

Referee #3 (Comments on Novelty/Model System for Author):

The authors investigated the role of nitric oxide (NO) signaling from nNOS+ inhibitory interneurons of the dentate gyrus in contributing to temporal lobe epilepsy (TLE). In a previous study, they show in a pilocarpine mouse model that induction of TLE leads to aberrant innervation of dentate granule cells by entorhinal cortex. Here, using conditional knockout of Nos1 from interneurons and dentate granule cells, rescue of Nos1, cFos staining, in vivo and in vitro electrophysiology, and monosynaptic retrograde rabies tracing, they show that loss of NO release from nNOS+ interneurons causes this aberrant circuit formation and development of TLE. This finding is novel, the experiments are thorough, and the data are convincing. Furthermore, identifying cell-type-specific mechanisms is crucial for developing targeted therapeutic approaches.

Referee #3 (Remarks for Author):

Addressing the following would improve the manuscript:

1) In Figure EV2A, viral-induced GFP and IHC-RFP for nNOS do not overlap. Thus, in this example, it appears the lentivirus has not infected nNOS+ cells.

Response:

We have replaced with high quality representative images and added arrows and arrowheads in new Fig. EV2A to indicate infected hilar nNOS-positive and -negative interneurons, respectively. Thanks for the advice!

2) The methods section description of whole-cell patch clamp electrophysiology is both incomplete and inaccurate. The authors state they performed "cell attached" recording, but that their data in Figure 4 are whole-cell recordings in voltage clamp. This section needs to be updated with proper method details. Furthermore, what were the ages of the mice used for these experiments?

Response:

We apologize for the incomplete and inaccurate description. The mice were at the age of 8-10 weeks. As you suggested, we have added more detailed information in the methods section.

Thanks very much.

3) If DG cells receive more excitatory drive, which contributes to seizures, how do the authors explain the lack of increased cFos staining in CA3? Wouldn't you expect excessive drive from mossy fiber inputs to cause increased activity

in CA3?

Response:

We observed no significant increase in cFOS⁺ cells in the CA3 region in nNOS general knockout mice possibly due to nonselective loss of nNOS in CA3 excitatory neurons, in line with no spontaneous SRS but just more sensitive to pilocarpine-induced SE phenotype were detected in these mice. Indeed, 2.5 months after selective knockout of nNOS in the hilar interneurons, cFOS⁺ cells both in the DG and CA3 remarkably increased (new Fig. 5H). Thanks for the good question!

H 2.5 months after AAV9-pGAD67-GFP-2A-Cre injection

4) Do the authors have any data on whether loss of NO signaling alters the intrinsic membrane properties of dentate granule cells?

Response:

As you suggested, we have shown data on intrinsic membrane properties of DGCs in new Fig. 5G. Thank you for suggestion.

5) N's (mice, cells, etc.) are missing throughout.

Response:

We apologize for the incomplete description and missing information. As

suggested by you, we have added all the details of the experiments in the legends or supplementary file named 'Supplemental statistics information', including statistical method, N, p-values, etc. Thank you for the suggestion.

6) On page 8, "Fig. 3C" is mislabeled as "Fig. 4C".

Response:

We feel really sorry for this mistake and have corrected in revised manuscript. Thank you for suggestion.

7) This manuscript needs editing for grammar.

Response:

As suggested, we have checked and corrected the writing throughout the whole manuscript carefully.

Thank you for the suggestion.

1. Asadi-Pooya AA, Brigo F, Lattanzi S, Blumcke I. Adult epilepsy. *Lancet*. Jul 29 2023;402(10399):412-424. doi:10.1016/S0140-6736(23)01048-6
2. Chen ZP, Wang S, Zhao X, *et al*. Lipid-accumulated reactive astrocytes promote disease progression in epilepsy. *Nat Neurosci*. Apr 2023;26(4):542-554. doi:10.1038/s41593-023-01288-6
3. Hochgerner H, Zeisel A, Lonnerberg P, Linnarsson S. Conserved properties of dentate gyrus neurogenesis across postnatal development revealed by single-cell RNA sequencing. *Nat Neurosci*. Feb 2018;21(2):290-299. doi:10.1038/s41593-017-0056-2

1st Oct 2024

Dear Dr. Meng,

Thank you for the submission of your revised manuscript to EMBO Molecular Medicine, and please accept my apologies for the delay in getting back to you as one referee needed more time to complete the review.

As you will see from the reports below, while referee #1 is satisfied with the revisions, referees #2 and #3 still have concerns on the study, mostly related to language, statistics, and general presentation of the manuscript. Given that the issues raised by the referees do not require performing new experiments, we would like to invite further revisions of the study. However, as EMBO Press usually encourages one single round of revisions, please be aware that this will be the last chance for you to address these issues. All concerns regarding text, figures and statistics will have to be satisfactorily addressed. The revised manuscript will once again be subjected to review, and we cannot guarantee a positive outcome at this stage.

Moreover, please address the following editorial requests:

1/ Manuscript text:

- Please remove the yellow highlights and only keep in track changes mode any new modification.
- Please correct the order of the manuscript sections as follows: Abstract / Introduction / Results / Discussion / Materials and Methods / Acknowledgements / Disclosure and competing interests statement / The Paper Explained / For More Information / References / Figure legends / Tables and their legends / Expanded View Figure legends.
- Data Availability: please make sure all primary datasets produced in this study are deposited in an appropriate public database, and the accession numbers and database listed under 'Data Availability'. These datasets must be public before acceptance of the manuscript.

2/ Methods: All Materials and Methods need to be described in the main text using our 'Structured Methods' format. According to this format, the Methods section includes a Reagents and Tools Table (listing key reagents, experimental models, software and relevant equipment and including their sources and relevant identifiers) followed by a Methods and Protocols section describing the methods, ideally using a step-by-step protocol format. The aim is to facilitate adoption of the methodologies across labs.

Please download and fill our Reagents and Tools Table template (.docx), which you can find in our author guidelines:

The supplementary methods should be merged with the main methods in the manuscript text.

3/ Figures and Appendix:

- Kindly ensure that all figures and figure panels are referenced in the text. Currently, callouts are missing for Fig. 5H and Fig. 7H.
- Figure re-use or partial re-use should be indicated in the figure legends (i.e. figure 4A/B, figure 6M/EV3F) and the method should be detailed.
- Appendix: files with figures and their legends should be merged into one PDF labelled "Appendix". The file needs a table of contents with page numbers. Ideally, appendix figure legends should be placed under each corresponding figure.
- Please address the queries from our data editors:
 1. Please note that the exact p values are not provided in the legends of figures 1f-g, j-k; 2a-c; 4f-g; 5c-d, f; 7f-g; EV 1e; EV 2a-b; EV 3e, g; EV 5d, g.
 2. Please indicate the statistical test used for data analysis in the legend of figure 7h.
 3. Please note that the white arrowheads are not defined in the legend of figure EV 3f. This needs to be rectified.

4/ Thank you for providing detailed statistics. Please include this information either in the main manuscript file (methods or figure legends), or in the Appendix as Appendix tables.

5/ Please clarify what is the "Supplementary approved ethical file".

6/ Thank you for providing Source Data. Please re-order and upload them as one (zipped) file per figure.

7/ Please note that all corresponding authors are required to supply an ORCID ID for their name upon submission of a revised

manuscript. Currently, an ORCID ID is missing for Bao-Sheng Huang.

8/ I included modifications in your Paper Explained, please let me know if you agree with the following or amend as you see fit:
Problem

Most antiseizure drugs developed for epilepsy treatment efficiently suppress status epilepticus, but do not impact the development of epileptogenesis, and can trigger drug resistance.

Results

We observed decreased nNOS expression in the hippocampus of patients with temporal lobe epilepsy (TLE). In transgenic mice, selective deletion of nNOS from hilar GABAergic interneurons induced epileptogenesis. Physiological levels of NO maintained the normal afferent circuit of DGCS, whereas chronic NO deficiency caused hyper-excitatory afferent circuit integration and hyperexcitability of DGCS. Replenishment of nNOS blocked the development of epileptogenesis and restored memory deficits in epileptic mice. Chronic NO donor treatment was sufficient to prevent the development of epileptogenesis.

Impact

Our findings reveal that NO donor prevents progression to TLE, highlighting a novel therapeutic strategy unrelated to neuronal ion channel blockers.

9/ introduced minor modifications in your synopsis, please let us know if you agree or amend as you see fit:

"Chronic nitric oxide (NO) donor treatment is a novel antiepileptic strategy that has an effect beyond neuronal ion channels.

- The level of nNOS declines in the hippocampus of patients with epilepsy.
- Selective deletion of nNOS in interneurons of hilus causes epileptogenesis.
- Replenishment of nNOS in the DG blocks epileptogenesis and memory deficits.
- Chronic NO donor treatment prevents epileptogenic development.

Thank you for providing a nice synopsis image. I have cropped a small portion to illustrate your article in our table of content (attached). Please let us know if you approve, or kindly suggest an alternative (115px x 70px). Changes at proof stage are usually not allowed.

10/ As part of the EMBO Publications transparent editorial process initiative (see our Editorial at <http://embomolmed.embopress.org/content/2/9/329>), EMBO Molecular Medicine will publish online a Review Process File (RPF) to accompany accepted manuscripts.

In the event of acceptance, this file will be published in conjunction with your paper and will include the anonymous referee reports, your point-by-point response and all pertinent correspondence relating to the manuscript. Let us know whether you agree with the publication of the RPF and as here, if you want to remove or not any figures from it prior to publication. Please note that the Authors checklist will be published at the end of the RPF.

I look forward to receiving your revised manuscript.

Yours sincerely,

Lise Roth

***** Reviewer's comments *****

Referee #1 (Comments on Novelty/Model System for Author):

The role of nNOS in epilepsy has been studied in the past. However, this current study further elaborated it as a therapeutic target, which is novel.

Referee #1 (Remarks for Author):

The authors have addressed my comments. I have no further questions and can support publication.

Referee #2 (Remarks for Author):

The authors changed a lot of things in their text according to the reviewer's comments and the manuscript has improved. However, there still are a couple of issues and it feels like the authors have slightly lost overview on this large dataset (which I fear the readers will also do).

1. According to my comment, the authors added the following statement: 'at a sampling rate of 1 with a high-frequency filter of 70 Hz in synchronization'. What do they mean by a sampling rate of 1? This requires a dimension!

2. In Figures 2C, D, E and 7E there are only scale bars for the time (x-axis), but the ones for the y-axis are missing.

3. Fig. 5H does not appear in the main text. And what is described in the text as Fig. 5G actually matches Fig. 5H, whereas what is now shown as Fig. 5G is not explained (the corresponding Figure legend is ok).

4. EEGs in Fig. 6D lack all scalebars.

5. The legends of the Figure contain p values for up to three stars, whereas the images often show 4 stars.

6. In Figure 7H a couple of p values are floating around, these do not seem to belong there.

7. Some improvements to the text have been made, still, the text requires more serious proof readings to avoid some rather strange statements.

8. The terminology used for the supplementary Figures should be the same for every supplementary Figure.

Referee #3 (Remarks for Author):

I appreciate that the authors added a new panel G to Figure 5 with additional data to address my feedback. However, they did not update the revised text to describe these new data. The new data in Figure 5G and discussion of their significance are missing.

The authors now include n's for number of mice, but do not specify how many are controls and how many are experimental. They simply state, for example, "n = 3 mice". In some places they state "n = 6 - 7 mice". Why a vague range? I appreciate that they provided tables for their statistics, but the n's for mice are not clear in those files either. For example, the table for Figure 5 provides the n's for number of cells, but it remains unclear how many mice were used to collect them.

The manuscript still needs major editing by a native English speaker.

We sincerely appreciate the reviewer's constructive suggestions and corrections. We have addressed each of these comments below, with highlighted new text that has been added to the manuscript and supplementary documents.

Referee #1 (Comments on Novelty/Model System for Author):

The role of nNOS in epilepsy has been studied in the past. However, this current study further elaborated it as a therapeutic target, which is novel.

Referee #1 (Remarks for Author):

The authors have addressed my comments. I have no further questions and can support publication.

Response:

Thank you for helping us improve our manuscript.

Referee #2 (Remarks for Author):

The authors changed a lot of things in their text according to the reviewer's comments and the manuscript has improved. However, there still are a couple of issues and it feels like the authors have slightly lost overview on this large dataset (which I fear the readers will also do).

1. According to my comment, the authors added the following statement: 'at a sampling rate of 1 with a high-frequency filter of 70 Hz in synchronization'. What do they mean by a sampling rate of 1? This requires a dimension.

Response:

Sorry for the missing information. The sampling rate was 1 kHz. We have added in the methods section.

Thanks for the comment.

2. In Figures 2C, D, E and 7E there are only scale bars for the time (x-axis), but the ones for the y-axis are missing.

Response:

We are sorry for the missing scale bars for the y-axis. As suggested by you, we have added all scale bars in Figures 2CDE, 3IJ, 6D, 7E, EV2D, EV5CF, Appendix Figure S3D, S4BD.

Thanks for the suggestion.

3. Fig. 5H does not appear in the main text. And what is described in the text as Fig. 5G actually matches Fig. 5H, whereas what is now shown as Fig. 5G is not explained (the corresponding Figure legend is ok).

Response:

We appreciate you for raising this mistake. We have added description of Fig. 5G and discussion of their significance in the manuscript as following:

“More importantly, we found that nNOS deficiency had no significant effects on the intrinsic membrane properties of DGCs, including input resistance, membrane potential, minimal current and spike numbers (Fig. 5G). It verified indirectly that nNOS deficiency induced hyperexcitation of DGCs through hyperexcitatory afferent inputs to DGCs other than intrinsic changes in DGCs. Similarly, a remarkable elevation was observed in the number of cFOS+ DGCs at 2.5 months after microinjection of AAV9-pGAD67-Cre into the hilus of *Nos1^{loxP/loxP}* mice, compared with WT mice (Fig. 5H).”

Thanks for the suggestion.

4. EEGs in Fig. 6D lack all scalebars.

Response:

As we responded in comment 2, we have added all scale bars of EEGs. Thanks.

5. The legends of the Figure contain p values for up to three stars, whereas the images often show 4 stars.

Response:

We have added '**** $P < 0.0001$ ' in the legends. Thanks.

6. In Figure 7H a couple of p values are floating around, these do not seem to belong there.

Response:

As you reminded, we have deleted these 'p values'. Thanks for your careful reading.

7. Some improvements to the text have been made, still, the text requires more serious proof readings to avoid some rather strange statements.

Response:

We have invited a native English speaker for thoroughly proof readings.

Thanks for the suggestion.

8. The terminology used for the supplementary Figures should be the same for every supplementary Figure.

Response:

The journal replaces Supplementary Figures with Expanded View (EV) Figures that are collapsible/expandable online. A maximum of 5 EV Figures can be typeset. EV Figures should be cited as 'Figure EV1, Figure EV2' etc. For the figures that we do NOT wish to display as Expanded View figures, they should be bundled together with their legends in a single PDF file called *Appendix*. Appendix figures should be referred to in the main text as: "Appendix Figure S1, Appendix Figure S2" etc. As requested by the editor, we displayed 5 EV Figures and 4 Appendix Figures.

We have corrected the 'cPTIO' to 'c-PTIO' in Appendix Figure S2. Thanks.

Referee #3 (Remarks for Author):

I appreciate that the authors added a new panel G to Figure 5 with additional data to address my feedback. However, they did not update the revised text to

describe these new data. The new data in Figure 5G and discussion of their significance are missing.

Response:

We appreciate you for raising this mistake. We have added description of Fig. 5G and discussion of their significance in the manuscript as following:

“More importantly, we found that nNOS deficiency had no significant effects on the intrinsic membrane properties of DGCs, including input resistance, membrane potential, minimal current and spike numbers (Fig. 5G). It verified indirectly that nNOS deficiency induced hyperexcitation of DGCs through hyperexcitatory afferent inputs to DGCs other than intrinsic changes in DGCs. Similarly, a remarkable elevation was observed in the number of cFOS+ DGCs at 2.5 months after microinjection of AAV9-pGAD67-Cre into the hilus of *Nos1^{loxp/loxp}* mice, compared with WT mice (Fig. 5H).”

Thanks for the suggestion.

The authors now include n's for number of mice, but do not specify how many are controls and how many are experimental. They simply state, for example, "n = 3 mice". In some places they state "n = 6 - 7 mice". Why a vague range? I appreciate that they provided tables for their statistics, but the n's for mice are not clear in those files either. For example, the table for Figure 5 provides the n's for number of cells, but it remains unclear how many mice were used to collect them.

Response:

Sorry for bringing you confusions about the n's. “n = 6-7 mice”, for example in Fig. 4F, meant the number of WT mice was 7 and that of *Nos1^{loxp/loxp}* mice was 6. We also added more description about n's in figure legends and statistics tables merged in “Appendix” file for Fig.5 as, for example in Fig. 5C, “n = 12 cells from 3 mice”.

As you reminded, we have added the n's in the figures. Thanks for the suggestion.

The manuscript still needs major editing by a native English speaker.

Response:

We have invited a native English speaker for thoroughly proof readings.

Thanks for the suggestion.

1/ Manuscript text:

- Please remove the yellow highlights and only keep in track changes mode any new modification.
- Please correct the order of the manuscript sections as follows: Abstract / Introduction / Results / Discussion / Materials and Methods / Acknowledgements / Disclosure and competing interests statement / The Paper Explained / For More Information / References / Figure legends / Tables and their legends / Expanded View Figure legends.
- Data Availability: please make sure all primary datasets produced in this study are deposited in an appropriate public database, and the accession numbers and database listed under 'Data Availability'. These datasets must be public before acceptance of the manuscript.

We have made correction and uploaded.

2/ Methods:

All Materials and Methods need to be described in the main text using our 'Structured Methods' format. According to this format, the Methods section includes a Reagents and Tools Table (listing key reagents, experimental models, software and relevant equipment and including their sources and relevant identifiers) followed by a Methods and Protocols section describing the methods, ideally using a step-by-step protocol format. The aim is to facilitate adoption of the methodologies across labs.

Please download and fill our Reagents and Tools Table template (.docx), which you can find in our author guidelines: <https://www.embopress.org/page/journal/14693178/authorguide#structuredmethods>.

An example of a Method paper with Structured Methods can be found here: <https://www.embopress.org/doi/10.15252/msb.20178071>.

We have added "Reagent Table" and uploaded.

The supplementary methods should be merged with the main methods in the manuscript text.

We have merged and uploaded.

3/ Figures and Appendix:

- Kindly ensure that all figures and figure panels are referenced in the text. Currently, callouts are missing for Fig. 5H and Fig. 7H.

We have made correction and uploaded.

- Figure re-use or partial re-use should be indicated in the figure legends (i.e. figure 4A/B, figure 6M/EV3F) and the method should be detailed.

Fig. 6M was not re-used. Fig.4AB, Fig. EV4AB and Appendix Fig. S1A were partial re-used. These figures only included zoomed pictures. We have indicated in the figure legends.

- Appendix: files with figures and their legends should be merged into one PDF labelled "Appendix". The file needs a table of contents with page numbers. Ideally, appendix figure legends should be placed under each corresponding figure.

We have merged into "Appendix" file and uploaded.

- Please address the queries from our data editors:

1. Please note that the exact p values are not provided in the legends of figures 1f-g, j-k; 2a-c; 4f-g; 5c-d, f; 7f-g; EV 1e; EV 2a-b; EV 3e, g; EV 5d, g.

We have displayed the exact p values in the figures.

2. Please indicate the statistical test used for data analysis in the legend of figure 7h.

The Fig. 7H was just a model graph without statistical test.

3. Please note that the white arrowheads are not defined in the legend of figure EV 3f. This needs to be rectified.

We have rectified in the legend.

4/ Thank you for providing detailed statistics. Please include this information either in the main manuscript file (methods or figure legends), or in the Appendix as Appendix tables.

We have merged the statistics file in the Appendix.

5/ Please clarify what is the "Supplementary approved ethical file".

The "Supplementary approved ethical file" was an ethical certificate for using DRE patient subjects approved by Sir Run Run Hospital, Nanjing Medical University.

6/ Thank you for providing Source Data. Please re-order and upload them as one (zipped) file per figure.

We have uploaded.

7/ Please note that all corresponding authors are required to supply an ORCID ID for their name upon submission of a revised manuscript. Currently, an ORCID ID is missing for Bao-Sheng Huang.

The ORCID for Bao-Sheng Huang is 0009-0004-6865-2168.

8/ I included modifications in your Paper Explained, please let me know if you agree with the following or amend as you see fit:

Problem

Most antiseizure drugs developed for epilepsy treatment efficiently suppress status epilepticus, but do not impact the development of epileptogenesis, and can trigger drug resistance.

Results

We observed decreased nNOS expression in the hippocampus of patients with temporal lobe epilepsy (TLE). In transgenic mice, selective deletion of nNOS from hilar GABAergic interneurons induced epileptogenesis. Physiological levels of NO maintained the normal afferent circuit of **dentate granular cells (DGCs)**, whereas chronic NO deficiency caused hyper-excitatory afferent circuit integration and hyperexcitability of DGCs. Replenishment of nNOS blocked the development of epileptogenesis and restored memory deficits in epileptic mice. Chronic NO donor treatment was sufficient to prevent the development of epileptogenesis.

Impact

Our findings reveal that NO donor prevents progression to TLE, highlighting a novel therapeutic strategy unrelated to neuronal ion channel blockers.

We appreciate you for modifying our “The Paper Explained”. We agree with your modifications with minor supplementary words as shown in blue.

9/ introduced minor modifications in your synopsis, please let us know if you agree or amend as you see fit:

"Chronic nitric oxide (NO) donor treatment is a novel antiepileptic strategy that has an effect beyond neuronal ion channels.

- The level of nNOS declines in the hippocampus of patients with epilepsy.
- Selective deletion of nNOS in interneurons of hilus causes epileptogenesis.
- Replenishment of nNOS in the DG blocks epileptogenesis and memory deficits.
- Chronic NO donor treatment prevents epileptogenic development.

Thanks for your kindness. We agree with your modifications

Thank you for providing a nice synopsis image. I have cropped a small portion to illustrate your article in our table of content (attached). Please let us know if you approve, or kindly suggest an alternative (115px x 70px). Changes at proof stage are usually not allowed.

We have made an alternative image and uploaded.

10/ As part of the EMBO Publications transparent editorial process initiative (see our Editorial at <http://embomolmed.embopress.org/content/2/9/329>), EMBO Molecular Medicine will publish online a Review Process File (RPF) to accompany accepted manuscripts.

In the event of acceptance, this file will be published in conjunction with your paper and will include the anonymous referee reports, your point-by-point response and all pertinent correspondence relating to the manuscript. Let us know whether you agree with the publication of the RPF and as here, if you want to remove or not any figures from it prior to publication.

We agree with the publication of the RPF.

17th Oct 2024

Dear Dr. Meng,

Thank you for submitting your revised files. Referee #2 has reviewed your manuscript, and as you will see below, is satisfied with the revisions. I will thus be able to accept your manuscript once the following minor editorial concerns will be addressed:

Manuscript text:

- Please remove the yellow highlights and only keep in track changes mode any new modification.
- Please add the title page with author names and affiliations (i.e. unblind the manuscript).

Methods:

- Please include the full statement confirming that informed consent was obtained from all subjects and that the experiments conformed to the principles set out in the WMA Declaration of Helsinki and the Department of Health and Human Services Belmont Report.
- Please note that we cannot use the ethical approval document you provided. Kindly state in the methods section details of authority granting ethics approval (IRB or equivalent committee(s), provide reference number for approval. Please update the checklist accordingly.

Data availability:

- Please check the link provided, which currently links to another unrelated study.
- Please note that the data must be publicly available before acceptance of the manuscript.

Figure legends:

- Please remove "Source data are available online for this figure".

Figure re-use:

- If the same pictures are used in figure 4A and 4B, please indicate it in the figure legends.
- If the same picture is used in figure 6M and figure EV3F, please indicate it in the figure legends.

Source Data:

- SD Fig. 1G: please check the raw data provided (identical values found for control 7d/14d).
- SD Fig. 5C and 5D: please check the raw data provided (identical values in columns WT/Nos1-/-)

I look forward to receiving your revised manuscript.

Yours sincerely,

Lise Roth

To submit your manuscript, please follow this link:
<https://embomolmed.msubmit.net/cgi-bin/main.plex>

***** Reviewer's comments *****

Remarks for Author:

Is suitable for publication

Dear Dr. Roth,

Thank you for carefully checking our raw data to avoid this minor mistake.

Fig. 1G was a RT-qPCR experiment to make comparisons about the mRNA levels of Nos1 between control and pilocarpine-induced seizure groups. As you see, there were 8 different time points, so we could not conduct all the tests in one time. After we collected all data, we integrated and make analysis. In this process, when we pasted the raw data of different time points, we made a mistake that pasted the values of control 14d to 7d.

We feel very sorry for this mistake and fortunately this mistake had no influence to the conclusion of this figure.

We hope this negligence would not affect the acceptance for our manuscript.

Thank you again.

Best regards,

Fan Meng

28th Oct 2024

Dear Dr. Meng,

Thank you for submitting your revised files. I am pleased to inform you that your manuscript is accepted for publication and is now being sent to our publisher to be included in the next available issue of EMBO Molecular Medicine.

Yours sincerely,

Lise Roth
